# The Oncology Biomarker Discovery framework reveals cetuximab and bevacizumab response patterns in metastatic colorectal cancer

Alexander J. Ohnmacht[1,2,10], Arndt Stahler[3,10], Sebastian Stintzing [3,4,10], Dominik P. Modest [3], Julian W. Holch[4,5], C. Benedikt Westphalen [5], Linus Hölzel[1], Marisa K. Schübel[1,2], Ana Galhoz[1,2], Ali Farnoud[1], Minhaz Ud-Dean[1], Ursula Vehling-Kaiser[6], Thomas Decker[7], Markus Moehler[8], Matthias Heinig[1], Volker Heinemann[5] ✉ & Michael P. Menden [1,2,9] ✉

Precision medicine has revolutionised cancer treatments; however, actionable biomarkers remain scarce. To address this, we develop the Oncology Biomarker Discovery (OncoBird) framework for analysing the molecular and biomarker landscape of randomised controlled clinical trials. OncoBird identifies biomarkers based on single genes or mutually exclusive genetic alterations in isolation or in the context of tumour subtypes, and finally, assesses predictive components by their treatment interactions. Here, we utilise the open-label, randomised phase III trial (FIRE-3, AIO KRK-0306) in metastatic colorectal carcinoma patients, who received either cetuximab or bevacizumab in combination with 5-fluorouracil, folinic acid and irinotecan (FOLFIRI). We systematically identify five biomarkers with predictive components, e.g., patients with tumours that carry chr20q amplifications or lack mutually exclusive ERK signalling mutations benefited from cetuximab compared to bevacizumab. In summary, OncoBird characterises the molecular landscape and outlines actionable biomarkers, which generalises to any molecularly characterised randomised controlled trial.

Precision medicine aims to tailor therapeutic interventions to specific patient subgroups defined by predictive biomarkers detected in tumours. Accordingly, strategies are required to identify such patient subgroups systematically[1]. For performing subgroup analysis and exploratory biomarker discovery, the European Medicines Agency (EMA) has provided specific guidelines[2]. According to these, biological knowledge should underpin subgroup definitions, and subgroup-specific effects in late-stage clinical trials should still be interpreted

[1]Computational Health Center, Helmholtz Munich, 85764 Neuherberg, Germany. [2]Department of Biology, Ludwig-Maximilians University Munich, 82152 Martinsried, Germany. [3]Charité Universitätsmedizin, corporate member of Freie Universität Berlin and Humboldt-Universität zu Berlin, Department of Hematology, Oncology, and Cancer Immunology, Charitéplatz 1, 10117 Berlin, Germany. [4]German Cancer Consortium (DKTK), partner sites Berlin and Munich, German Cancer Research Center (DKFZ), 69120 Heidelberg, Germany. [5]Department of Medicine III and Comprehensive Cancer Center Munich, University Hospital, Ludwig-Maximilians University Munich, 81377 Munich, Germany. [6]Oncological Practice, 84028 Landshut, Germany. [7]Oncological Practice, 88212 Ravensburg, Germany. [8]Department of Medicine I and Research Center for Immunotherapy (FZI), Johannes Gutenberg-University Clinic, 55131 Mainz, Germany. [9]Department of Biochemistry and Pharmacology, University of Melbourne, Victoria 3010, Australia. [10]These authors contributed equally: Alexander J. Ohnmacht, Arndt Stahler, Sebastian Stintzing. ✉e-mail: volker.heinemann@med.uni-muenchen.de; michael.menden@helmholtz-munich.de

with caution owing to the exploratory and retrospective nature of the analyses. For this purpose, a large number of computational methods have been proposed and discussed[3–5], e.g., tree-based methods using recursive partitioning[6–8], virtual twins[9], outcome weighted methods[10,11], causal forests[12] and metalearners for estimating heterogeneous treatment effects[13]. However, most of these computational methods neglect cancer biology, i.e., exploiting the molecular landscape of a clinical trial and customising models to cancer subtypes and mutational patterns.

Clinical outcomes of patients with metastatic colorectal cancer (mCRC) significantly improved upon the introduction of targeted treatments, including anti-EGFR and anti-VEGF directed monoclonal antibodies such as cetuximab and bevacizumab, respectively[14]. Tumours of colorectal cancer patients were shown to exhibit, for instance, either *KRAS* or *NRAS* mutations (referred to as *RAS* mutations) with a rate of about 50%, which tend to occur mutually exclusive[15,16]. These *RAS* mutations are clinically approved predictive biomarkers of resistance against anti-EGFR directed monoclonal antibodies such as cetuximab[17]. Bevacizumab has been reported to improve progression-free survival in first-line mCRC trials[18]; however, no comparable biomarker has been depicted yet.

In this study, we focused on the open-label randomised phase III clinical trial FIRE-3. Here, patients with *KRAS* exon 2 wild-type mCRC were randomised to receive either cetuximab or bevacizumab in combination with 5-fluorouracil, leucovorin and irinotecan (FOLFIRI) as a first-line regimen. Several retrospective subgroup analyses revealed potential prognostic and predictive biomarkers based on tumour DNA and clinical characteristics, such as the relevance of the molecular status, i.e., alterations other than *KRAS* exon 2, such as *KRAS* exon 3-4, *NRAS* exon 2-4 and BRAF V600E, or primary tumour sidedness[19–23]. For example, targeting EGFR in *RAS* wild-type mCRC tumours located in the left hemicolon (left-sided) was shown to be beneficial, whilst *RAS* wild-type tumours located in the right colon (right-sided) were less likely to respond[24]. Additionally, in the more recent FIRE-4.5 study, it was demonstrated that patients with BRAF V600E mutant tumours may benefit from the treatment with 5-fluorouracil, oxaliplatin, leucovorin and irinotecan (FOLFOXIRI) backbone plus bevacizumab[25], whereas in contrast, these patients lacked benefits from cetuximab[26,27]. This hints towards tumour subtype-specific interactions and alternative mechanisms to acquire EGFR inhibitor resistance[28].

Previously proposed tumour subtypes in colorectal adenocarcinoma are based on the gene expression-derived consensus molecular subtypes (CMS) and could identify subtypes that reflected distinct tumour biology[29]. Recently, the prognostic value of CMS has been confirmed in the FIRE-3, CALGB/SWOG 80405 and AGITG MAX clinical trials for FOLFIRI combined with either cetuximab or bevacizumab[21,30,31]. In particular, CMS4 patients with *RAS* wild-type have shown a significantly longer overall survival when treated with cetuximab compared to bevacizumab in metastatic disease[21]. However, the clinical translation of the CMS classification of colorectal cancer is still in its infancy and is further investigated in multiple clinical trials[32]. These sparse results have illustrated that modelling interactions between somatic alterations and tumour subtypes can yield insights into complex biomarkers and highlight the urgent need for computational frameworks to systematically decipher the molecular landscape, tumour subtypes and biomarkers. Thus, we hypothesised that predictive response biomarkers may be revealed by systematically deconvoluting cancer genetic events and tumour subtypes within a clinical trial.

Here, we present the Oncology Biomarker Discovery (OncoBird) framework, which empowers the systematic identification of actionable biomarkers for clinical trials in oncology. OncoBird is publicly available as a software package at https://github.com/MendenLab/OncoBird and a demo run is available at https://codeocean.com/ capsule/9911222/tree/v1. Furthermore, users can run a graphical user interface within a docker container (Supplementary Fig. 1).

The OncoBird workflow is divided into five distinct steps: it systematically (1) investigates the molecular landscape of a clinical trial, i.e., copy number alterations, somatic mutations, mutually exclusive patterns and predefined tumour subtypes; (2) identifies biomarkers within a treatment arm based on genetic alterations, and (3) in relation to the predefined tumour subtypes; consecutively, (4) evaluates their predictive component across treatment arms; and finally, (5) it comprehensively corrects for multiple hypothesis testing and adjusts treatment effects of biomarkers based on resampling methods. To enhance the biological signal, this analysis integrates the molecular and biomarker landscape of cancer clinical trials by customising models to established cancer subtypes and mutational patterns. In essence, OncoBird yields subtype-specific biomarkers with treatment benefits in an interpretable and transparent manner and therefore operates complementary to existing methods. The utility of OncoBird is exemplified by the application to the FIRE-3 clinical trial, generalises to the ADJUVANT clinical trial[33–35], and in fact, would generalise to any molecularly characterised randomised controlled trial (RCT) in oncology.

## Results

OncoBird is applicable to RCTs accompanied with molecular characteristics, including genetic sequencing panels which yield copy number alterations and somatic driver mutations (Fig. 1a, b). In addition, a second layer of stratification can be supplied in the form of predefined tumour subtypes (Fig. 1a). Then, OncoBird systematically assesses the genetic landscape in the context of tumour subtypes (Fig. 1c) and outlines the biomarker landscape across multiple clinical responses (Fig. 1d), i.e., time-to-event data (overall or progression-free survival; "Methods"), and binary variables capturing treatment success (objective response rate; "Methods").

Here, we leveraged the FIRE-3 RCT, including 752 mCRC patients who have been treated with FOLFIRI and either cetuximab or bevacizumab. We defined tumour subtypes based on CMS[21], and tumour sidedness, i.e., left- or right-sided mCRC. In addition, 373 tumours were genetically characterised, i.e., the mutational status of 277 frequently altered cancer genes. To reveal the biomarker landscape, we employed the following stratification and modelling strategies (Supplementary Data 1; "Methods"): We first investigated each alteration for stratifying patients by their prognosis within each treatment arm (Fig. 1e). Consecutively, we inspected alterations in tumour subtypes (Fig. 1f), revealing subtype-specific biomarkers. Finally, we tested for treatment interactions to reveal biomarkers with predictive effects (Fig. 1g). Importantly, subtypes and genetic alterations ought to be independent of the treatment assignment. The molecular landscape and individual treatment arm analysis could be applied to any trial design without limitations.

Exemplified with a well-established biomarker of cetuximab response[17], *RAS* wild-type mCRC patients showed longer overall survival (Fig. 1h; $p = 0.0002$, HR = 0.53 [0.38–0.73]). Consistent with a previous study[36] and more recently defined treatment guidelines for mCRC[37], the cetuximab overall survival (OS) benefit for patients with *RAS* wild-type tumours was conserved in left-sided tumours (Fig. 1i; $p = 7.6 \times 10^{-5}$, HR = 0.44 [0.29–0.66]). Furthermore, we observed interactions between *RAS* mutations and the treatment arm in left-sided tumours ($p_{int} = 0.07$): Cetuximab remained superior to bevacizumab in *RAS* wild type and left-sided tumours (Fig. 1j; $p = 0.05$, HR = 0.73 [0.52–1.00]) in terms of OS, whilst bevacizumab and cetuximab achieved comparable OS for patients with *RAS* mutant and left-sided tumours (Supplementary Fig. 2; $p = 0.32$, HR = 1.22 [0.85–1.75]).

Whilst we particularly focused on the FIRE-3 trial in colorectal cancer, we also demonstrate the generalisability of OncoBird by applying it with the same default biomarker thresholds to the

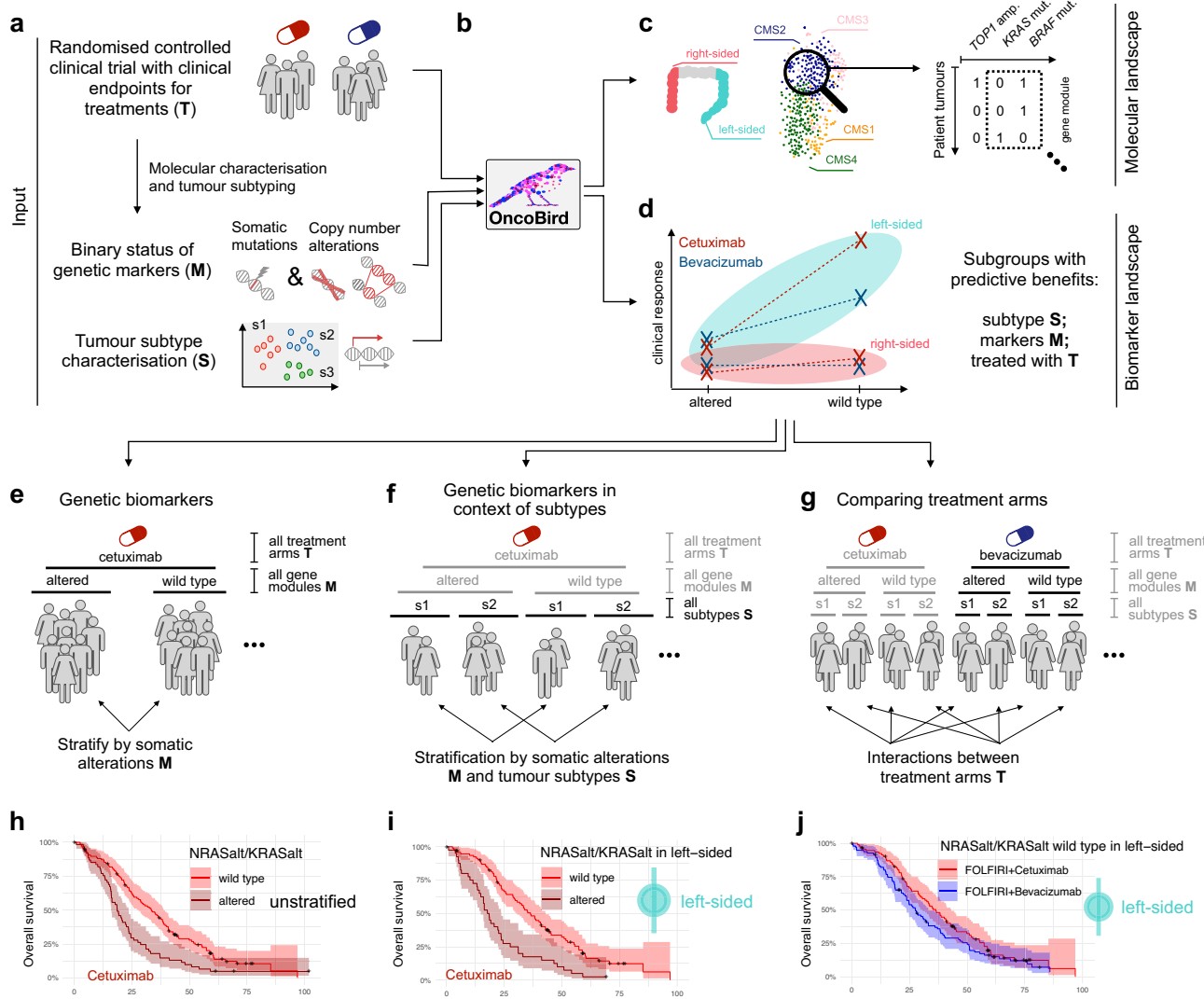

**Fig. 1 | The Oncology Biomarker Discovery (OncoBird) workflow. a** Patients in clinical trials were treated with (T) two treatment regimens with measured clinical endpoints. Subsequently, their tumours are characterised according to (M) tumour genetic alterations (somatic mutations and copy number alterations) and (S) tumour subtypes. **b** With this input, OncoBird outlines **c** the molecular landscape and **d** the biomarker landscape. For the latter, **e** somatic alterations are explored for a differential patient prognosis for each treatment arm. **f** Consecutively, for each treatment arm, subtype-specific biomarkers are derived. **g** Finally, interactions between treatment arms are examined. The grey shadings indicate the data included in the previous analysis step. Here, this is exemplified in the FIRE-3 clinical trial using Kaplan–Meier plots, including 95% confidence intervals (CI) and summary statistics of the Cox regression models. **h** *RAS* mutations are established biomarkers of cetuximab resistance. **i** Patients with *RAS* wild-type tumours showed a better prognosis when treated with cetuximab within left-sided tumours compared to right-sided tumours. In addition, **j** the *RAS* wild-type subpopulation in left-sided tumours showed benefits when treated with cetuximab compared to bevacizumab.

ADJUVANT clinical trial ("Methods"), which explored gefitinib in non-small cell lung cancer (NSCLC)[33–35]. The ADJUVANT study reported predictive components of five alterations, i.e., *TP53* mutations, *RB1* alterations and copy number amplifications of *NKX2-1*, *CDK4* and *MYC*[35]. Four out of five biomarkers were concordantly identified for disease-free survival with OncoBird (FDR$_{int}$ < 0.2; Supplementary Data 2; Supplementary Fig. 3–6). In addition, OncoBird suggests that the mutual exclusivity patterns play a role in the biomarker landscape of NSCLC (Supplementary Fig. 3c, d). In more detail, we observed gefitinib benefits in tumours that were characterised by mutations in either *TP53*, *SMAD4* or *CDK4* amplifications (*p* = 0.0002, HR = 0.37 [0.21–0.63]; Supplementary Data 2; Supplementary Figs. 5c and 6a), for which the resampling-based adjustment of the conditional average treatment effect yielded $p_{adj}$ = 0.001 with HR = 0.32 [0.14–0.86] (Supplementary Data 2; "Methods"). These findings highlight the accessibility, reproducibility and interoperability of OncoBird.

## The molecular landscape of the FIRE-3 clinical trial

Leveraging OncoBird, we assessed the genetic landscape of patient tumours in the FIRE-3 clinical trial. In total, 373 tumours were genetically characterised, including 31 frequently altered cancer genes observed in at least 12 patients (Fig. 2a). We observed amplifications in chromosome arm 20q (chr20q) in 74/373 tumours (19.8%), which includes *SRC*, *TOP1*, *BCL2L1*, *ZNF217*, *AURKA*, *GNAS* and *ARFRP1* (Fig. 2a). Indeed, chr20q amplifications have been reported to define a distinct subtype of left-sided colon cancers[38]. In addition, we identified 39 mutually exclusive somatic alterations (gene modules) using the Mutex algorithm (Fig. 2b; "Methods")[39], thus grouping low frequent but functionally similar somatic events within a signalling pathway. We could confirm that chr20q amplifications were mutually exclusive to somatic mutations in the ERK signalling pathway (*KRAS*, *NRAS* or *BRAF*; *p* = 0.0002, Fisher's exact test).

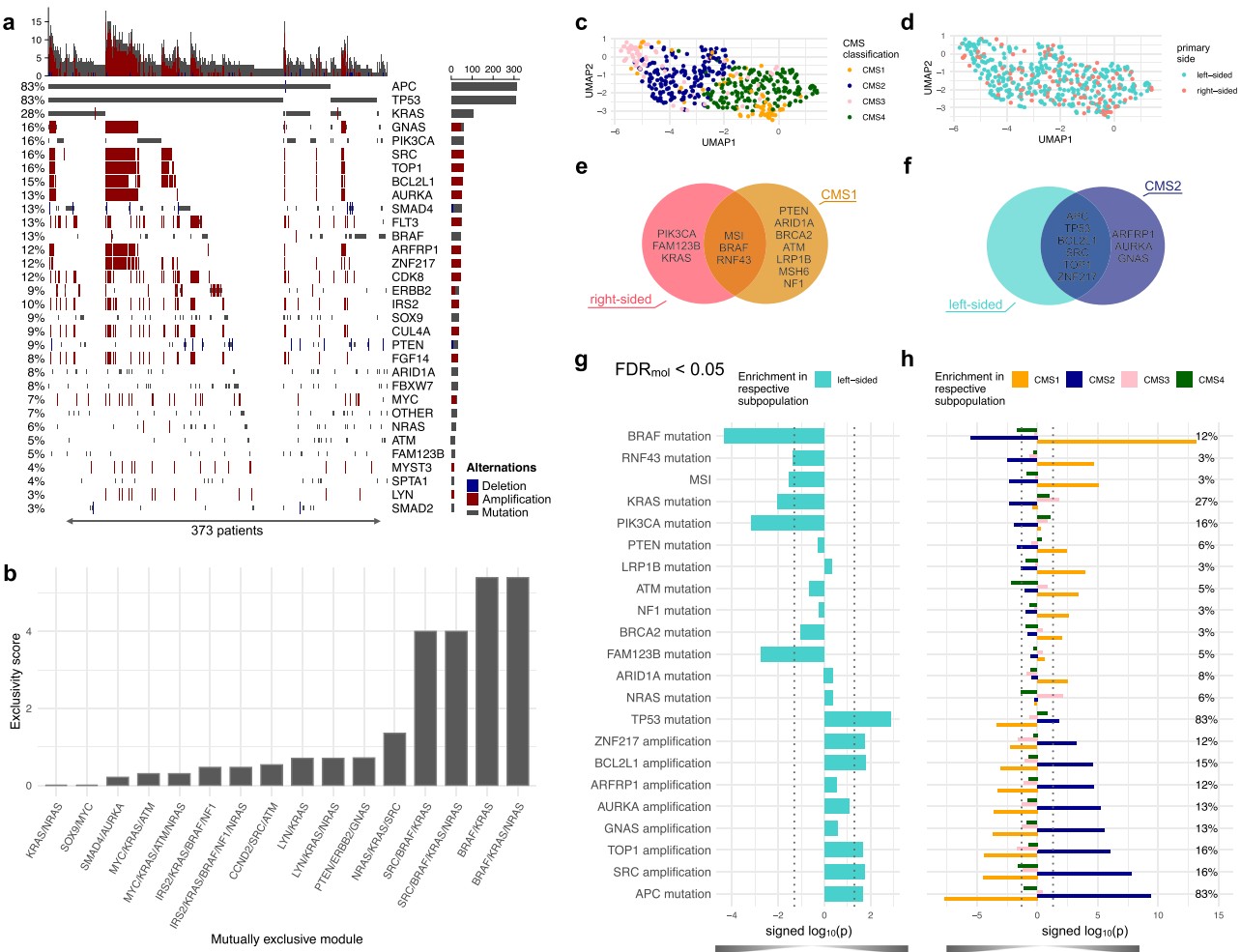

**Fig. 2 | Molecular landscape of the FIRE-3 clinical trial. a** Oncoprint of 373 mCRC tumours, including mutations and copy number alterations detected in more than 12 tumours. **b** The mutually exclusive alteration patterns were derived with the Mutex algorithm. Gene expression profiles of 451 mCRC tumours are annotated by **c** the consensus molecular subtypes (CMS) and **d** the primary tumour side. **e** Venn diagram showing all enriched somatic alterations for CMS1 and right-sided tumours, and **f** enriched somatic alterations for CMS2 and left-sided tumours. **g** Frequently altered cancer genes tested for enrichment in left- or right-sided tumours, and **h** tested against CMS subtypes using one-sided hypergeometric tests. Source data for the figure panels are provided as Source Data file.

In addition, we analysed 451 gene expression profiles and showed consistency with their derived CMS subtypes (Fig. 2c), whilst the primary tumour side displayed a heterogeneous gene expression pattern (Fig. 2d). Right-sided tumours were particularly enriched in CMS1 tumours ($p = 0.009$, hypergeometric test; Supplementary Fig. 7) and depleted in CMS2 tumours ($p = 0.007$, hypergeometric test; Supplementary Fig. 7).

The concordance between right-sided tumours and CMS1 (Fig. 2e) was reflected by genetic alterations that were enriched in both tumour subtypes. Microsatellite instabilities (MSI) and somatic mutations in *BRAF* and *RNF43* were enriched in both CMS1 and right-sided tumours ($FDR_{mol} < 0.05$, hypergeometric test). Additionally, mutations in *PIK3CA*, *FAM123B* and *KRAS* were only associated with right-sided tumours (Fig. 2e; $FDR_{mol} < 0.05$, hypergeometric test). In contrast, the similarity of left-sided tumours and CMS2 (Fig. 2f) was characterised by mutations in *APC*, *TP53* and chr20q amplifications (*SRC*, *TOP1*, *BCL2L1*, *ZNF217*), which were all significantly enriched in both left-sided and CMS2 tumours (Fig. 2g, h; $FDR_{mol} < 0.05$, hypergeometric test). Somatic mutations in *PTEN*, *ARID1A*, *ATM*, *LRP1B*, *BRCA2* and *NF1* did not show a preference for a particular primary tumour side, but were enriched in CMS1 tumours (Fig. 2h), and were associated with an increased tumour mutational burden ($p = 0.008$, $p = 0.002$, $p = 0.017$, $p = 0.0001$, $p = 0.010$ and $p = 0.051$, respectively, Fisher's exact test).

In summary, leveraging OncoBird and investigating patterns of genetic events in tumour subtypes revealed meaningful tumour biology. For example, mutations of either *BRAF* or *KRAS* promote ERK signalling and therefore occur mutually exclusive. *BRAF* mutations were predominantly found in CMS1, but nevertheless, 27 out of 53 *BRAF* mutant tumours were distributed among CMS2-4. Therefore, it is of utmost importance to gain an enhanced understanding of the molecular landscape of mCRC prior to the interpretation of biomarkers, which is further empowered by OncoBird.

## Genetic biomarkers of cetuximab

First, independent of tumour subtypes, we assessed single genes and mutually exclusive gene modules (Fig. 2a, b) as biomarkers for cetuximab. For this, we leveraged Cox proportional hazards regression and logistic regression models ("Methods"), considering overall survival (OS; Fig. 3a–h), progression-free survival (PFS; Supplementary Fig. 8) and the objective response rate (ORR; Supplementary Fig. 9). We quantified effect sizes by hazard ratios (HR) for survival data and odds ratios (OR) for binary data including 95% confidence intervals (Supplementary Data 3).

The clinically established resistance biomarkers of cetuximab were recovered, i.e., mutations in *RAS* (either *KRAS* or *NRAS*) referred to a poorer OS in the cetuximab treatment arm (Fig. 1h; $p = 0.0002$,

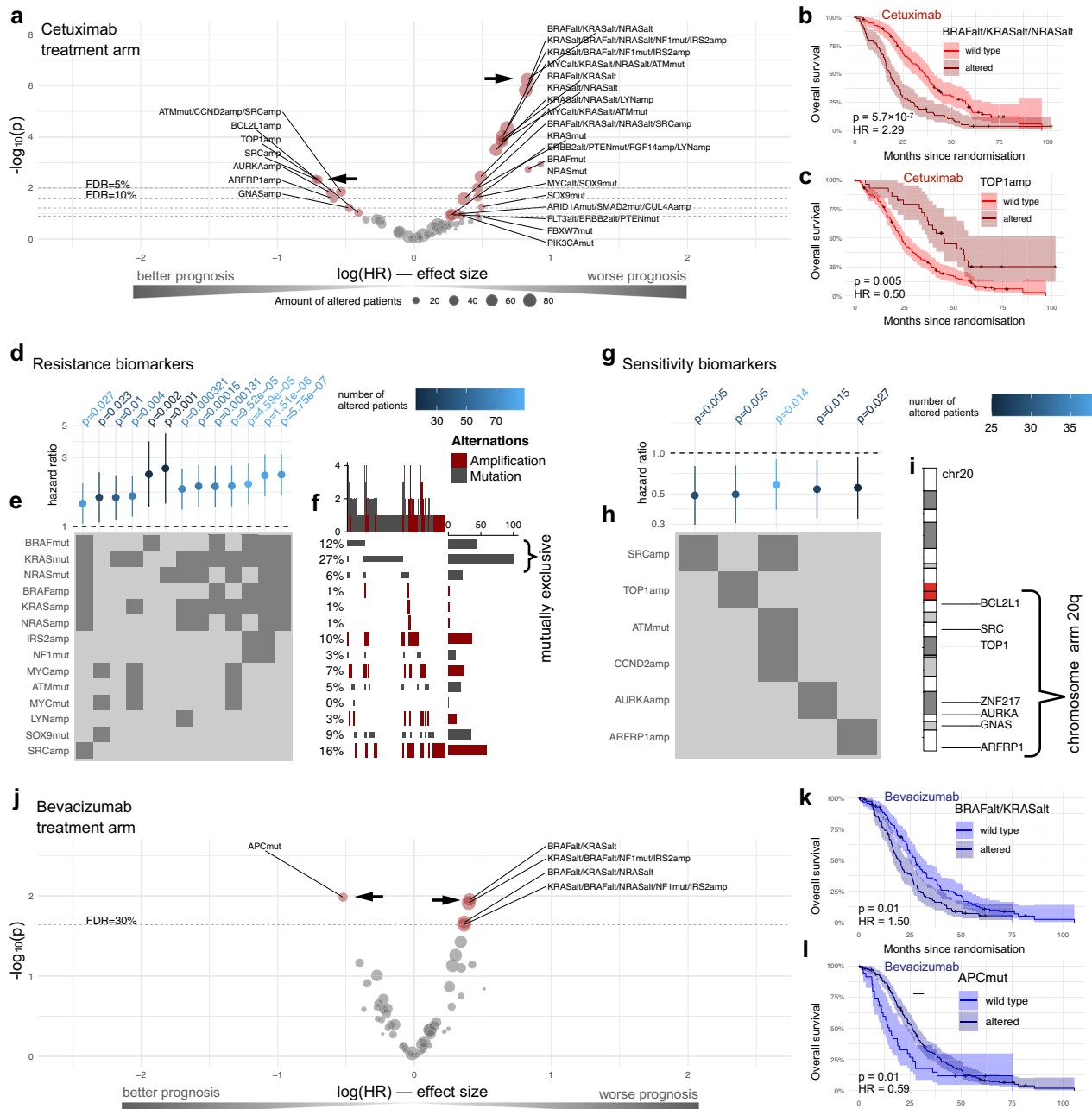

**Fig. 3 | Identification of genetic biomarkers for FOLFIRI plus cetuximab or bevacizumab. a** Volcano plot for genetic biomarkers of cetuximab in the form of mutually exclusive gene modules or single gene mutations. Each point shows the effect of a particular group of alterations summarised by its hazard ratio derived by the Cox regression models and its raw *p*-value derived by a Wald test. Exemplifying the most significant associations, Kaplan–Meier plots, including 95% confidence intervals (CI) and summary statistics of the Cox regression models, are shown for **b** the mutually exclusive module consisting of *RAS* and *BRAF* mutations, and **c** the amplification of *TOP1* treated with cetuximab. For investigating the biomarker composition, we focus on **d** resistance biomarkers of FOLFIRI plus cetuximab with FDR_cet < 0.1, showing their hazard ratios and 95% CIs. For these, **e** the composition of mutually exclusive genes is indicated by dark grey colour, and **f** an oncoprint

highlighting mutational frequencies of biomarker combinations is shown. In like manner, **g** cetuximab sensitivity biomarkers, shown by their hazard ratios and 95% CIs, and **h** their composition are summarised. **i** Karyoplot showing transcription start sites of co-amplified genes on chromosome 20q. **j** Volcano plot of the genetic biomarkers of bevacizumab with FDR_bev < 0.3, shown in brown colour by their hazard ratios derived by the Cox regression models and their raw *p*-values derived by a Wald test. Kaplan–Meier plot including 95% CIs and summary statistics of the Cox regression models of **k** mutations in *KRAS or BRAF* and **l** *APC* mutations treated with bevacizumab. The compositions of bevacizumab biomarkers are shown in Supplementary Fig. 10a, b. A Source Data file is provided, which contains the source data for the figure panels and the sample sizes of the conducted statistical tests.

HR = 1.90 [1.36–2.65], FDR_cet < 0.1). In addition, we confirmed that *BRAF* mutations are mutually exclusive to *RAS* mutations (Fig. 2b; *p* = 0.0008, Fisher's exact test), and both contributed to a poor OS when treated with cetuximab (Fig. 3b; $p = 5.7 \times 10^{-7}$, HR = 2.29 [1.65–3.16], FDR_cet < 0.1), which has been consistently observed in an independent cohort[40].

Most resistance biomarker modules grouped mutations in *KRAS* and *BRAF* (FDR_cet < 0.1). In addition, we found a gene module including mutations in *SOX9* and *MYC* amplifications, for which mutant tumours displayed a worse prognosis based on OS (Fig. 3d, e; *p* = 0.02, HR = 1.50 [1.07–2.37], FDR_cet < 0.1). By inspecting their oncoprint (Fig. 3f), 27/59 tumours harboured mutations in either *SOX9* or *MYC* and were wild-

type in either *BRAF*, *KRAS* or *NRAS*, hinting towards an alternative cetuximab resistance mechanism.

In addition, we found *TOP1* amplifications to be a strong predictor of a prolonged OS for treatment with cetuximab (Fig. 3c; $p = 0.005$, HR = 0.50 [0.30–0.81], FDR$_{cet}$ < 0.1). In fact, we could identify multiple co-amplifications that showed prognostic value for the cetuximab treatment arm, which are located on chromosome 20q. Among the most predictive amplifications for a longer OS were *SRC*, *TOP1*, *AURKA* and *ARFRP1* (Fig. 3g–i; Supplementary Data 3). Consistent trends were observed with *SRC* amplifications in PFS ($p = 0.10$, HR = 0.69 [0.44–1.07], median PFS wild-type tumours 9.6 months vs mutants 11.1 months) and ORR ($p = 0.18$, OR = 0.45 [0.14–1.45], ratio ORR wild-type 0.66 vs mutant tumours 0.83).

### Genetic biomarkers of bevacizumab

Analogously to the cetuximab biomarker analysis, for the bevacizumab treatment arm, we also built Cox proportional hazards regression models ("Methods") applied to OS (Fig. 3j–l; Supplementary Fig. 10) and PFS (Supplementary Fig. 11), and logistic regression models for ORR (Supplementary Fig. 12). For exploring bevacizumab biomarker trends, we employed a lenient threshold of FDR$_{bev}$ < 0.3, which deviates from the default setting ("Methods"). The mutually exclusive module of *KRAS* and *BRAF* mutations showed lower OS (Fig. 3j, k; $p = 0.01$, HR = 1.50 [1.10–2.04], FDR$_{bev}$ < 0.3), which is consistent with literature reports[41,42]. A better predictor for poor OS was the *APC* wild-

type status for tumours treated with FOLFIRI plus bevacizumab (Fig. 3j, l; $p = 0.01$, HR = 1.69 [1.14–2.50], FDR$_{bev}$ < 0.3).

### Subtype-specific biomarkers of cetuximab and bevacizumab

The previous analyses focused on genetic biomarkers in isolation, whilst here, we investigated them within the context of tumour subtypes ("Methods"). In FIRE-3, tumour subtypes are defined as either left- or right-sided tumours, or alternatively, classified according to the consensus molecular subtypes, i.e., CMS1-4 ("Methods")[29]. Here, we tested stratifications based on each single gene or gene module within tumour subtypes for OS (Fig. 4a, b), PFS (Supplementary Fig. 13) and ORR (Supplementary Fig. 14).

In total, we found 38 subtype-specific biomarkers of cetuximab for OS (FDR$_{cet}$ < 0.1; "Methods"). In particular, we recovered favourable OS of CMS2 patients treated with cetuximab (Fig. 4a), if their tumours additionally carried chr20q amplifications, i.e., *ARFRP1* (Fig. 4c; $p = 0.01$, HR = 0.32 [0.13–0.77], FDR$_{cet}$ < 0.1), *TOP1* (Supplementary Fig. 15a; $p = 0.01$, HR = 0.34 [0.15–0.74], FDR$_{cet}$ < 0.1) and *SRC* (Supplementary Fig. 15b; $p = 0.01$, HR = 0.37 [0.17–0.78], FDR$_{cet}$ < 0.1). Additionally, CMS4 *KRAS* mutant tumours treated with cetuximab showed worse OS (Fig. 4d; $p = 0.002$, HR = 2.60 [1.44–4.70], FDR$_{cet}$ < 0.1) and PFS (Supplementary Fig. 13a, c).

For reporting bevacizumab biomarker trends, we employed a lenient false discovery rate (FDR$_{bev}$ < 0.3), which deviates from the conservative OncoBird default setting ("Methods"). Tumours with

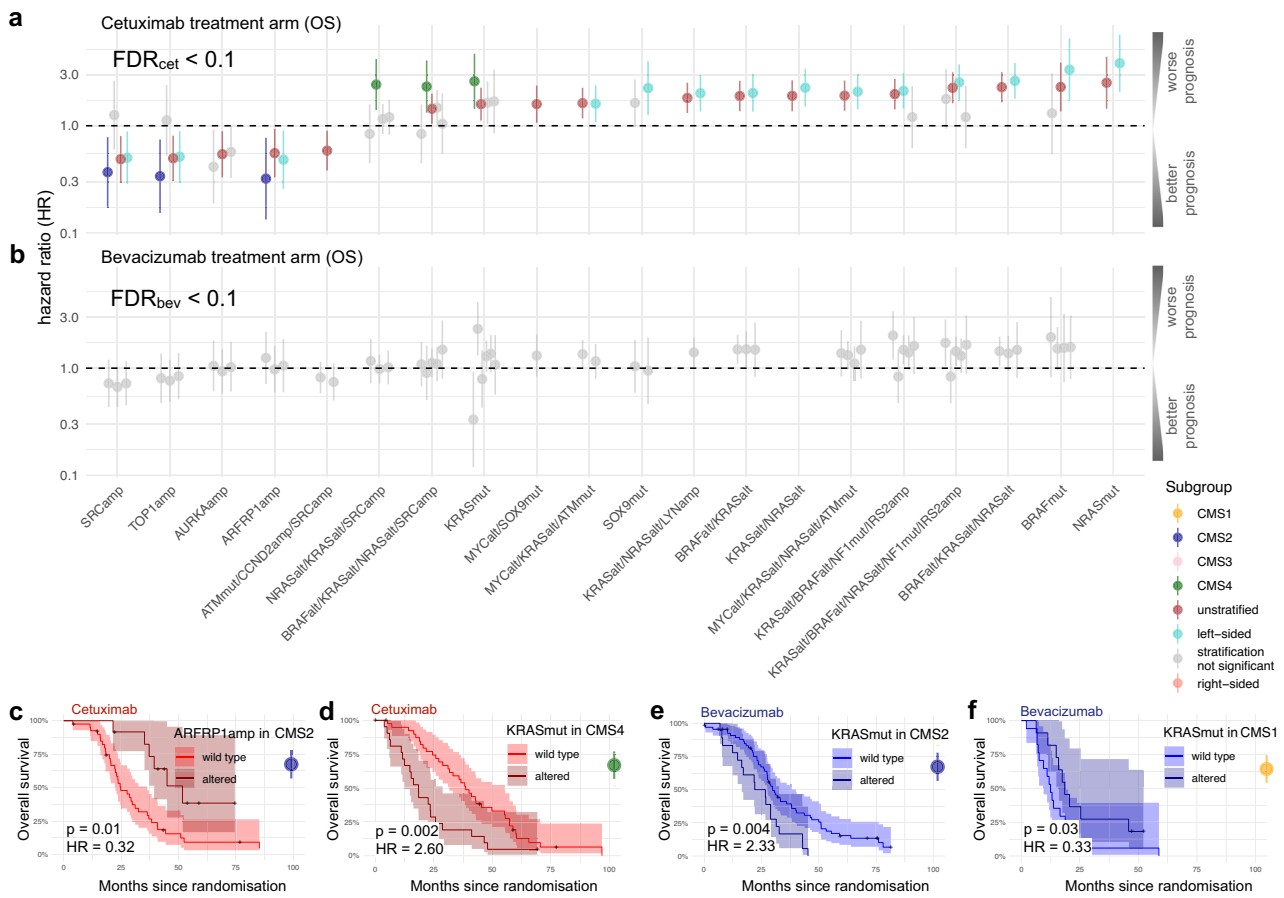

**Fig. 4 | Identification of subtype-specific genetic biomarkers for FOLFIRI plus cetuximab or bevacizumab.** Subtype-specific genetic biomarkers for OS of **a** cetuximab and **b** bevacizumab using hazard ratios including 95% confidence intervals (CI) derived from single Cox regression models. Subtypes are defined by either the primary tumour side, CMS or unstratified (reference model). Kaplan–Meier plots including 95% CIs, hazard ratios and raw *p*-values derived by

Wald tests from the Cox regression models of subtype-specific genetic biomarkers for **c** *ARFRP1* in CMS2, **d** *KRAS* in CMS4, **e** *KRAS* in CMS2 and **f** *KRAS* in CMS1 in either the cetuximab or bevacizumab treatment arm. A Source Data file is provided, which contains the source data for the figure panels and the sample sizes of the conducted statistical tests.

*KRAS* mutations classified as CMS2 tended to show worse OS when treated with bevacizumab (Fig. 4e; $p = 0.004$, HR = 2.33 [1.31–4.15], FDR$_{bev}$ < 0.3). In contrast, *KRAS* mutated tumours classified as CMS1 tended to show a longer OS compared to wild-type tumours when treated with bevacizumab (Fig. 4f; $p = 0.03$, HR = 0.33 [0.12–0.93], FDR$_{bev}$ < 0.3).

### Predictive components of biomarkers

For assessing the predictive component of response biomarkers, here, we compared the cetuximab and bevacizumab treatment arms against each other by focusing on interactions between genetic alterations in the context of tumour subtypes ("Methods"). Subsequently, we compared the prognosis of both inhibitors for each subgroup according to the interaction biomarkers, thus assessing potential treatment benefits. In addition, we corrected the conditional average treatment effects in the identified subgroups using resampling methods to obtain multiplicity-adjusted *p*-values and bias-corrected confidence intervals ("Methods"). The results were summarised for OS (Fig. 5a, b) and PFS (Supplementary Fig. 16), whereas no significant interactions were detected for ORR. In total, we found five putative interactions (Supplementary Data 4; FDR$_{int}$ < 0.2; "Methods"). For reporting other biomarker trends, we also included summary statistics of 57 subgroups with a lenient threshold of FDR$_{int}$ < 0.6, which deviates from the default setting (Supplementary Data 3).

For example, we found predictive value of chr20q amplifications in CMS2 tumours treated with FOLFIRI plus cetuximab (Fig. 5a, b), which is evident by the significant interactions of *TOP1* ($p_{int}$ = 0.07, FDR$_{int}$ < 0.2) and *ARFRP1* ($p_{int}$ = 0.01, FDR$_{int}$ < 0.2). *ARFRP1* amplifications showed the largest predictive component among the chr20q

amplifications. Accordingly, we observed longer OS in the cetuximab treatment arm compared to bevacizumab in CMS2 (Fig. 5a, c; *ARFRP1*: $p = 0.003$, HR = 0.21 [0.07–0.59], FDR$_{int}$ < 0.2; Supplementary Data 3). The resampling-based adjusted treatment effect confirmed this observation and yielded a hazard ratio in this subgroup of HR = 0.21 [0.09–0.54] with $p_{adj}$ = 0.04 (Fig. 5a, c). Previous reports have indicated a prognostic value of chr20q amplifications in colorectal cancer patients[38,43], whilst OncoBird yielded additional evidence that they harbour a predictive component.

Another interaction example was tumours with *KRAS* mutations that showed CMS-specific responses. In CMS4, patients with *KRAS* wild-type tumours responded better to cetuximab compared to patients treated with bevacizumab (Fig. 5b, d; *KRAS* wild types: $p = 0.02$, HR = 0.57 [0.35–0.93]; $p_{int}$ = 0.02, FDR$_{int}$ < 0.2), for which the resampling-based adjusted treatment effect yielded HR = 0.70 [0.25–2.35] with $p_{adj}$ = 0.14 (Fig. 5b, d). Our results suggest a predictive role of *KRAS* mutations in CMS4 for cetuximab, which we also identified for PFS (Supplementary Fig. 16c, d). Notably, modules containing alterations in *NRAS*, *BRAF* and *SRC* showed similar statistics since only four, eight and twelve mutant tumours were present in CMS4. Insignificant but numerically longer OS was observed for patients with *KRAS* mutated tumours classified as CMS4 treated with bevacizumab (Fig. 5e, *KRAS* mutants: $p = 0.24$, HR = 0.66 [0.33–1.31]), with a median OS 28.3 months compared to 18.4 months when treated with cetuximab.

In order to assess the ability of OncoBird to discover the same biomarkers for different datasets, we applied 5-fold cross-validation repeated five times and extracted the ten most significant biomarkers for OS across each of the 25 models (Fig. 6a). Consistent with our

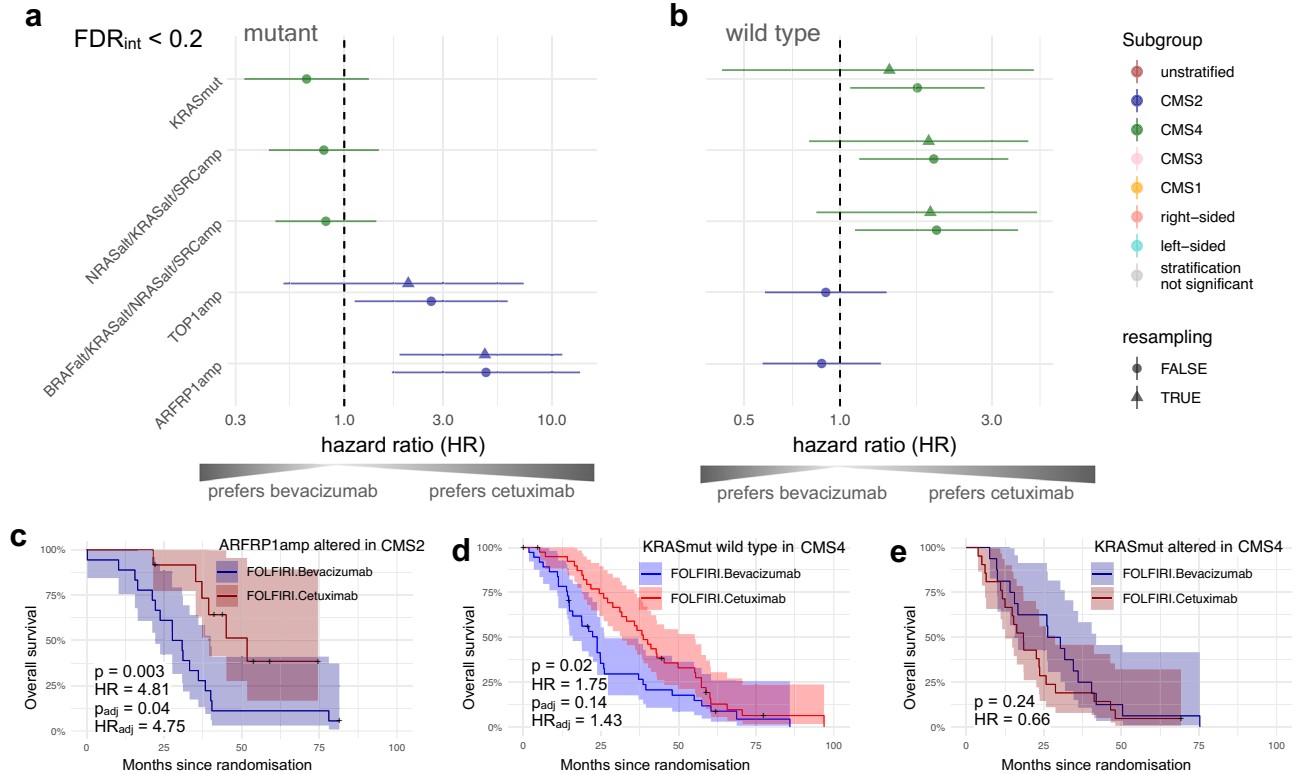

**Fig. 5 | Predictive biomarkers in the context of tumour subtypes.** Overview of interaction biomarkers (FDR$_{int}$ < 0.2) focusing on **a** mutant and **b** wild-type tumours when comparing cetuximab and bevacizumab treatment, using hazard ratios including 95% confidence intervals (CI) derived from single Cox regression models fitted on OS. Triangle points and confidence intervals were obtained from the bootstrap-based bias-correction of treatment effects. For the conducted statistical tests, the sample sizes are given in Supplementary Data 3. Here exemplified, Kaplan–Meier plots including 95% CIs, hazard ratios and raw *p*-values derived by Wald tests from the Cox regression models compare treatments in subgroups for **c** *ARFRP1* amplifications in CMS2 and **d**, **e** *KRAS* mutations in CMS4. Source data for the figure panels are provided as Source Data file.

previous findings, gene modules containing *KRAS* mutations for CMS4 were found in 21/25 training sets and chr20q amplifications in CMS2 were reproduced in 22/25 training sets (Fig. 6a).

### Benchmarking of methods for subgroup analysis

For benchmarking OncoBird, we compared it to alternative methods that can be used to investigate predictive biomarkers based on overall survival. Together with OncoBird, eight algorithms and implementations were used in order to identify subgroups with differential treatment effects, i.e., virtual twins (VT)[9], model-based partitioning (MOB)[8], an outcome-weighting method (OWE)[11], causal random forests (CRF)[12], policy learning (POL)[44], GUIDE[45] and PRISM[46] (Supplementary Table 1; "Methods"; Fig. 6b).

For the evaluation, we first derived hazard ratios for cetuximab benefit based on OS in the subgroups according to the predicted biomarkers for all methods across five times 5-fold cross-validation ("Methods"). We also focused on the current treatment guidelines for mCRC, according to which patients should receive cetuximab if their tumours are *RAS* wild-type and left-sided (std; Fig. 6b)[37]. While the treatment benefit was not significant for the std-positive subgroup

(Fig. 6b, median HR = 0.78, $p_{cv}$ = 0.129), the methods that found the highest significant benefits were OncoBird (median HR = 0.74, $p_{cv}$ = 0.046), POL (median HR = 0.81, $p_{cv}$ = 0.048), MOB (median HR = 0.83, $p_{cv}$ = 0.048) and OWE (median HR = 0.84, $p_{cv}$ = 0.049) ordered by the magnitude of the hazard ratio (Fig. 6b).

Next, we leveraged the whole dataset to identify cetuximab sensitivity biomarkers with each method and compared them to the treatment guidelines. On average, 73% of methods identified cetuximab benefit for a patient in the std-positive subgroup, whereas only 33% of methods detected further benefits in the std-negative subgroup (Fig. 6c). 7/8 (88%) methods found mutually exclusive alterations in *KRAS*, *NRAS* or *BRAF* as a predictive biomarker, from which one, two and four methods proposed this marker in conjunction with tumour sidedness, CMS and across all patients, respectively (Supplementary Table 1). Only 2/8 (25%) methods highlighted *TOP1* amplifications as a potential biomarker (Supplementary Table 1). This highlights that current subgroup analysis methods mostly recover standard clinical practice, whilst sparsely identifying complementary predictive subgroups, thus highlighting the unmet need for cancer biology-driven frameworks such as OncoBird.

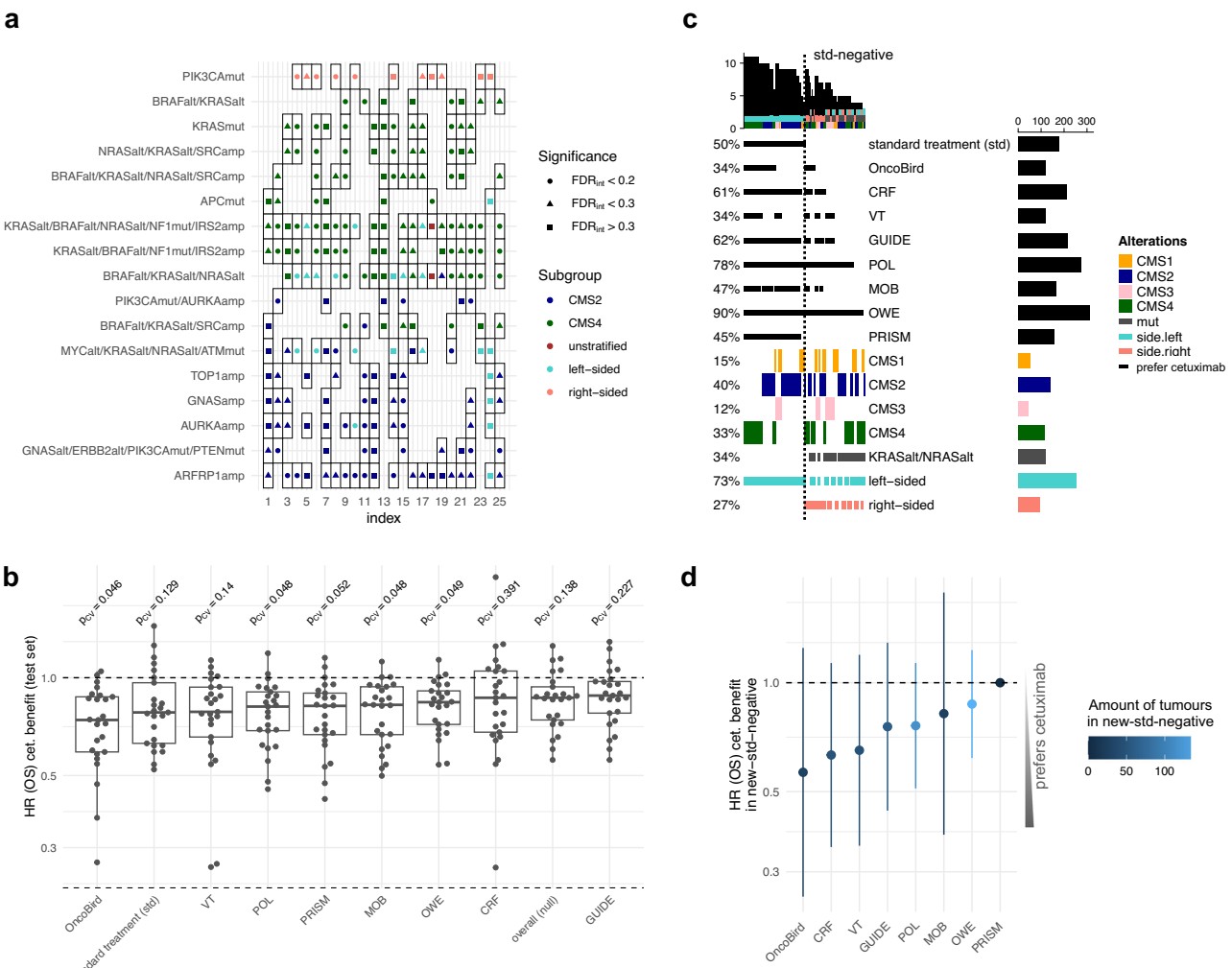

**Fig. 6 | Stability analysis and benchmark with other methods. a** The ten most significant biomarkers across 25 models of five times repeated 5-fold cross-validation. **b** Boxplots of treatment effects in terms of hazard ratios for the predicted subgroups in the 25 test sets for the benchmarked methods, including standard treatment guidelines (std) and overall across all patients (null). The centre line depicts the median; the box represents the inter-quartile range (IQR) and the whiskers the interval 1.5 times the IQR. **c** Oncoprint showing identified subgroups

for the benchmarked methods, including std, CMS subtypes, tumour sidedness and mutations in *KRAS* and *NRAS*. **d** Forest plot showing hazard ratios including 95% confidence intervals (CI) and amount of patients in the subgroup for which standard treatment is not recommended and which was found by subgroup analysis methods (new-std-negative). A Source Data file is provided, which contains the source data for the figure panels and the sample sizes of the conducted statistical tests.

Ideally, subgroup analysis should reveal subgroups with high treatment effects for refining treatment strategies and recover subgroups in the standard treatment strategy. Therefore, we evaluated the newly proposed subgroup for which standard treatment is not recommended (new-std-negative) for each method. We derived the hazard ratios for cetuximab benefit based on OS for all methods in the new-std-negative subgroups (Fig. 6d). Lower hazard ratios in new-std-negative patients indicate the discovery of off-label subgroups for which cetuximab is currently not recommended (Fig. 6d). Accordingly, OncoBird showed the numerically lowest hazard ratio HR = 0.57 ($p = 0.16$, $N = 29$) for the new-std-negative subgroup compared to all other methods (Supplementary Table 1; Fig. 6d).

In summary, many of the computational methods reproduced the clinically established biomarkers, whilst OncoBird empowers advanced biomarker identification by thoroughly integrating biological priors in the form of tumour subtypes. The simplicity of statistical models leveraged in OncoBird further increases interpretability and transparency.

## Discussion

We demonstrated that OncoBird has the capabilities to characterise the molecular and biomarker landscape of RCTs. Here exemplified, we captured the established clinical biomarkers of FIRE-3, and proposed five predictive biomarker hypotheses ($FDR_{int} < 0.2$). The biomarkers were based on either individual cancer genes or mutually exclusive patterns and exploited these genetic events in the context of well-characterised cancer subtypes. In addition, OncoBird thoroughly corrects for multiple hypothesis testing and includes resampling-based adjustments of treatment effects. In essence, OncoBird systematically investigated the molecular landscape of the FIRE-3 clinical trial, suggested biomarkers based on genetic alterations, performed a data-driven subgroup analysis, and finally, presented the results in an interpretable and intuitive way.

The statistical power of detecting biomarkers depends on the amount of screened genes and subtypes, sample sizes and magnitude of treatment effects. For example, subtype-specific analyses reduce patient subgroup sizes, thus limiting the power for detecting interactions. In order to gain statistical power to detect genetic biomarkers with low mutational frequency, Oncobird exploits mutually exclusive modules ("Methods"). Despite the use of resampling-based treatment effect estimation in the found subgroups, hypotheses generated by exploratory tools such as OncoBird ought to be replicated in independent clinical trials. Nonetheless, OncoBird identified promising patient subpopulations within the FIRE-3 and ADJUVANT clinical trials with supported biological interpretation, which indicated refined predictive benefits in cancer subtypes.

A limitation of data-driven subgroup analysis is that these may produce spurious results if not biologically interpretable[47]. To mitigate this risk, we used established tumour subtypes with distinct tumour biology in mCRC, i.e., here, the consensus molecular subtypes (CMS)[29] and primary tumour sidedness[21]. Furthermore, the grouping of functionally similar mutually exclusive somatic mutations in the cancer gene sequencing panel reinforced the identification of biological signals.

Somatic mutations may drive tumour subtypes, therefore, we systematically investigated mutational patterns within CMS1-4 and tumour sidedness. We found the majority of BRAF mutations in CMS1 and observed a co-occurrence between CMS2, left-sided tumours and amplifications in chr20q. In particular, CMS2 is characterised by a MYC signalling activation[29], which may be co-regulated by activation of the co-amplified AURKA[48]. While we predominantly identified CMS-specific biomarkers, our results suggest that both primary tumour side and CMS subtypes play a major role in the landscape of predictive biomarkers. This highlights the need for OncoBird, an integrated biomarker discovery framework, which integrates the molecular landscape of RCTs with its biomarkers.

Several genes were co-amplified in chr20q, i.e., ARFRP1, TOP1, and SRC, thus determining the drivers among these biomarker candidates is challenging. Among the prominent chr20q amplifications, TOP1 was previously proposed as a biomarker for irinotecan efficacy in metastatic colorectal cancer[49,50], which is part of the chemotherapeutic backbone of the FIRE-3 trial. Literature suggests that TOP1 abundance is essential for irinotecan-induced DNA double-strand breaks during DNA replication[51]. Additionally, TOP1 was identified to regulate EGFR through an endogenous interaction with the transcription factor c-Jun[52], which supports the hypothesis that TOP1 amplifications may be the actionable biomarker. SRC has been reported to play a role in cancer progression[53,54], whereas for ARFRP1, no functional evidence has been presented yet.

The resulting co-amplifications between these cancer genes complicate the determination of the genetic driver in chr20q. To understand the causality of cancer aetiologies, further efforts require additional treatment regimes. Alternative clinical trials for metastatic colorectal cancer often involve different chemotherapy backbones, i.e., fluorouracil, leucovorin, and oxaliplatin (FOLFOX) or fluorouracil, leucovorin, and irinotecan (FOLFIRI)[30]. The use of other therapy backbones may unravel the role of ARFRP1, TOP1 and SRC amplifications regarding better efficacy for patients treated with cetuximab. However, discrepancies may arise due to the synergism and antagonism of the different chemotherapy backbones and targeted treatments[55].

The prognostic potential of APC wild-type tumours for bevacizumab has been previously reported[56], whereas OncoBird did not yield enough evidence to support this. Indeed, a confounding factor is the enrichment of BRAF mutations in the APC wild-type tumours ($p = 1.4 \times 10^{-10}$, Fisher's exact test). This is, 48% of APC wild-type tumours were BRAF mutated in the bevacizumab treatment arm, whereas in the cetuximab treatment arm, only 29% were BRAF mutated ($p = 0.13$, Fisher's exact test). Nevertheless, independently a correlation between VEGFA expression and the mutational status of APC has been previously observed in primary colorectal tumour samples[57], suggesting that within APC mutated tumours, anti-VEGF treatment may indeed be beneficial.

Furthermore, RAS/BRAF mutations are known to harbour prognostic value in terms of overall survival[38,43]. Furthermore, we observed that KRAS mutations showed highly CMS-specific responses. In particular, treatment response differed for tumours classified as CMS4 by KRAS status, showing better response for cetuximab in KRAS wild-type and for bevacizumab in KRAS mutated tumours, respectively. CMS4 has been reported to be associated with VEGF pathway activation and is thus associated with angiogenesis[29]. Thus, patients with tumours resistant towards anti-EGFR treatment may benefit from VEGF inhibition. Further exclusion of BRAF mutations did not elevate the predictive potential of KRAS mutations in CMS4. However, the statistical power is limited by the fact that only six tumours harboured the prognostically unfavourable BRAF V600E mutation in CMS4[20].

In summary, OncoBird reproduced clinically established biomarkers and derived five hypotheses of biomarkers with predictive roles for FOLFIRI plus either cetuximab or bevacizumab. Highlighted examples include chr20q amplifications in CMS2 and KRAS mutations in CMS4, which may optimise patient stratification for metastatic colorectal cancer. Leveraging OncoBird for molecular profiling in the FIRE-3 clinical trial offered an expanded perspective on the molecular and biomarker landscape of these patients.

In the future, we anticipate that the analysis of clinical trials will progressively demand molecular patient tumour data, including predefined subtypes, highlighting the urgent need for integrative analysis tools such as OncoBird. Notably, OncoBird was developed for RCT designs and is generalisable to any trial designs for which the intention-to-treat population was defined before the treatment randomisation, i.e., the treatment assignment is independent of patient characteristics. According to this, OncoBird is applicable to modern clinical trial

designs based on master protocols[58], i.e., basket, umbrella, and platform trials if control arms are included. In an emerging landscape of predictive molecular biomarkers in cancer, OncoBird may untangle complex dependencies between somatic alterations and tumour subtypes in RCTs. Furthermore, OncoBird is generalisable to any cancer entity, thus ultimately paving the way for the next generation of precision oncology therapies.

## Methods

### Clinical data of the FIRE-3 clinical trial
FIRE-3 is an open-label, randomised phase III trial to compare first-line treatment in *KRAS* exon 2 wild-type metastatic colorectal cancer patients (mCRC) with either cetuximab or bevacizumab in combination with 5-fluorouracil, leucovorin and irinotecan (FOLFIRI). The protocol and rules of conduct were previously published[23,59] (NCT00433927). The trial was conducted in accordance with the declaration of Helsinki (1996). All translational analyses were approved by the local ethics committee (University of Munich, registry no. 186-15). All patients included in this analysis provided written informed consent. 24% and 34% of the patients had female sex in the FOLFIRI plus cetuximab and bevacizumab arm, respectively. The sex is reported according to the study protocol[23,59], and gender cannot be distinguished retrospectively. The biological sex of patients (i.e., male or female) was assigned by the study doctor of the respective trial centre and reported to the clinical research organisation (CRO). The original intention-to-treat population consisted of 752 patients in total. Primary and secondary endpoints of the FIRE-3 trial, including the median overall survival (OS) and progression-free survival (PFS), were expressed as months and defined as stated in the respective articles[23,59]. The objective response rate (ORR) was evaluated by the RECIST 1.0 criteria[23,59].

### Next-generation sequencing and genetic alterations in FIRE-3
Primary tumour tissues from 373 patients have been molecularly characterised by next-generation sequencing (NGS) with the FoundationOne® panel (Foundation Medicine, Inc., MA, USA; catalogue number not available), which identified somatic mutations and copy number alterations, i.e., deletions and amplifications, of 277 key cancer genes, microsatellite instability (MSI) and tumour mutational burden[20]. Somatic alterations were delivered in the form of binary matrices, that reflect the mutant or wild-type status of a given gene based on single nucleotide variants (SV), copy number amplifications (AMP) and deletions (DEL). MSI is an important prognostic predictor and enriched in CMS1[60], which is observed in our study, with 8 of 10 MSI-H tumours being classified as CMS1. However, MSI-H tumours are less prevalent in metastatic disease (-5%)[60]. Furthermore, only six and four MSI-H tumours were treated with bevacizumab and cetuximab, respectively.

### Gene expression profiling in FIRE-3
The genetic characterisation is complemented with gene expression profiles from Xcel® microarrays (Almac Ltd, Belfast, UK; catalogue number: 902016) in a subset of 451 patients. The clinical data and the layers of molecular characterisation led to 163 and 186 patients, which are fully characterised in the cetuximab and bevacizumab treatment arms, respectively.

### Tumour subtypes in FIRE-3
A clinically established subtype for mCRC is its primary tumour sidedness. Left-sided tumours were located in the left hemicolon, e.g., splenic flexure to the rectum. In contrast, right-sided tumours were located in the right colon, e.g., coecum to the transverse colon. In addition, annotations for molecular subtypes of mCRC were obtained from transcriptome data that has been previously used to classify patients into their closest consensus molecular subtype (CMS)[21,29] using the *cmsclassifier* package with the SSP predictor. Thereby, 24 of out 373 patient tumours were not allocated to any CMS because of missing transcriptomics data and were left out of the CMS-specific analysis. The CMS classification was used as a complementary alternative to the primary tumour side and is currently discussed in multiple clinical settings[61].

### Oncology Biomarker Discovery workflow
The Oncology Biomarker Discovery (OncoBird) framework applies to RCTs for which patients received either treatment $t \epsilon \{0,1\}$ according to the treatment indicator $T$, had an associated outcome $Y$ and can be classified into $q$ subtypes $\{s_1, \ldots, s_q\}$ according to the subtype variable $S$ (clinical data). Additionally, patient tumours are characterised by $m$ candidate genetic biomarkers $\mathbf{X} = X_1, \ldots, X_m$ with the observed biomarkers for patients $\mathbf{x} = x_1, \ldots, x_m$ (genetic data). The genetic data can be used to group functionally similar genes that can be added to the set of candidate biomarkers. Furthermore, it is possible to add additional binary features to $\mathbf{X}$ such as binarised copy number alterations with appropriate cutoffs or the MSI status of a tumour. Both genetic data (*MUT*) and clinical data (*CLIN*) are required inputs to the OncoBird workflow (Supplementary Data 1), which is described in the following sections. All implemented thresholds of OncoBird can be adjusted by the user, thus empowering more lenient or stringent analyses.

### Characterising the molecular landscape in clinical trials
OncoBird first examines genetic features $\mathbf{X}$ in tumour subtypes $\{s_1, \ldots, s_q\}$ independent of the treatment and patient response (function GET-MUTATIONS-IN-SUBTYPES in Supplementary Data 1). For examining enrichment or depletion of each genetic feature in tumour subtypes, one-sided hypergeometric tests are performed using the 'phyper' R function. Consecutively, the resulting $p$-values are corrected for multiple hypothesis testing with the Benjamini–Hochberg (BH) method[62]. The FDR cutoff for this analysis step is denoted by $FDR_{mol}$ and controlled at $FDR_{mol} = 0.05$ as our default setting. Our method generalises to any binary tumour characterisation, e.g., the MSI status in FIRE-3. As a default setting, we test genetic features that were mutated in at least ten tumours ($n = 10$).

### Identifying mutual exclusivity
For the identification of mutually exclusive modules, we used the Mutex algorithm[39] (function GET-MUTATIONS-MODULES in Supplementary Data 1). It leverages a signalling network[63] collecting interactions from Pathway Common[64], SPIKE[65] and SignaLink[66] in order to scan for common downstream effects of combinations of somatic alterations $\mathbf{X}$. The default setting only uses somatic variants that were altered in at least ten tumours ($n = 10$).

### Genetic and subtype-specific biomarkers
OncoBird tests single somatic alterations and previously derived mutually exclusive somatic alterations for differential prognosis in each treatment arm separately (function GET-TREATMENT-SPECIFIC-BIOMARKERS in Supplementary Data 1). The patient outcome $Y(T = t, S = s_k)$ for the treatment arm $T = t$ in subtype $S = s_k$ with $k = 1, \ldots, q$ may be defined by survival data (OS or PFS) or a binary variable measuring the objective response rate (ORR). Depending on the type of outcome, this is modelled with either Cox proportional hazards regression models or logistic regression models expressed by their linear predictor function $f(\mathbf{x}, t)$. Using this classical approach for subgroup analysis, the treatment-specific regression models in subtypes take the form

$$f(\mathbf{x}, t) = \alpha_{0j} + \alpha_{1j} x_j + \sum_l C_l \qquad (1)$$

Cox proportional hazards regression models for survival endpoints were implemented with the 'coxph' function from the *survival* R package or logistic regression models for binary response variables were implemented using the 'glm' function. We test each $\mathbf{x} = x_1, \ldots, x_m$ first across all tumours, and subsequently in tumour subtypes $\{s_1, \ldots, s_q\}$, i.e., CMS or primary sidedness. $\alpha_{1j}$ is the coefficient estimating the contribution of candidate biomarker $j = 1, \ldots, m$ for patient outcomes in the context of each treatment arm $T = t$ in the subtype $S = s_k$. The predictors $C_1, \ldots, C_l$ include additional prognostic covariates and their coefficients.

The $p$-value $p_{\alpha_{1j}}$ derived by a Wald test from the coefficient $\alpha_{1j}$ is multiplicity-adjusted for each treatment arm $t$ and across all biomarkers $x_j$ with $j = 1, \ldots, m$ for either all patients or across subtypes $s_k$ with $k = 1, \ldots, q$ and yields adjusted $p$-values $\widetilde{p}_{\alpha_{1j}}$ using the Benjamini–Hochberg (BH) method[62]. The default false discovery rates (FDR) are controlled at $\text{FDR}_\alpha = 0.1$ for either treatment-specific component $\alpha_{1j}$.

The adjustable default setting of OncoBird is to only perform statistical tests if, for a given candidate biomarker $x_j$ and tumour subtype $s_k$, at least $n = 10$ samples were present in each mutant and wild-type population. Additionally, OncoBird only tested alterations for which its corresponding gene module had at least $n$ tumours redistributed compared to the single gene alteration.

## Predictive components of biomarkers

For the subsequent comparison of treatment arms, OncoBird tests for significant statistical interactions between treatment arms and genetic alterations in tumour subtypes (function GET-PREDICTIVE-BIOMARKERS in Supplementary Data 1). For that, we modelled the outcome $Y(S = s_k)$ in subtype $S = s_k$ with $k = 1, \ldots, q$ using regression models with interactions between $T$ and $X_j$ which take the form

$$f(\mathbf{x}, t) = \beta_{0j} + \beta_{1j}x_j + \beta_{2j}x_j t + \sum_l C_l, \tag{2}$$

where the coefficients $\beta_{1j}$ and $\beta_{2j}$ estimate the prognostic and predictive component of biomarker $x_j$ in subtype $s_k$, respectively. The $p$-value $p_{\beta_{2j}}$ derived with a Wald test from the coefficient $\beta_{2j}$ is multiplicity-adjusted across all $m$ biomarkers for either all patients or across subtypes $s_k$ with $k = 1, \ldots, q$ and yields BH adjusted $p$-values $\widetilde{p}_{\beta_{2j}}$. The default FDR is controlled at $\text{FDR}_\beta = 0.2$ for predictive components. The biomarker $X_j$ in subtype $s_k$ is a putatively predictive biomarker if $\widetilde{p}_{\alpha_{1j}} < \text{FDR}_\alpha$ for either $t$ and $\widetilde{p}_{\beta_{2j}} < \text{FDR}_\beta$.

Furthermore, OncoBird only performs statistical tests if for a given genetic alteration $X_j$ and tumour subtype $s_k$, at least $n = 10$ samples were present in each mutant and wild-type population for each treatment arm as default setting.

## Resampling for correction of conditional average treatment effects

Lastly, we estimate the conditional average treatment effect (CATE) for the found biomarkers (function GET-PREDICTIVE-BIOMARKERS in Supplementary Data 1). For each significant $X_j$ in $s_k$, there is one CATE estimate in each found subpopulation with a positive (mutant) biomarker $x_j = 1$ and negative (wild type) biomarker $x_j = 0$. In each population, we estimate the CATE by modelling the outcome $Y$ by

$$f(\mathbf{x}, t) = \gamma_0 + \gamma_1 t + \sum_l C_l, \tag{3}$$

where $\gamma_1$ estimates the (biased) CATE in terms of either hazard ratios or odds ratios dependent on outcome type in the subgroup defined by biomarker $x_j$ and subtype $s_k$. The population with the larger absolute estimate $\gamma_1$ is used to estimate the subgroups $A_{x_j, s_k}$.

For each found subgroup $A$, we assess the significance to the associated CATE estimate $\gamma_1$ and derive the $p$-value $p_{\gamma_1}$ using a Wald test. Furthermore, we perform a multiplicity-adjustment of $p_{\gamma_1}$ and derive honest estimates of the CATE.

The $p$-values are adjusted for multiplicity using a permutation-based approach that takes into account the entire subgroup search strategy[3]. For that, we permuted the treatment labels $U = 1000$ times to obtain null datasets without any differential treatment effects. Next, for each null dataset, we select significant subgroups $A^{(u)}$ for the same thresholds and record the treatment effect $p$-value of the best subgroup $p^{(u)}$ with $u = 1, \ldots, U$. The adjusted $p$-values are then given by

$$\widetilde{p}_{\gamma_1} = \frac{1}{U} \sum_{u=1}^{U} I_{\{p^{(u)} \leq p_{\gamma_1}\}}(p^{(u)}), \tag{4}$$

the fraction of $p$-values $p^{(u)}$ that are smaller or equal than $p_{\gamma_1}$ with the indicator function $I$. Furthermore, we derive an honest estimate of the treatment effect $\gamma_1$. Since subgroups $A$ are derived from the same data as the treatment effect estimates, the estimates from the resubstitution $\gamma_1(A_{x_j, s_k})$ will be biased. In order to derive a bias-corrected estimate $\widetilde{\gamma}_1$, we use a previously proposed non-parametric bootstrap approach[9]. For that, we generated $B = 500$ bootstrapped datasets. For each resampled dataset $b = 1, \ldots, B$ we estimate subgroups $\hat{A}_{x_j, s_k}^{(b)}$. The treatment effects can then be either estimated on the b-th resampled dataset $\gamma_1^{(b)}(A^{(b)})$ or on the original dataset $\gamma_1(A^{(b)})$. The bias-corrected CATE estimate is then given by

$$\hat{\gamma}_1 = \frac{1}{B} \sum_{b=1}^{B} \left( \gamma_1(A) + \gamma_1\left(A^{(b)}\right) - \gamma_1^{(b)}\left(A^{(b)}\right) \right). \tag{5}$$

The 95% confidence intervals are constructed by the 0.025 and 0.975 quantiles of the bootstrapped distribution.

## OncoBird parameterisation for FIRE-3

We used the function GET-MUTATIONS-IN-SUBTYPES to evaluate the primary tumour side and CMS as tumour subtypes with the default setting $\text{FDR}_{mol} < 0.05$. In total, we performed 156 and 312 statistical tests for the primary tumour sidedness and CMS, respectively. Using the GET-MUTATIONS-MODULES function with default settings, we analysed 42 genes which yielded 29 mutually exclusive modules. Mutations in *KRAS* or *NRAS* are the established clinical biomarkers for anti-EGFR treatment, thus we jointly modelled *KRAS* and *NRAS* as *RAS* mutations resulting in 10 additional modules.

The GET-TREATMENT-SPECIFIC-BIOMARKERS function was used with the number of metastatic sites and the information about a prior tumour resection as added covariates $C_1, C_2$. With the OncoBird default setting, we performed 816 statistical tests across all readouts $Y$ (OS, PFS and ORR), the cetuximab and bevacizumab treatment arm and tumour subtypes, i.e., CMS1-4, left- and right-sided and across all tumours. FDR cutoffs are employed for each treatment arm separately and are denoted $\text{FDR}_{cet}$ and $\text{FDR}_{bev}$ for the analysis in the cetuximab and bevacizumab treatment arms, respectively. In total, we found 92 significant associations with the default setting $\text{FDR}_{cet/bev} < 0.1$. The criteria $HR < 1$ and $OR < 1$ corresponded to a better prognosis for the mutant tumours compared to the wild-type tumours and vice versa. To consistently report $HR < 1$ and $OR < 1$ as beneficial risk reduction, reciprocal values of HRs and ORs were used if wild-type tumours displayed a better prognosis. We represent $p$-values, hazard/odds ratios with the 95% confidence intervals (CI) in square brackets and the associated FDRs.

In FIRE-3, the `GET-PREDICTIVE-BIOMARKERS` function with default settings resulted in a total amount of 396 statistical tests across the readouts $Y$ (OS, PFS and ORR) and the tumour subtypes $s_k$. FDR cutoffs for the interaction tests across both treatment arms are denoted by $FDR_{int}$. We explored 57 associations with $FDR_{int} < 0.6$ and $FDR_{cet/bev} < 0.1$ (Supplementary Data 3) and further focused on a subset of five biomarkers with default setting $FDR_{int} < 0.2$ for OS, i.e., two gene modules and three single genes (Supplementary Data 4). For the cross-validation analysis, a more lenient $FDR_{int} < 0.3$ was employed, which deviated from default setting to account for reduced sample sizes in the training and testing splits. HRs and ORs >1 and <1 corresponded to benefit with cetuximab and bevacizumab, respectively. To report the benefits of cetuximab treatment, the reciprocal values of HRs and ORs were used in the manuscript in order to display treatment benefits consistently with HR < 1 and OR < 1. We reported $p$-values and hazard/odds ratios with the 95% CIs for the treatment comparison and the $p$-values and associated FDRs for the interaction tests.

### OncoBird parameterisation for ADJUVANT

The ADJUVANT clinical trial in *EGFR* mutant non-small cell lung cancer (NSCLC) aimed to assess the efficacy of gefitinib versus chemotherapy with vinorelbine and cisplatin (NCT01405079)[34]. The trial was previously approved by the research ethics boards of Guangdong Provincial People's Hospital and all other participating hospitals[35]. Of note, 58% and 59% of patients had female sex in the gefitinib and chemotherapy arm, respectively. The sex was reported according to the study protocol[34], and gender cannot be distinguished retrospectively. We used the EGFR subtype, i.e., exon 19 deletion or exon 21 Leu858Arg, and the smoking history as putative tumour subtypes and clinical endpoints were disease-free survival (DFS) and overall survival (OS). We analysed 22 somatic alterations in 171 patients, from which 76 patients were treated with chemotherapy alone, and 95 were treated with gefitinib. For the subsequent analysis, we used the OncoBird default settings. The obtained results (Supplementary Data 2; Supplementary Figs. 3–6) and an associated extensive report can be reproduced in a runnable demo on Code Ocean (https://codeocean.com/capsule/9911222/tree/v1).

### Benchmarking of alternative methods with FIRE-3

For benchmarking the biomarker identification, we compared OncoBird to seven competing subgroup analysis algorithms leveraging the overall survival of FIRE-3 (Supplementary Table 1)[8,9,11,12,44–46]. We formed predictors by concatenating clinical annotations, including information about tumour resection, number of metastatic sites, age, gender, MSI and lung metastatic status. We added single genetic alterations and mutually exclusive modules observed across at least ten patients and in both investigated tumour subtypes, thus mirroring the OncoBird default settings. Furthermore, we investigated interactions between genetic alterations and tumour primary sidedness or CMS as predictors. Subgroups for the method evaluation were formed as the union of the subgroups showing cetuximab benefit according to the identified biomarkers (Supplementary Table 1).

All benchmarked models were 5-fold cross-validated with five repetitions. A univariate Cox proportional hazards model assessed performances leveraging the treatment effect based on OS in the subgroups with predicted benefits according to the found biomarkers. This included the treatment effect across the whole test set and in the subgroup defined by the current treatment guidelines, i.e., left-sided and *RAS* wild-type tumours[37]. The significance of the treatment effect in the subgroups of the test set was assessed using a modified $t$-test for resampled performance metrics[67], denoted by $p_{cv}$.

For comparing computational methods and their predicted biomarkers, the models were fitted on the whole dataset. The parameterisation of these methods was followed according to the suggested default settings unless in conflict with the above outlined

use case. For example, for tree-based methods, the features contained in the resulting tree were used as biomarkers with tree depths = 2, with a minimum subgroup size of $n = 10$. For the implementation of the virtual twins method (VT)[9], we used the R package *randomForestSRC* with default parameters and averaged predictions over 10 times repeated 10-fold cross-validation. Subsequently, a regression tree was fitted to the original data. In order to perform model-based recursive partitioning[8], we used the R package *model4you*[68] using an exponential model with default conditional inference tree control parameters. The PRISM method[46] was implemented in the R package *StratifiedMedicine*, for which we used Cox proportional hazards regression. We used the implementation of causal survival forests[69] (CRF) in the R package *grf*[70] for estimating conditional treatment effects. The propensity scores were set as constant and the target estimand was set to restricted mean survival time (RMST) with horizon = 100. After model fitting, variable importance scores were extracted, and biomarkers were selected according to predictors with significant linear projections of the conditional average treatment effects ($p < 0.05$). Next, we employed policy learning (POL)[44] to find optimal treatment regimens using the R package *policytree*[71]. We used the 50 most important predictors according to the CRF causal survival forest model variable importance scores and their treatment effect estimates to produce a decision tree.

The remaining methods were not based on trees. For the outcome weighted method (OWE)[11], implemented in the R package *personalized*[72], we used a constant propensity score, lasso loss and 10-fold cross-validation. The GUIDE method[45] was available as a binary executable under https://pages.cs.wisc.edu/~loh/guide.html. We used Cox proportional hazards regression with interactions tests and mean-based trees with pruning. For the SIDES method (R package *SIDES*)[7], we used level_control=0 and alpha=0.05.

### Statistics and reproducibility

The investigators were not blinded to the randomised treatment allocation during the data collection and outcome assessment. Since the conducted subgroup analysis is retrospective, the sample sizes were not predetermined. No data were excluded from the analysis. Details of the conducted statistical tests are provided in the figure captions, Supplementary Data 2–4 and Source Data. The results of the statistical analysis of the ADJUVANT clinical trial are reproducible from a demo run on Code Ocean (https://codeocean.com/capsule/9911222/tree/v1).

### Reporting summary

Further information on research design is available in the Nature Portfolio Reporting Summary linked to this article.

## Data availability

The clinical data summary from the FIRE-3 clinical trial analysed in this study has been deposited in the Pharmnet.bund online platform of the German Federal Ministry of Health (https://portal.dimdi.de/data/ctr/O-0329_01-2-1-B80630-20190731152224.pdf) and was published before[19]. The clinical and molecular data is available under restricted access due to data privacy laws. The raw and processed data can be obtained through the corresponding author at volker.heinemann@med.uni-muenchen.de. The data from the results of OncoBird v0.1.0 executed on the FIRE-3 trial are available in Supplementary Data 3 and Source Data. The processed data from the ADJUVANT clinical trial is available on Zenodo[33,35]. The data from the results of OncoBird v0.1.0 executed on the ADJUVANT trial are available in Supplementary Data 2, Source Data and on Code Ocean (https://codeocean.com/capsule/9911222/tree/v1). Source data are provided with this paper.

## Code availability

Oncology Biomarker Discovery (OncoBird) is publicly available at https://github.com/MendenLab/OncoBird. The repository contains an

R package as well as a Shiny application with a graphical user interface in a local docker container (Supplementary Fig. 1). Additionally, a demo run of OncoBird v0.1.0 used for analysis is available on Code Ocean (https://codeocean.com/capsule/9911222/tree/v1).

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

## Acknowledgements

This project has received funding from the European Research Council (ERC) under the European Union's Horizon 2020 research and innovation programme (grant agreement No. 950293, M.P.M.). The clinical study received industrial funding from Merck KGaA, Darmstadt, Germany and Pfizer GmbH, Germany. The transcriptome-based microarray for gene expression using Xcel® Array received funding from Almac Ltd, Belfast, UK. The FoundationOne® based sequencing analysis (MSI) received funding from Roche Pharma AG, Grenzach, Germany (grant numbers: n/a, V.H., S.S.).

## Author contributions

Conceptualisation, M.P.M. and V.H.; Data curation, A.S., S.S., D.P.M., U.V., T.D., M.M. and A.J.O.; Formal analysis, A.J.O.; Methodology, A.J.O., A.S. and M.P.M.; Supervision, V.H. and M.P.M.; Visualisation, A.J.O. and L.H.; Writing original draft, A.J.O., A.S., V.H. and M.P.M.; Writing, review and editing, all authors.

## Funding

## Competing interests

A.S. served on advisory boards for BMS and Novocure, received honoraria for talks by Roche, Servier and Taiho Pharmaceuticals and

received reimbursement for travel by Roche, Merck KGaA, MSD Sharp & Dohme, Pfizer, Lilly Oncology, and Amgen. V.H., S.S. and D.P.M. received honoraria for talks, advisory boards and travel expenses by Merck KGaA, Amgen, Roche, Pfizer, BMS, MSD, AstraZeneca, Novartis, Terumo, Oncosil, Nordic, Seagen, GSK, Takeda, Servier, Pierre Fabre, Taiho, Lilly Oncology, Servier, Sanofi and Bayer Pharmaceuticals. M.P.M. is a former employee at AstraZeneca, academically collaborates with AstraZeneca, GSK and Roche, and receives funding from GSK and Roche. J.W.H. served on an advisory board for Roche, has received honoraria from Roche, and travel support from Novartis. M.M. received honoraria for advisory boards or talks by Amgen, BMS, Roche, Merck KGaA, MSD Sharp & Dohme, Lilly Oncology, Servier, Pierre Fabre, Taiho Sanofi and Bayer Pharmaceuticals and serves as officer for the European Organisation on Research and Treatment of Cancer (EORTC), and Arbeitsgemeinschaft internistische Onkologie (AIO). C.B.W. has received honoraria from Amgen, Bayer, Chugai, Celgene, GSK, MSD, Merck, Janssen, Ipsen, Roche, Servier, SIRTeX, Taiho; served on advisory boards for Bayer, BMS, Celgene, Servier, Shire/Baxalta, Rafael Pharmaceuticals, RedHill, Roche, has received travel support by Bayer, Celgene, RedHill, Roche, Servier, Taiho and research grants (institutional) by Roche. C.B.W. serves as an officer for the European Society of Medical Oncology (ESMO), Deutsche Krebshilfe (DKH) and AIO. The remaining authors declare no competing interests.
