## [Peer Review File · Nature Communications]

REVIEWERS' COMMENTS

Reviewer #1, expert in colorectal cancer genomics (Remarks to the Author):

The manuscript describes a series of analyses of biomarkers, mostly based on mutation or gene expression data, of outcome in a specific clinical trial. There is an element of biomarker discovery. The statistical methods used include some techniques that have not been routinely applied to such data, although all the methods are established and all have been used or considered by a number of clinical trials for cancer in which molecular analysis of tumours has been performed. The "toolkit" - not an ideal name - is applied to a specific clinical trial of moderate size, resulting in some discovery of potential biomarkers, although there is no validation. I have no major concerns with the quality of the trial, molecular data or analytical methods, but the advances reported seem very limited.

Reviewer #2, expert in biomarker use for colorectal cancer (Remarks to the Author):

Ohnmacht et al have developed a toolkit for biomarker discovery in randomised clinical trials that takes into account treatment arm as well as additional subgroups within 1 treatment arm. They demonstrate the value of their toolkit on data from the FIRE-3 trial in metastatic colorectal cancer patients and their analysis reveals some interesting new insights. However, I have some questions/concerns for the authors as listed below:

1) My biggest concern is with the large number of sub-analyses in sometimes relatively small patient groups. Even though this is also acknowledged by the authors, not all their conclusions take this into account. For example on page 12 lines 327-332, I think the fact that CMS2 shows an effect independent of treatment arm whereas CMS4 only shows this in 1 treatment arm may very well be explained by the higher number of CMS2 versus CMS4. So I would urge the authors to only draw conclusions on the associations they do observe and not based on the absence of a significant observation.

2) I am also wondering how the used cut-offs for FDR were chosen. It appears random and differs for different parts of the toolkit (FDR<0.25 vs FDR<0.6). In general these FDRs are quite high and I am wondering what remains if a more generally accepted FDR of 0.1 or even 0.05 is used. In addition, are these thresholds fixed or do you need to choose them yourself when using this toolbox?

3) With regard to the CMS classification the authors need to clarify what algorithm was used exactly. Is this based on the random forest model, or the single sample predictor for example? This may influence the results as calls may differ for a proportion of the samples. I was surprised to see all samples actually got a call as in my experience a substantial proportion of samples usually cannot be assigned to a specific subgroup.

4) Why was MSI status not used for molecular subgrouping of the samples but treated as a clinical variable instead? It is known that CMS1 is enriched for MSI but not completely overlapping. I think it is important to take into account that prognosis differs between MSI and MSS metastatic colorectal cancers.

5) How useful is this toolkit for clinical trials in which a biomarker is used upfront to select which patients go into which treatment arm? As we are shifting towards this type of clinical trials it would be good if the authors comment on this in their manuscript.

5) Overall, I think the paper is quite methodological, which is not necessarily a bad thing. However I would like to see the application of the toolkit to a different dataset to be able to judge the general merit as is claimed by the authors. It seems very tailored towards the FIRE-trial and I think the paper would gain a lot by an additional demonstration on a completely different dataset.

6) Finally, could the authors specify what type of input data is actually used by their toolbox? Are

there any recommendations on how to score certain molecular alterations? Is it also possible to include for example chromosomal regions instead of copy numbers per gene?

Reviewer #3, expert in bioinformatics analysis (Remarks to the Author):

In this paper, the author developed a kind of Cancer Biomarker Discovery Toolkit (CBDT) to identify biomarkers based on single genes or mutually exclusive genetic alteration patterns in tumour subtypes. The authors have tried to identify the biomarkers with predictive components, e.g. mutual exclusivity of chr20q amplifications and ERK signaling mutations resulting in treatment benefits with cetuximab over bevacizumab. In summary, the results in this paper are prepared with a series of analysis, which is enough to show the capability of the CBDT method. Many data from clinical trial are also provided to verify the hypothesis from CBDT. From my point of view, the work is well-done and provides interesting results to the CBDT and thus it merits to be published. Just, I suggest some minor modifications before publication:

1. The manuscript is not so easy to read, since it contains a lot of information about both logical frameworks and biological terms. It is difficult for the reader to catch the essential elements in this work. The CBDT should be compared with other integrative analysis tools, in order to demonstrate its merits and capability. The data in Table S2 should be further explained, including the time efficiency, accuracy, cost and so on.
2. The scientific advances in this paper is not enough. The CBDT is a data-driven subgroup analyzing method. However, in order to fit the standard of Nat Common, the author should also provide some interesting findings about cancer treatment or biomaker development based on this CBDT.
3. How to identify and compare the sensitivity and the reaction time of different biomaker by using this CBDT? More information should also be considered in CBDT, including the effect biomarkers at molecular levels and the effects at the advanced biological level. Also, the repeatability and individual differences of the selected biomarker analysis need be in the acceptable range. The author should further confirm all these factors
4. How do you use this CBDT to evaluate new drugs or therapies? Can machine learning technology be applied to data analysis?
5. In figure 3, more information about using the combination of multiple biomarkers to predict the prognosis of non-progressive or control patients should be provided? The prediction accuracy and the comparison with other analysis tools should be provided.
6. It is quite difficult to understand the configuration of CBDT. There are too many small letters and curves in all the figures. Thus, the quality of all the figures in this paper should be further improved, where the letters and the sketch should be clear and informative. The author should modify Figures to guarantee the readers' smooth reading.

RESPONSE TO REVIEWERS' COMMENTS

**Reviewer #1, expert in colorectal cancer genomics (Remarks to the Author):**

**The manuscript describes a series of analyses of biomarkers, mostly based on mutation**
**or gene expression data, of outcome in a specific clinical trial. There is an element of**
**biomarker discovery. The statistical methods used include some techniques that have**
**not been routinely applied to such data, although all the methods are established and**
**all have been used or considered by a number of clinical trials for cancer in which**
**molecular analysis of tumours has been performed. The "toolkit" - not an ideal name -**
**is applied to a specific clinical trial of moderate size, resulting in some discovery of**
**potential biomarkers, although there is no validation. I have no major concerns with the**
**quality of the trial, molecular data or analytical methods, but the advances reported**
**seem very limited.**

We thank the reviewer for highlighting the quality of the trial, data and methods and the feedback.
We also feel that similar computational methods are considered by a number of clinical trials,
but lack standardisation, accessibility, interoperability, reusability and transparency for
biomarker identification. In order to address this, we have developed a statistical framework to
characterise 1) the molecular and 2) the biomarker landscape of clinical trials.

We disagree that the advances are limited and want to highlight the following key novelties of
our approach and findings:

- - An integrative and standardised analysis framework of the molecular and biomarker
landscape of a clinical trial in an interpretable and transparent manner, whilst sharing
the software findable, accessible, interoperable and reusable (FAIR)
- - Ensuring its utility and generalisability by providing benchmarks with other methods
and applying it to the ADJUVANT clinical trial
- - A validation framework based on a resampling-based correction of the conditional
average treatment effect
- - Proposed novel strategies for patient stratification in metastatic colorectal cancer
treated with cetuximab or bevacizumab

We revised the manuscript to emphasise its novelties to a greater extent.

For the molecular landscape of FIRE-3, we would like to draw the attention to **Fig. 2**, which
 we consider the most comprehensive molecular landscape characterisation of FIRE-3 so far:

**Figure 2: Molecular landscape of the FIRE-3 clinical trial.** (a) Oncoprint of 373 mCRC tumours including
 mutations and copy number alterations detected in more than 12 tumours. (b) The mutually exclusive alteration
 patterns were derived with the *Mutex* algorithm. Gene expression profiles of 451 mCRC tumours are annotated by
 (c) the consensus molecular subtypes (CMS) and (d) the primary tumour side. (e) Venn diagram showing all
 enriched somatic alterations for CMS1 and right-sided tumours, and (f) enriched somatic alterations for CMS2 and
 left-sided tumours. (g) Frequently altered cancer genes tested for enrichment in left- or right-sided tumours, and
 (h) tested against CMS subtypes.

 We regret that the term 'toolkit' did not resonate, and thus decided to rename our analysis
 framework to **Oncology Biomarker Discovery**, in short *OncoBird*. We wish to clarify that the
 stated advancements empower the characterisation of the molecular and biomarker landscape
 of any randomised clinical trial. To highlight the methodology novelties of *OncoBird*, we have
 majorly revised **Fig. 1**:

 **Figure 1: The Oncology Biomarker Discovery (OncoBird) workflow.** (a) Patients in clinical trials are
 characterised according to tumour somatic mutations and copy number alterations (M), tumour subtypes (S) and
 were treated with two treatment regimens (T). (b) With this input, *OncoBird* outlines (c) the molecular landscape
 and (d) the biomarker landscape. For the latter, (e) somatic alterations are explored for differential patient prognosis
 for each treatment arm. (f) Consecutively, for each treatment arm, subtype-specific biomarkers are derived. (g)
 Finally, interactions between treatment arms are examined. Here exemplified on the FIRE-3 clinical trial, (h) *RAS*
 mutations are established biomarkers of cetuximab resistance. (i) Patients with *RAS* wild type tumours showed a
 better prognosis when treated with cetuximab within left-sided tumours compared to right-sided tumours. In
 addition, (j) the *RAS* wild type subpopulation in left-sided tumours showed benefits when treated with cetuximab
 compared to bevacizumab.

 In addition, we have revised the **Abstract**, objective paragraph of the **Introduction** and
 **Results** section accordingly to reflect these changes:

**“Abstract**

Precision medicine has revolutionised cancer treatments, however, actionable
 biomarkers remain scarce. To address this, we developed the Oncology Biomarker
 Discovery (*OncoBird*) framework for analysing the molecular and biomarker landscape

of randomised controlled clinical trials. *OncoBird* identifies biomarkers based on single
genes or mutually exclusive genetic alterations in isolation or in context of tumour
subtypes, and finally, assesses the predictive component by their treatment
interactions. Here, we utilised the open-label, randomised phase III trial (FIRE-3, AIO
KRK 0306) in metastatic colorectal carcinoma patients, who received either cetuximab
or bevacizumab in combination with 5-fluorouracil, folinic acid and irinotecan
(FOLFIRI). We systematically identified five biomarkers with predictive components,
e.g. patients carrying chr20q amplifications and mutually exclusive ERK signalling
mutations benefited from cetuximab compared to bevacizumab. In summary, *OncoBird*
characterises the molecular landscape and outlines actionable biomarkers, which
generalises to any molecularly characterised randomised controlled trial.”

“Introduction

[...] The *OncoBird* workflow is divided into five distinct steps: it systematically 1)
investigates the molecular landscape of a clinical trial, i.e. copy number variants,
somatic mutations, mutually exclusive patterns and predefined tumour subtypes; 2)
identifies biomarkers within a treatment arm based on genetic alterations, and 3) in
relation to the predefined tumour subtypes; consecutively, 4) evaluates their predictive
component across treatment arms; and finally, 5) *OncoBird* comprehensively corrects
for multiple hypothesis testing and computationally validates biomarkers based on
resampling methods. This integrated analysis allows the molecular characterisation of
clinical trials and the identification of subtype-specific biomarkers with treatment
benefits in an interpretable and transparent manner and therefore operates
complementary to existing methods. The utility of *OncoBird* is exemplified by the
application to the FIRE-3 clinical trial, generalises to the ADJUVANT clinical trial, and
in fact, would generalise to any molecularly characterised randomised controlled trials
(RCT) in oncology.”

“Results

*OncoBird* is applicable to RCTs accompanied with molecular characteristics including
genetic sequencing panels which yield copy number variations and somatic driver
mutations (**Fig. 1a**). In addition, a second layer of stratification can be supplied in the
form of the predefined tumour subtypes (**Fig. 1a**). Then, *OncoBird* systematically
assesses the genetic landscape in the context of tumour subtypes (**Fig. 1b**) and
outlines the biomarker landscape across multiple clinical responses (**Fig. 1c**), i.e. time-
to-event data (overall or progression-free survival; **Methods**), and binary variables
capturing treatment success (objective response rate; **Methods**) [...]

[...] To reveal the biomarker landscape we employed the following stratification and
modelling strategies (**Methods, Algorithm 1**): We first investigated each alteration for
stratifying patients by their prognosis within each treatment arm (**Fig. 1d**).
Consecutively, we inspected alterations in tumour subtypes (**Fig. 1e**), revealing
subtype-specific biomarkers. Finally, we tested for treatment interactions to reveal
biomarkers with predictive effects (**Fig. 1f**). Importantly, subtypes and genetic
alterations ought to be independent from the treatment assignment. The molecular
landscape and individual treatment arm analysis could be applied to any trial design
without any limitations.”

We thank the reviewer for acknowledging the discovery of potential biomarkers in FIRE-3, and
fully agree that an external validation is required to conclusively confirm these biomarker
hypotheses, which may impact patient stratification in the near future. Currently, independent
follow-up clinical trials are carried out to empower a comprehensive validation, whilst this is
out-of-scope for this computational study. In the meantime, we expanded our computational
validation efforts. This is, we incorporated resampling-based corrections of the conditional
average treatment effects in subgroups for raising the evidence of benefits in the found
subgroups, which is described in the completely revised methods section:

“Resampling for correction of conditional average treatment effects

Lastly, we estimate the conditional average treatment effect (CATE) for the found
biomarkers (function `GET-PREDICTIVE-BIOMARKERS` in **Algorithm 1**). For each
significant X_j in s_k , there is one CATE estimate in each found subpopulation with a
positive (‘mutant’) biomarker $x_j = 1$ and negative (‘wild type’) biomarker $x_j = 0$. In
each population, we estimate the CATE by modelling the outcome Y by

$$\mathbb{E}(Y(S = s_k, X_j = x_j) | \mathbf{X} = \mathbf{x}, T = t) = f(\gamma_0 + \gamma_1 t + \sum_l C_l),$$

where γ_1 estimates the (biased) CATE in terms of either hazard ratios or odds ratios
dependent on outcome type in the subgroup defined by biomarker x_j and subtype s_k .
The population with the larger absolute estimate γ_1 is used to estimate the subgroups
A_{x_j, s_k} .

For each found subgroup A , we assess the significance to the associated CATE
estimate γ_1 and derive the p-value p_{γ_1} . Furthermore, we perform a multiplicity-
adjustment of p_{γ_1} and derive honest estimates of the CATE.

The p-values are adjusted for multiplicity using a permutation-based approach that
takes into account the entire subgroup search strategy³. For that, we permuted the

treatment labels $U = 1000$ times to obtain null datasets without any differential
 treatment effects. Next, for each null dataset, we select significant subgroups $A^{(u)}$ for
 the same thresholds and record the treatment effect p-value of the best subgroup $p^{(u)}$
 with $u = 1, \dots, U$. The adjusted p-values are then given by

$$\tilde{p}_{\gamma_1} = \frac{1}{U} \sum_{u=1}^U I_{\{p^{(u)} \leq p_{\gamma_1}\}}(p^{(u)}),$$

the fraction of p-values $p^{(u)}$ that are smaller or equal than p_{γ_1} with the indicator function
 I . Furthermore, we derive an honest estimate of the treatment effect $\hat{\gamma}_1$. Since
 subgroups A are derived from the same data as the treatment effect estimates, the
 treatment estimates from resubstitution $\gamma_1(A_{x_j, s_k})$ will be biased. In order to derive a
 bias-corrected estimate $\tilde{\gamma}_1$, we use a previously proposed non-parametric bootstrap
 approach⁹. For that, we generated $B = 500$ bootstrapped datasets. For each resampled
 dataset $b = 1, \dots, B$ we estimate subgroups $\hat{A}_{x_j, s_k}^{(b)}$. The treatment effects can then
 be either estimated on the b-th resampled dataset $\gamma_1^{(b)}(A^{(b)})$ or on the original dataset
 $\gamma_1(A^{(b)})$. The bias-corrected CATE estimate is then given by

$$\tilde{\gamma}_1 = \frac{1}{B} \sum_{b=1}^B \left(\gamma_1(A) + \gamma_1(A^{(b)}) - \gamma_1^{(b)}(A^{(b)}) \right).$$

The 95% confidence intervals are constructed by the 0.025 and 0.975 quantiles of the
 bootstrapped distribution.”

 Furthermore, we have applied *OncoBird* to additional and publicly available data (i.e. the
 ADJUVANT trial) to further demonstrate the generalisability of our computational framework:

“Whilst here we focused on the FIRE-3 trial in colorectal cancer, we show the
 generalisability of *OncoBird* by applying it to the ADJUVANT clinical trial, which
 explores gefitinib in non-small cell lung cancer (NSCLC; **Supplementary Note 1**)^{35–37}.
 The ADJUVANT study reported predictive components of five alterations, i.e. *TP53*
 mutations, *RB1* alterations and copy number amplifications of *NKX2-1*, *CDK4* and *MYC*
 164³⁷. Four out of five biomarkers were concordantly identified for disease-free survival
 with *OncoBird* and are presented in an extensive report ($\text{FDR}_{\text{int}} < 0.2$; **Supplementary**
 **Note 1**). In addition, *OncoBird* suggests that the mutual exclusivity patterns play a role
 in the biomarker landscape of NSCLC (**Supplementary Note 1**). In more detail, we
 observed gefitinib benefits in tumours that were characterised by mutations in either
 *TP53*, *SMAD4* or *CDK4* amplifications ($p = 0.0002$, HR = 0.37 [0.21-0.63]), for which
 the resampling-based validation of the conditional average treatment effect yielded p_{adj}

= 0.001 with HR = 0.32 [0.14-0.86] (**Supplementary Note 1**). These findings highlight
 the accessibility, reproducibility and interoperability of *OncoBird*.”

The reusability is further improved by FAIR (findable, accessible, interoperable, reusable)
 sharing the full source code and comprehensive documentation via codeocean and github.

“Here, we present the **Onco**logy **Bi**omarker **D**iscovery (*OncoBird*) framework which
 empowers the systematic identification of actionable biomarkers for clinical trials in
 oncology. *OncoBird* is publicly available as a software package at
 <https://github.com/aljoshoh/OncoBird> and a demo run is available at
 <https://codeocean.com/capsule/3676298/tree>. Furthermore, users can run a graphical
 user interface within a docker container (**Fig. S1**).”

Notably, we put great effort into making *OncoBird* accessible for clinicians by providing a
 Graphical User Interface (GUI) based on a shiny application, which produces a report on the
 molecular and biomarker landscape of any given clinical trial. For this, we have added
 **Supplementary Fig. 1** for highlighting screenshots of *OncoBird*'s graphical user interface:

a

b Uploading Files

c Genomic Enrichments

d Mutual Exclusivity

e

f

g

h Summary

gene	mut_count	phr_OS	phr_PFS	HR_OS	HR_PFS	hazard_OS_1	hazard_OS_2	hazard_PFS_1	hazard_PFS_2	mean_OS_1	mean_OS_2	mean_PFS_1	mean_PFS_2	mi
1. SMYD3	2	0.011	0.022	0.28	0.97	0.21(0.0188)	1.56(0.7175)	0.74(0.3125)	0.97(0.4121)	29.21	29.04	11.17	10.98	0.18
2. SLC16A7	4	0.002	0.004	0.23	0.86	1.03(0.0126)	0.53(0.0161)	1.54(0.0116)	0.74(0.0478)	28.89	29.75	8.8	9.62	0.14
3. SLC16A7	4	0.001	0.001	0.18	0.51	1.02(0.0162)	0.27(0.0016)	1.86(0.0039)	1.02(0.0017)	38.39	32.12	16.71	8.26	0.14
4. MUC6	4	0.006	0.007	0.18	0.51	1.02(0.0162)	1.04(0.0114)	1.02(0.0117)	0.74(0.0171)	37.9	32.12	16.71	8.26	0.14
5. SLC16A7	4	0.001	0.002	0.23	0.86	1.02(0.0126)	0.53(0.0161)	1.54(0.0116)	0.74(0.0478)	28.89	29.75	8.8	9.62	0.14
6. TOP2A	2	0.02	0.029	0.28	0.96	0.39(0.0088)	1.19(0.7172)	0.8(0.0116)	1.02(0.113)	32.16	27.4	12.16	10.25	0.14

**Figure S1: OncoBird dockerised Shiny application.** (a) Each tab corresponds to one analysis step, for example
(b) the user interface for data input. Then, results generated for (c) enrichment of genetic alterations in tumour
subtypes, (d) mutual exclusivity, treatment-specific biomarkers in (e) all tumours and (f) in tumour subtypes. Also,
results for predictive biomarkers using (g) interaction tests and (h) the table summary.

In essence, we strongly believe that *OncoBird* will become a highly used computational
framework to analyse the molecular and biomarker landscape of historical and recent
randomised clinical trials, which is here exemplified with the FIRE-3 and ADJUVANT clinical
trial.

**Reviewer #2, expert in biomarker use for colorectal cancer (Remarks to the Author):**

**Ohnmacht et al have developed a toolkit for biomarker discovery in randomised clinical**
**trials that takes into account treatment arm as well as additional subgroups within 1**
**treatment arm. They demonstrate the value of their toolkit on data from the FIRE-3 trial**
**in metastatic colorectal cancer patients and their analysis reveals some interesting new**
**insights.**

We want to thank the reviewer for the kind evaluation and highlighting the novelty of our
findings.

**However, I have some questions/concerns for the authors as listed below:**

**1) My biggest concern is with the large number of sub-analyses in sometimes relatively**
**small patient groups. Even though this is also acknowledged by the authors, not all**
**their conclusions take this into account.**

We agree with the reviewer that progressively smaller sample sizes in the analysis workflow
is the major limitation. For that, we incorporate frequency filters, i.e statistical tests are only
conducted if a minimum of ten patient tumours are within each group. This default threshold
can be adjusted by the user. In addition, we majorly expanded our computational efforts by
employing a resampling-based correction of the conditional average treatment effect to raise
evidence of potential treatment benefits in subgroups:

**“Resampling for correction of conditional average treatment effects**

Lastly, we estimate the conditional average treatment effect (CATE) for the found
biomarkers (function `GET-PREDICTIVE-BIOMARKERS` in **Algorithm 1**). For each

significant X_j in s_k , there is one CATE estimate in each found subpopulation with a
 positive ('mutant') biomarker $x_j = 1$ and negative ('wild type') biomarker $x_j = 0$. In
 each population, we estimate the CATE by modelling the outcome Y by

$$\mathbb{E}(Y(S = s_k, X_j = x_j) | \mathbf{X} = \mathbf{x}, T = t) = f(\gamma_0 + \gamma_1 t + \sum_l C_l),$$

where γ_1 estimates the (biased) CATE in terms of either hazard ratios or odds ratios
 dependent on outcome type in the subgroup defined by biomarker x_j and subtype s_k .
 The population with the larger absolute estimate γ_1 is used to estimate the subgroups
 A_{x_j, s_k} .

For each found subgroup A , we assess the significance to the associated CATE
 estimate γ_1 and derive the p-value p_{γ_1} . Furthermore, we perform a multiplicity-
 adjustment of p_{γ_1} and derive honest estimates of the CATE.

The p-values are adjusted for multiplicity using a permutation-based approach that
 takes into account the entire subgroup search strategy³. For that, we permuted the
 treatment labels $U = 1000$ times to obtain null datasets without any differential
 treatment effects. Next, for each null dataset, we select significant subgroups $A^{(u)}$ for
 the same thresholds and record the treatment effect p-value of the best subgroup $p^{(u)}$
 with $u = 1, \dots, U$. The adjusted p-values are then given by

$$\tilde{p}_{\gamma_1} = \frac{1}{U} \sum_{u=1}^U I_{\{p^{(u)} \leq p_{\gamma_1}\}}(p^{(u)}),$$

the fraction of p-values $p^{(u)}$ that are smaller or equal than p_{γ_1} with the indicator function
 I . Furthermore, we derive an honest estimate of the treatment effect γ_1 . Since
 subgroups A are derived from the same data as the treatment effect estimates, the
 treatment estimates from resubstitution $\gamma_1(A_{x_j, s_k})$ will be biased. In order to derive a
 bias-corrected estimate $\tilde{\gamma}_1$, we use a previously proposed non-parametric bootstrap
 approach⁹. For that, we generated $B = 500$ bootstrapped datasets. For each
 resampled dataset $b = 1, \dots, B$ we estimate subgroups $\hat{A}_{x_j, s_k}^{(b)}$. The treatment effects
 can then be either estimated on the b-th resampled dataset $\gamma_1^{(b)}(\hat{A}^{(b)})$ or on the original
 dataset $\gamma_1(A^{(b)})$. The bias-corrected CATE estimate is then given by

$$\tilde{\gamma}_1 = \frac{1}{B} \sum_{b=1}^B \left(\gamma_1(A) + \gamma_1(A^{(b)}) - \gamma_1^{(b)}(\hat{A}^{(b)}) \right).$$

The 95% confidence intervals are constructed by the 0.025 and 0.975 quantiles of the
 bootstrapped distribution."

This resampling analysis increases robustness and the evidence of reported interaction
biomarkers. Accordingly, we revised the **Results** section ‘Predictive components of
biomarkers’ that reports the adjusted treatment effect p-values and confidence intervals:

“For example, we found predictive value of chr20q amplifications in CMS2 tumours
treated with FOLFIRI plus cetuximab (**Fig. 5a,b**), which is evident by the significant
interactions of *TOP1* ($p_{\text{int}} = 0.07$, $\text{FDR}_{\text{int}} < 0.2$) and *ARFRP1* ($p_{\text{int}} = 0.01$, $\text{FDR}_{\text{int}} < 0.2$).
*ARFRP1* amplifications showed the largest predictive component among the chr20q
amplifications. Accordingly, we observed longer OS in the cetuximab treatment arm
compared to bevacizumab in CMS2 (**Fig. 5a,c**; *ARFRP1*: $p = 0.003$, $\text{HR} = 0.21$ [0.07-
0.59], $\text{FDR}_{\text{int}} < 0.2$; **Table S1**). The resampling-based adjusted treatment effect
confirmed this observation and yielded a hazard ratio in this subgroup of $\text{HR} = 0.21$
[0.09-0.54] with $p_{\text{adj}} = 0.04$ (**Fig. 5a,c**).”

And:

“[...] To highlight another example of interactions, tumours with *KRAS* mutations showed
CMS-specific responses. In CMS4, patients with *KRAS* wild type tumours responded
better to cetuximab compared to patients treated with bevacizumab (**Fig. 5b,d**; *KRAS*
wild types: $p = 0.02$, $\text{HR} = 0.57$ [0.35-0.93]; $p_{\text{int}} = 0.02$, $\text{FDR}_{\text{int}} < 0.2$), for which the
resampling-based adjusted treatment effect yielded $\text{HR} = 0.70$ [0.25-2.35] with $p_{\text{adj}} = 0.12$
(**Fig. 5b,d**). Our results suggest a predictive role of *KRAS* mutations in CMS4 for
cetuximab, which we also identified for PFS (**Fig. S10c,d**).”

**For example on page 12 lines 327-332, I think the fact that CMS2 shows an effect**
**independent of treatment arm whereas CMS4 only shows this in 1 treatment arm may**
**very well be explained by the higher number of CMS2 versus CMS4. So I would urge the**
**authors to only draw conclusions on the associations they do observe and not based**
**on the absence of a significant observation.**

We thank the reviewer for raising this invaluable comment about the absence of a significant
effect. We adjusted the paragraph and **Figure 5** to discuss only significant associations or
trends. The highlighted example of the reviewer has been adjusted accordingly:

“To highlight another example of interactions, tumours with *KRAS* mutations showed
CMS-specific responses. In CMS4, patients with *KRAS* wild type tumours responded

better to cetuximab compared to patients treated with bevacizumab (Fig. 5b,d; KRAS
 wild types: $p = 0.02$, HR = 0.57 [0.35-0.93]; $p_{\text{int}} = 0.02$, $\text{FDR}_{\text{int}} < 0.2$), [...]”

 **Figure 5: Predictive biomarkers in the context of tumour subtypes.** Overview of interaction biomarkers ($\text{FDR}_{\text{int}} < 0.2$) focusing on (a) mutant and (b) wild type tumours when comparing cetuximab and bevacizumab treatment. The models were fitted on OS, and error bars correspond to 95% CIs. Triangle points and confidence intervals were obtained from bootstrap-based bias-correction of treatment effects. Here exemplified, Kaplan-Meier curves compare treatments in subgroups for (c) *ARFRP1* amplifications in CMS2 and (d,e) *KRAS* mutations in CMS4.

In addition, we also revised the Results section ‘Subtype-specific biomarkers of cetuximab and bevacizumab’ with Figure 4 accordingly:

 “In total, we found 38 subtype-specific biomarkers of cetuximab ($\text{FDR}_{\text{cet}} < 0.1$;
 **Methods**). In particular, we recovered favourable OS of CMS2 patients treated with
 cetuximab (Fig. 4a), if their tumours additionally carried chr20q amplifications, i.e.
 *ARFRP1* (Fig. 4c; $p = 0.01$, HR = 0.32 [0.13-0.77], $\text{FDR}_{\text{cet}} < 0.1$), *TOP1* (Fig. S8c; p
 = 0.01, HR = 0.34 [0.15-0.74], $\text{FDR}_{\text{cet}} < 0.1$) and *SRC* (Fig. S8d; $p = 0.01$, HR = 0.37
 [0.17-0.78], $\text{FDR}_{\text{cet}} < 0.1$). Additionally, CMS4 *KRAS* mutant tumours treated with
 cetuximab showed worse OS (Fig. 4d; $p = 0.002$, HR = 2.60 [1.44-4.70], $\text{FDR}_{\text{cet}} < 0.1$)
 and PFS (Fig. S8a).

For reporting bevacizumab biomarker trends, we employed a lenient false discovery
 rate ($\text{FDR}_{\text{bev}} < 0.3$), which deviates from the conservative *OncoBird* default setting
 (**Methods**). Tumours with *KRAS* mutations classified as CMS2 tended to show worse

OS when treated with bevacizumab (Fig. 4e; $p = 0.004$, $HR = 2.33$ [1.31-4.15], $FDR_{bev} < 0.3$). In contrast, *KRAS* mutated tumours classified as CMS1 tended to show a longer
 OS compared to wild type tumours when treated with bevacizumab (Fig. 4f; $p = 0.03$,
 $HR = 0.33$ [0.12-0.93], $FDR_{bev} < 0.3$).”

**Figure 4: Identification of subtype-specific genetic biomarkers for FOLFIRI plus cetuximab or bevacizumab.**
 Subtype-specific genetic biomarkers for OS of (a) cetuximab and (b) bevacizumab using single Cox regression
 models including 95% confidence intervals (CI). Subtypes are defined by either the primary tumour side, CMS or
 unstratified (reference model). Kaplan-Meier plots of subtype-specific genetic biomarkers for (c) *ARFRP1* in CMS2,
 (d) *KRAS* in CMS4, (e) *KRAS* in CMS2 and (f) *KRAS* in CMS1 in either the cetuximab or bevacizumab treatment
 arm.

Furthermore, we cross-checked this across the whole manuscript, and are happy to confirm
 that this issue has been comprehensively addressed across the whole manuscript now.

 **2) I am also wondering how the used cut-offs for FDR were chosen. It appears random**
 **and differs for different parts of the toolkit (FDR<0.25 vs FDR<0.6). In general these**
 **FDRs are quite high and I am wondering what remains if a more generally accepted FDR**
 **of 0.1 or even 0.05 is used.**

We apologise for these inconsistencies and agree that our reported thresholds were too
 lenient. Therefore, we conservatively revised the multiplicity corrections in the form of false

discovery rates (FDR) and employed more conservative default cut-offs. We also recognised
that there is a high chance for confusion of different FDR thresholds, since the tool
incorporates three types of analyses and adjustments, i.e. molecular landscape analysis
(FDR_{mol}), treatment-specific ($FDR_{cet/bev}$) and interaction analysis (FDR_{int}). This annotation is
now consistently used across the whole manuscript to avoid confusion.

As suggested, for the enrichment test of the molecular landscape, we use a conservative
$FDR_{mol} < 0.05$, and all presented results remain conserved. For the identification of treatment-
and subtype-specific genetic biomarkers of FOLFIRI plus cetuximab or bevacizumab, we now
also apply a stringent $FDR_{cet/bev} < 0.1$. We made only one exception in the **Results** sections
‘Genetic biomarkers of bevacizumab’ and ‘Subtype-specific biomarkers of [...] bevacizumab’,
for which we used a more lenient threshold for exploration of trends, which is clearly flagged
to the reader:

“For exploring bevacizumab biomarker trends, we employed a lenient threshold of
$FDR_{bev} < 0.3$.”

Moreover, for the interaction biomarkers, we now consistently and more conservatively employ
a default $FDR_{int} < 0.2$ instead of 0.6. Whilst agreeing that our previous cut-off $FDR_{int} < 0.6$ was
too optimistically chosen, we want to highlight that statistical interactions commonly use more
lenient thresholds. This is, interaction tests subset the data, thereby reducing sample sizes,
and ultimately, decreasing statistical power, which we also discuss as limitation. Nevertheless,
we believe that these reported thresholds are reasonable now.

For being inclusive of putative biomarker trends, more lenient thresholds for the interaction
biomarkers are reported in the supplementary tables (**Table S1**). Notably, in our previous
manuscript, we reported 65 interaction biomarkers ($FDR_{int} < 0.6$) and 16 biomarkers ($FDR_{int} <$
0.25), whereas now we focus on five high confidence interaction biomarkers ($FDR_{int} < 0.2$).
Importantly, all highlighted and discussed results of our previous manuscript were amongst
these five high-confidence examples. The manuscript has been adjusted accordingly:

“In total, we found five putative interactions (**Methods; Table S2**; $FDR_{int} < 0.2$). For
reporting further biomarker trends, we also included summary statistics of 57
subgroups with a lenient threshold of $FDR_{int} < 0.6$ (**Table S1**).”

According to the new stringent thresholds of $FDR_{mol} < 0.05$, we adjusted **Fig. 2g,h**:

Figure 2: Molecular landscape of the FIRE-3 clinical trial. [...] (g) Frequently altered cancer genes tested for enrichment in left- or right-sided tumours, and (h) tested against CMS subtypes.

And revised **Fig. 4a,b** with an $FDR_{cet/bev} < 0.1$:

Figure 4: Identification of subtype-specific genetic biomarkers for FOLFIRI plus cetuximab or bevacizumab. Subtype-specific genetic biomarkers for OS of (a) cetuximab and (b) bevacizumab using single Cox regression models including 95% confidence intervals (CI). [...]

And finally adjusted **Fig. 5a,b** with an threshold of $FDR_{int} < 0.2$:

**Figure 5: Predictive biomarkers in the context of tumour subtypes.** Overview of interaction biomarkers (FDR_{int}
 < 0.2) focusing on (a) mutant and (b) wild type tumours when comparing cetuximab and bevacizumab treatment.
 The models were fitted on OS, and error bars correspond to 95% CIs. Triangle points and confidence intervals
 were obtained from bootstrap-based bias-correction of treatment effects. [...]

 **In addition, are these thresholds fixed or do you need to choose them yourself when**
 **using this toolbox?**

 As discussed above, we suggest default FDR cut-offs for all three types of analysis. In addition,
 the user can flexibly set all FDR cut-offs in order to apply more lenient or conservative FDRs.
 Both, the R package and graphical user interface (GUI) enable a full parameterization of
 *OncoBird*, as shown in **Supplementary Fig. 1:**

a

b Uploading Files

c Genomic Enrichments

d Mutual Exclusivity

e

f

g

h Summary

gene	mut_variables	mut_OS	mut_PFS	mut_34_OS	mut_34_PFS	mut_OS_1	mut_OS_2	mut_PFS_1	mut_PFS_2	mut_OS_1	mut_OS_2	mut_PFS_1	mut_PFS_2	mut_OS_1	mut_OS_2	mut_PFS_1	mut_PFS_2	mut_OS_1	mut_OS_2	mut_PFS_1	mut_PFS_2		
1. KRAS	2	0.01	0.022	0.28	0.93	0.21(0.072,0.58)	1.1(0.71,1.72)	0.7(0.33,1.46)	0.9(0.4,1.72)	21.21	21.04	11.17	10.98	21.71	21.6	11.17	10.98	21.71	21.6	11.17	10.98	21.71	21.6
2. EGFR	4	0.002	0.016	0.29	0.96	1.4(0.62,3.1)	0.4(0.2,0.75)	1.4(0.6,3.1)	0.7(0.3,1.7)	20.99	21.75	0.8	0.92	22.26	21.9	0.8	0.92	22.26	21.9	0.8	0.92	22.26	21.9
3. BRAF	4	0.001	0.001	0.29	0.93	1.0(0.5,1.9)	0.6(0.3,1.1)	1.0(0.5,1.9)	0.6(0.3,1.1)	20.83	21.76	10.71	0.92	21.21	21.6	10.71	0.92	21.21	21.6	10.71	0.92	21.21	21.6
4. PIK3CA	4	0.005	0.017	0.28	0.93	1.0(0.5,1.9)	0.6(0.3,1.1)	1.0(0.5,1.9)	0.6(0.3,1.1)	20.79	21.72	10.71	0.92	21.21	21.6	10.71	0.92	21.21	21.6	10.71	0.92	21.21	21.6
5. BRAF	4	0.001	0.002	0.29	0.94	1.0(0.5,1.9)	0.6(0.3,1.1)	1.0(0.5,1.9)	0.6(0.3,1.1)	20.8	21.73	10.71	0.92	21.21	21.6	10.71	0.92	21.21	21.6	10.71	0.92	21.21	21.6
6. KRAS	2	0.002	0.0279	0.28	0.94	0.9(0.4,1.8)	1.0(0.5,1.9)	0.9(0.4,1.8)	1.0(0.5,1.9)	20.78	21.74	10.71	0.92	21.21	21.6	10.71	0.92	21.21	21.6	10.71	0.92	21.21	21.6

 **Figure S1: OncoBird dockerised Shiny application.** (a) Each tab corresponds to one analysis step, for example
 (b) the user interface for data input. Then, results generated for (c) enrichment of genetic alterations in tumour
 subtypes, (d) mutual exclusivity, treatment-specific biomarkers in (e) all tumours and (f) in tumour subtypes. Also,
 results for predictive biomarkers using (g) interaction tests and (h) the table summary.

**3) With regard to the CMS classification the authors need to clarify what algorithm was**
**used exactly. Is this based on the random forest model, or the single sample predictor**
**for example? This may influence the results as calls may differ for a proportion of the**
**samples. I was surprised to see all samples actually got a call as in my experience a**
**substantial proportion of samples usually cannot be assigned to a specific subgroup.**

We thank the reviewer for pointing this out. This is correct, and tumours were classified
according to their closest CMS. Furthermore, 24 samples were unclassifiable because of
missing transcriptomics data, therefore, were omitted for the subtype-specific analysis. We
have revised the method section accordingly to supply this missing detail:

“In addition, annotations for molecular subtypes of mCRC were obtained from
transcriptome data that has been previously used to classify patients into their closest
consensus molecular subtype (CMS)^{21,29}, which used the *cmsclassifier* package with
the SSP predictor. Thereby, for 24 of out 373 patient tumours were not allocated to any
CMS because of missing transcriptomics data and were left out of the CMS-specific
analysis. The CMS classification was used as a complementary alternative to the primary
tumour side and is currently discussed in multiple clinical settings⁶¹.”

**4) Why was MSI status not used for molecular subgrouping of the samples but treated**
**as a clinical variable instead? It is known that CMS1 is enriched for MSI but not**
**completely overlapping. I think it is important to take into account that prognosis differs**
**between MSI and MSS metastatic colorectal cancers.**

We fully agree with the reviewer that MSI status is an important predictor for prognosis in
colorectal cancer. However, only 10 tumours were classified as MSI-H tumours in our dataset.
From those, only 6 and 4 were treated with bevacizumab and cetuximab, respectively, which
is in line with our statistical inclusion criteria for the analysis of the molecular landscape, but
did not meet the criteria for the biomarker analysis. We now acknowledge this in the manuscript
as the following:

“MSI is an important prognostic predictor and enriched in CMS1⁶³, which is observed in
our study with 8 of 10 MSI-H tumours being classified as CMS1. However, MSI-H
tumours are less prevalent in metastatic disease (~5%)⁶³. Furthermore, only 6 and 4
MSI-H tumours were treated with bevacizumab and cetuximab, respectively. [...]

[...] Our method generalises to any binary tumour characterisation, e.g. the MSI status
in FIRE-3. [...]

[...] Furthermore *OncoBird* only performs statistical tests if for a given genetic alteration
x and tumour subtype s_k , at least $n = 10$ samples were present in each mutant and wild
type population for each treatment arm as default setting.”

**5) How useful is this toolkit for clinical trials in which a biomarker is used upfront to**
**select which patients go into which treatment arm? As we are shifting towards this type**
**of clinical trials it would be good if the authors comment on this in their manuscript.**

Our analysis framework (i.e. re-named *OncoBird* upon suggestion of Reviewer #1) relies on a
randomised and controlled design for the clinical trials. The necessary condition to run our
framework is that the patient cohort chosen with a biomarker is independent of the treatment
assignment and therefore does not violate unconfoundedness. In a trial design in which the
intend-to-treat population according to the biomarker as inclusion criteria is defined before the
treatment randomisation, this assumption remains valid. Notably, the molecular landscape
analysis and individual treatment arm analysis could be applied without any limitations. Thus,
we have revised the **Results** section accordingly:

“Importantly, subtypes and genetic alterations ought to be independent from the
treatment assignment. The molecular landscape and individual treatment arm analysis
could be applied to any trial design without any limitations.”

Furthermore, we expanded the **Discussion** section for highlighting the generalisability of
*OncoBird* to different clinical trial designs:

“Notably, *OncoBird* was developed for RCT designs, and is generalisable to any trial
designs for which the intention-to-treat population was defined before the treatment
randomisation, i.e. the treatment assignment is independent from patient
characteristics. According to this, *OncoBird* is applicable to modern clinical trial designs
based on master protocols⁵⁸, i.e. basket, umbrella, and platform trials if control arms
are included.”

**5) Overall, I think the paper is quite methodological, which is not necessarily a bad**
**thing. However I would like to see the application of the toolkit to a different dataset to**
**be able to judge the general merit as is claimed by the authors. It seems very tailored**

**towards the FIRE-trial and I think the paper would gain a lot by an additional**
**demonstration on a completely different dataset.**

We fully agree with the reviewer that we present a novel method, i.e. *OncoBird*, and exemplify
its utility with the FIRE-3 clinical trial revealing novel biomarker insights. For highlighting the
methodology character of our study and the generalisability of *OncoBird*, the demo run on
codeocean applies *OncoBird* on the publicly accessible ADJUVANT clinical trial. This trial
focuses on gefitinib in non-small cell lung cancer. We added the full analysis report of *OncoBird*
to the supplements (**Supplementary Note 1**), and in addition, highlighted the concordant
biomarkers identified by *OncoBird*:

“Whilst here we focused on the FIRE-3 trial in colorectal cancer, we show the
generalisability of *OncoBird* by applying it to the ADJUVANT clinical trial, which
explores gefitinib in non-small cell lung cancer (NSCLC; **Supplementary Note 1**)^{35–37}.
The ADJUVANT study reported predictive components of five alterations, i.e. *TP53*
mutations, *RB1* alterations and copy number amplifications of *NKX2-1*, *CDK4* and *MYC*
484³⁷. Four out of five biomarkers were concordantly identified for disease-free survival
with *OncoBird* and are presented in an extensive report ($FDR_{int} < 0.2$; **Supplementary**
**Note 1**). In addition, *OncoBird* suggests that the mutual exclusivity patterns play a role
in the biomarker landscape of NSCLC (**Supplementary Note 1**). In more detail, we
observed gefitinib benefits in tumours that were characterised by mutations in either
*TP53*, *SMAD4* or *CDK4* amplifications ($p = 0.0002$, $HR = 0.37 [0.21-0.63]$), for which
the resampling-based validation of the conditional average treatment effect yielded p_{adj}
$= 0.001$ with $HR = 0.32 [0.14-0.86]$ (**Supplementary Note 1**). These findings highlight
the accessibility, reproducibility and interoperability of *OncoBird*.”

**6) Finally, could the authors specify what type of input data is actually used by their**
**toolbox?**

Thanks for raising this concern, and correspondingly, we have revised **Figure 1** to clarify the
input and workflow of *OncoBird*:

 **Figure 1: The Oncology Biomarker Discovery (OncoBird) workflow.** (a) Patients in clinical trials are
 characterised according to tumour somatic mutations and copy number alterations (M), tumour subtypes (S) and
 were treated with two treatment regimens (T). (b) With this input, *OncoBird* outlines (c) the molecular landscape
 and (d) the biomarker landscape. For the latter, (e) somatic alterations are explored for differential patient prognosis
 for each treatment arm. (f) Consecutively, for each treatment arm, subtype-specific biomarkers are derived. (g)
 Finally, interactions between treatment arms are examined. Here exemplified on the FIRE-3 clinical trial, (h) *RAS*
 mutations are established biomarkers of cetuximab resistance. (i) Patients with *RAS* wild type tumours showed a
 better prognosis when treated with cetuximab within left-sided tumours compared to right-sided tumours. In
 addition, (j) the *RAS* wild type subpopulation in left-sided tumours showed benefits when treated with cetuximab
 compared to bevacizumab.

 In addition, we have revised the **Results** section accordingly to reflect these changes:

“*OncoBird* is applicable to RCTs accompanied with molecular characteristics including
 genetic sequencing panels which yield copy number variations and somatic driver
 mutations (Fig. 1a,b). In addition, a second layer of stratification can be supplied in the
 form of the predefined tumour subtypes (Fig. 1a). Then, *OncoBird* systematically
 assesses the genetic landscape in the context of tumour subtypes (Fig. 1c) and

outlines the biomarker landscape across multiple clinical responses (**Fig. 1d**), i.e. time-
to-event data (overall or progression-free survival; **Methods**), and binary variables
capturing treatment success (objective response rate; **Methods**).

[...]

[...] To reveal the biomarker landscape we employed the following stratification and
modelling strategies (**Methods, Algorithm 1**): We first investigated each alteration for
stratifying patients by their prognosis within each treatment arm (**Fig. 1e**).
Consecutively, we inspected alterations in tumour subtypes (**Fig. 1f**), revealing
subtype-specific biomarkers. Finally, we tested for treatment interactions to reveal
biomarkers with predictive effects (**Fig. 1g**). Importantly, subtypes and genetic
alterations ought to be independent from the treatment assignment. The molecular
landscape and individual treatment arm analysis could be applied to any trial design
without any limitations.”

Furthermore, we clarify in the method section that any binary feature could be used as input
to stratify the patient cohort.

“The **Oncology Biomarker Discovery** (*OncoBird*) framework applies to RCTs for which
patients received either treatment $t \in \{0, 1\}$ according to the treatment indicator T ,
had an associated outcome Y and can be classified into q subtypes $\{s_1, \dots, s_q\}$
according to the subtype variable S (clinical data). Additionally, patient tumours are
characterised by m candidate genetic biomarkers $\mathbf{X} = X_1, \dots, X_m$ and the
observed biomarkers for patients are $\mathbf{x} = x_1, \dots, x_m$ (genetic data). Whilst only
genetic data can be used to group functionally similar genes, it is possible to add
additional binary features to \mathbf{X} such as binarised copy number alterations with
appropriate cutoffs or the MSI status of a tumour. Both genetic data (*MUT*) and clinical
data (*CLIN*) are required inputs to the *OncoBird* workflow (**Algorithm 1**). All analysis
steps of *OncoBird* are described in the following sections.”

Below is the added pseudocode of *OncoBird* to support transparency and reproducibility:

Input: *MUT* // $n \times m$ dataframe containing n patients and m binary mutational alterations including single nucleotide variants ('SV'), amplifications ('AMP') and deletions ('DEL')
CLIN // $n \times k$ dataframe containing n patients with k columns containing clinical endpoints, a treatment indicator ('treatment'), subtypes ('subtypes') and additional covariates

Algorithm 1 OncoBird workflow

```

1: function GET-MUTATIONS-IN-SUBTYPES(MUT, CLIN)
   // Get enrichments of mutations in tumour subtypes
2:   scores.e ← list() // initialise empty scores list for enrichments
3:    $q \leftarrow \text{LENGTH}(\text{CLIN}[\text{'subtypes'}])$  // count how many subtypes in column
4:   for  $j = 1$  to  $m$  do
5:     for  $s = 1$  to  $q$  do
6:       score ← HYPERGEOMETRIC-TEST( $j, s, \text{MUT}, \text{CLIN}$ ) // test for enrichments
7:       scores.e ← APPEND(score, scores.e)
8:     end for
9:   end for
10:  scores.e.adjusted ← HOLM-CORRECTION(scores.e) // multiplicity adjustment for all scores
11:  return scores.e.adjusted
12: end function
13:
14: function GET-MUTATIONS-MODULES(MUT)
   // Call mutually exclusive modules of somatic mutations
15:  MUT.modules ← MUTEX-ALGORITHM(MUT)
16:  return MUT.modules
17: end function
18:
19: function GET-TREATMENT-SPECIFIC-BIOMARKERS(MUT, CLIN)
   // Model outcome for each treatment arm and FDR multiplicity correction
20:  scores.tsb ← list() // initialise empty scores list for treatment-specific biomarkers
21:   $T \leftarrow \text{LENGTH}(\text{CLIN}[\text{'treatment'}])$  // count how many treatments in column
22:   $q \leftarrow \text{LENGTH}(\text{CLIN}[\text{'subtypes'}])$  // count how many subtypes in column
23:  for  $t = 1$  to  $T$  do
24:    for  $j = 1$  to  $m$  do
25:      score.t ← TREATMENT-SPECIFIC-MODEL-FIT( $t, j, \text{MUT}, \text{CLIN}$ ) // subtype-independent
   // models
26:      scores.tsb ← APPEND(score.t, scores.tsb)
27:    end for
28:    scores.tsb.adjusted ← BENJAMINI-HOCHBERG-CORRECTION( $t, \text{scores.tsb}$ ) // multiplicity
   // adjustment for each treatment arm
29:    for  $s = 1$  to  $q$  do
30:      for  $j = 1$  to  $m$  do
31:        score.t ← TREATMENT-SPECIFIC-MODEL-FIT( $t, j, s, \text{MUT}, \text{CLIN}$ ) // subtype-specific
   // models
32:        scores.tsb ← APPEND(score.t, scores.tsb)
33:      end for
34:    end for
35:    scores.tsb.adjusted ← BENJAMINI-HOCHBERG-CORRECTION( $t, \text{scores.tsb}$ ) // multiplicity
   // adjustment for each treatment arm across subtypes
36:  end for
37:  return scores.tsb.adjusted
38: end function
39:
40: function GET-PREDICTIVE-BIOMARKERS(MUT, CLIN)
   // Model outcome with treatment interactions and FDR multiplicity correction
41:  scores.pd ← list() // initialise empty scores list for predictive biomarkers

```

2

```
42: q ← LENGTH(CLIN['subtypes']) // count how many subtypes in column
43: for j = 1 to m do
44:   score.pb ← INTERACTION-MODEL-FIT(j, MUT, CLIN) // subtype-independent models
45:   scores.pb ← APPEND(score.pb, scores.pb)
46: end for
47: scores.pb.adjusted ← BENJAMINI-HOCHBERG-CORRECTION(scores.pb) // multiplicity
   adjustment for all scores
48: for s = 1 to q do
49:   for j = 1 to m do
50:     score.pb ← INTERACTION-MODEL-FIT(j, s, MUT, CLIN) // subtype-specific models
51:     scores.pb ← APPEND(score.pb, scores.pb)
52:   end for
53: end for
54: scores.pb.adjusted ← BENJAMINI-HOCHBERG-CORRECTION(scores.pb) // multiplicity
   adjustment across subtypes
55: return scores.pb.adjusted
56: end function
57:
58: function GET-CONDITIONAL-AVERAGE-TREATMENT-EFFECT(MUT, CLIN, scores.pb.adjusted,
   scores.tsb.adjusted, fdr.pb, fdr.tsb)
   // Get subgroups from biomarkers with predictive components and estimate conditional average
   treatment effects with resampling
59: scores.cate ← list() // initialise empty scores list for treatment effects
60: subgroup ← GET-PREDICTIVE-GENE-IN-SUBTYPE(
   scores.pb.adjusted < fdr.pb,
   scores.tsb.adjusted < fdr.tsb) // get biomarkers and subtypes with treatment-specific and
   predictive component
61: scores.cate ← GET-CATE(subgroup, MUT, CLIN) // get unadjusted treatment effect in
   subgroup
62: scores.cate.adjusted ← PERMUTATION-CORRECTION(scores.cate) // adjust treatment effect
   p-values
63: scores.cate.adjusted.CIs ← BOOTSTRAP-CORRECTION(scores.cate) // adjust treatment effect
   confidence intervals
64: return scores.cate.adjusted, scores.cate.adjusted.CIs
65: end function
66:
67: // Run OncoBird

68: GET-MUTATIONS-IN-SUBTYPES(MUT, CLIN)
69: GET-MUTATIONS-MODULES(MUT)
70: GET-TREATMENT-SPECIFIC-BIOMARKERS(MUT, CLIN)
71: GET-PREDICTIVE-BIOMARKERS(MUT, CLIN)
72: GET-CONDITIONAL-AVERAGE-TREATMENT-EFFECT(MUT, CLIN,
   GET-PREDICTIVE-BIOMARKERS(),
   GET-TREATMENT-SPECIFIC-BIOMARKERS(),
   fdr.pb,
   fdr.tsb)
```

**Are there any recommendations on how to score certain molecular alterations?**

If we understand the reviewer correctly, the comment addresses the ranking of cancer genes
according to their importance independent of treatment records. Unfortunately, *OncoBird* is
neither designed to identify novel oncogenes, tumour suppressors, nor does it weights cancer

genes *a priori*. For identifying novel cancer genes, larger deep molecular characterised cohorts
are required, e.g. TCGA or ICGC. However, we recommend that a genetic cancer functional
event should at least occur in 10 patients (adjustable default cut-off to perform statistical tests),
thus warranting clinical translatability.

**Is it also possible to include for example chromosomal regions instead of copy**
**numbers per gene?**

We apologise that this was not clearly communicated. As highlighted above (response to point
4; **Fig. 1a**), the input features generalise to any binary feature. Therefore, we support binarized
copy number alterations of genes and chromosomal regions.

**Reviewer #3, expert in bioinformatics analysis (Remarks to the Author):**

**In this paper, the author developed a kind of Cancer Biomarker Discovery Toolkit**
**(CBDT) to identify biomarkers based on single genes or mutually exclusive genetic**
**alteration patterns in tumour subtypes. The authors have tried to identify the**
**biomarkers with predictive components, e.g. mutual exclusivity of chr20q**
**amplifications and ERK signaling mutations resulting in treatment benefits with**
**cetuximab over bevacizumab. In summary, the results in this paper are prepared with a**
**series of analysis, which is enough to show the capability of the CBDT method. Many**
**data from clinical trial are also provided to verify the hypothesis from CBDT. From my**
**point of view, the work is well-done and provides interesting results to the CBDT and**
**thus it merits to be published.**

We thank the reviewer for the positive and constructive feedback. We would like to highlight
that we have revised the method name to **Oncology Biomarker Discovery** (OncoBird)
according to the suggestions from Reviewer #1 to further highlight the novel methodology
character.

**Just, I suggest some minor modifications before publication:**

**1. The manuscript is not so easy to read, since it contains a lot of information about**
**both logical frameworks and biological terms. It is difficult for the reader to catch the**
**essential elements in this work.**

We fully agree with the reviewer that the methodology advances were not appropriately
 conveyed, which was also raised by the other reviewers. Therefore, we have majorly revised
 **Fig. 1** in order to demonstrate the method design and analysis workflow:

 **Figure 1: The Oncology Biomarker Discovery (OncoBird) workflow.** (a) Patients in clinical trials are
 characterised according to tumour somatic mutations and copy number alterations (M), tumour subtypes (S) and
 were treated with two treatment regimens (T). (b) With this input, OncoBird outlines (c) the molecular landscape
 and (d) the biomarker landscape. For the latter, (e) somatic alterations are explored for differential patient prognosis
 for each treatment arm. (f) Consecutively, for each treatment arm, subtype-specific biomarkers are derived. (g)
 Finally, interactions between treatment arms are examined. Here exemplified on the FIRE-3 clinical trial, (h) *RAS*
 mutations are established biomarkers of cetuximab resistance. (i) Patients with *RAS* wild type tumours showed a
 better prognosis when treated with cetuximab within left-sided tumours compared to right-sided tumours. In
 addition, (j) the *RAS* wild type subpopulation in left-sided tumours showed benefits when treated with cetuximab
 compared to bevacizumab.

 In addition, we modified the **Results** section accordingly:

 “OncoBird is applicable to RCTs accompanied with molecular characteristics including
 genetic sequencing panels which yield copy number variations and somatic driver

mutations (**Fig. 1a**). In addition, a second layer of stratification can be supplied in the
form of the predefined tumour subtypes (**Fig. 1a**). Then, *OncoBird* systematically
assesses the genetic landscape in the context of tumour subtypes (**Fig. 1b**) and
outlines the biomarker landscape across multiple clinical responses (**Fig. 1c**), i.e. time-
to-event data (overall or progression-free survival; **Methods**), and binary variables
capturing treatment success (objective response rate; **Methods**). [...]

[...] To reveal the biomarker landscape we employed the following stratification and
modelling strategies (**Methods, Algorithm 1**): We first investigated each alteration for
stratifying patients by their prognosis within each treatment arm (**Fig. 1d**).
Consecutively, we inspected alterations in tumour subtypes (**Fig. 1e**), revealing
subtype-specific biomarkers. Finally, we tested for treatment interactions for the
previously outlined stratification strategies to reveal biomarkers with predictive effects
(**Fig. 1f**). Importantly, subtypes and genetic alterations ought to be independent from
the treatment assignment. The molecular landscape and individual treatment arm
analysis could be applied to any trial design without any limitations.”

Furthermore, we expanded the Method section including more details on the implementation
of *OncoBird*:

“**Oncology Biomarker Discovery workflow**

The **Oncology Biomarker Discovery** (*OncoBird*) framework applies to RCTs for which
patients received either treatment $t \in \{0, 1\}$ according to the treatment indicator T ,
had an associated outcome Y and can be classified into q subtypes $\{s_1, \dots, s_q\}$
according to the subtype variable S (clinical data). Additionally, patient tumours are
characterised by m candidate genetic biomarkers $\mathbf{X} = X_1, \dots, X_m$ and the
observed biomarkers for patients are $\mathbf{x} = x_1, \dots, x_m$ (genetic data). Whilst only
genetic data can be used to group functionally similar genes, it is possible to add
additional binary features to \mathbf{X} such as binarised copy number alterations with
appropriate cutoffs or the MSI status of a tumour. Both genetic data (*MUT*) and clinical
data (*CLIN*) are required inputs to the *OncoBird* workflow (**Algorithm 1**). All analysis
steps of *OncoBird* are described in the following sections. [...]

[...] *OncoBird* tests single somatic alterations and previously derived mutually
exclusive somatic alterations for differential prognosis in each treatment arm
separately (function `GET-TREATMENT-SPECIFIC-BIOMARKERS` in **Algorithm 1**).
The patient outcome Y may be defined by survival data (OS or PFS) or a binary
variable measuring the objective response rate (ORR). We modelled the expected

outcome for an endpoint $Y(T = t, S = s_k)$ of the treatment arm $T = t$ in subtype
 $S = s_k$ with $k = 1, \dots, q$ dependent on biomarkers $\mathbf{X} = \mathbf{x}$ using regression models
 as a classical approach for subgroup analysis. The treatment-specific regression
 models in subtypes take the form

$$\mathbb{E}(Y(T = t, S = s_k) | \mathbf{X} = \mathbf{x}) = f(\alpha_{0j} + \alpha_{1j}x_j + \sum_l C_l),$$

where f is either a hazard function for a Cox proportional hazards model or a logistic
 function for logistic regression dependent on the type of outcome.

Cox proportional hazards regression models for survival endpoints were implemented
 with the 'coxph' function from the *survival* R package or a logistic regression model for
 binary response variables were implemented using the 'glm' function. We test each
 $\mathbf{x} = x_1, \dots, x_m$ first across all tumours, and subsequently in tumour subtypes
 $\{s_1, \dots, s_q\}$, i.e. CMS or primary sidedness. α_{1j} is the coefficient estimating the
 contribution of candidate biomarker $j = 1, \dots, m$ for patient outcomes in the context
 of each treatment arm $T = t$ in the subtype $S = s_k$. The predictors C_1, \dots, C_l include
 additional prognostic covariates and their coefficients.

The p-value $p_{\alpha_{1j}}$ derived from the coefficient α_{j1} is multiplicity-adjusted for each
 treatment arm t and across all biomarkers x_j with $j = 1, \dots, m$ for either all patients
 or across subtypes s_k with $k = 1, \dots, q$ and yields adjusted p-values $\tilde{p}_{\alpha_{1j}}$ using the
 Benjamini-Hochberg (BH) method⁶². The false discovery rates (FDR) are controlled at
 $\text{FDR}_\alpha = 0.1$ for either treatment-specific component α_{1j} .

The adjustable default setting of *OncoBird* is to only perform statistical tests if for a
 given genetic alteration \mathbf{x} and tumour subtype s_k , at least $n = 10$ samples were present
 in each mutant and wild type population. Additionally, *OncoBird* only tested alterations
 for which its corresponding gene module had at least n tumours redistributed compared
 to the single gene alteration. [...]

 [...] For the subsequent comparison of treatment arms, *OncoBird* tests for significant
 statistical interactions between treatment arms and genetic alterations in tumour
 subtypes (function `GET-PREDICTIVE-BIOMARKERS` in **Algorithm 1**). For that, we
 modelled the outcome $Y(S = s_k)$ in subtype $S = s_k$ with $k = 1, \dots, q$ using
 regression models with interactions between T and X_j which take the form

$$\mathbb{E}(Y(S = s_k) | \mathbf{X} = \mathbf{x}, T = t) = f(\beta_{0j} + \beta_{1j}x_j + \beta_{2j}x_jt + \sum_l C_l),$$

where the coefficients β_{1j} and β_{2j} estimate the prognostic and predictive component
 of biomarker j in subtype s_k , respectively. The p-value $P_{\beta_{2j}}$ derived from the coefficient
 β_{2j} is multiplicity-adjusted across all m biomarkers for either all patients or across
 subtypes s_k with $k = 1, \dots, q$ and yields adjusted p-values $\tilde{p}_{\beta_{2j}}$. The FDR is controlled
 at $\text{FDR}_{\beta} = 0.2$ for predictive components. The biomarker X_j in subtype s_k is a
 putatively predictive biomarker if $\tilde{p}_{\alpha_{1j}} < \text{FDR}_{\alpha}$ for either t and $\tilde{p}_{\beta_{2j}} < \text{FDR}_{\beta}$.

Furthermore *OncoBird* only performs statistical tests if for a given genetic alteration \mathbf{x}
 and tumour subtype s_k , at least $n = 10$ samples were present in each mutant and wild
 type population for each treatment arm as default setting.”

 In addition, we would like to highlight that *OncoBird* delivers novel insights by 1) characterising
 the molecular landscape (Fig. 1c), and 2) biomarker landscape of any randomised clinical trial
 (Fig. 1d). Here exemplified, we show the molecular landscape of the FIRE-3 clinical trial (Fig.
 2):

 **Figure 2: Molecular landscape of the FIRE-3 clinical trial.** (a) Oncoprint of 373 mCRC tumours including
 mutations and copy number alterations detected in more than 12 tumours. (b) The mutually exclusive alteration
 patterns were derived with the *Mutex* algorithm. Gene expression profiles of 451 mCRC tumours are annotated by

(c) the consensus molecular subtypes (CMS) and (d) the primary tumour side. (e) Venn diagram showing all
enriched somatic alterations for CMS1 and right-sided tumours, and (f) enriched somatic alterations for CMS2 and
left-sided tumours. (g) Frequently altered cancer genes tested for enrichment in left- or right-sided tumours, and
(h) tested against CMS subtypes.

In addition, we have further restructured the **Abstract** and objective paragraph of the
**Introduction** to highlight the methodology advancements exemplified with the FIRE-3 trial:

“**Abstract**

Precision medicine has revolutionised cancer treatments, however, actionable
biomarkers remain scarce. To address this, we developed the Oncology Biomarker
Discovery (*OncoBird*) framework for analysing the molecular and biomarker landscape
of randomised controlled clinical trials. *OncoBird* identifies biomarkers based on single
genes or mutually exclusive genetic alterations in isolation or in context of tumour
subtypes, and finally, assesses the predictive component by their treatment
interactions. Here, we utilised the open-label, randomised phase III trial (FIRE-3, AIO
KRK 0306) in metastatic colorectal carcinoma patients, who received either cetuximab
or bevacizumab in combination with 5-fluorouracil, folinic acid and irinotecan
(FOLFIRI). We systematically identified five biomarkers with predictive components,
e.g. patients carrying chr20q amplifications and mutually exclusive ERK signalling
mutations benefited from cetuximab compared to bevacizumab. In summary, *OncoBird*
characterises the molecular landscape and outlines actionable biomarkers, which
generalises to any molecularly characterised randomised controlled trial.”

“**Introduction**

[...] The *OncoBird* workflow is divided into five distinct steps: it systematically 1)
investigates the molecular landscape of a clinical trial, i.e. copy number variants,
somatic mutations, mutually exclusive patterns and predefined tumour subtypes; 2)
identifies biomarkers within a treatment arm based on genetic alterations, and 3) in
relation to the predefined tumour subtypes; consecutively, 4) evaluates their predictive
component across treatment arms; and finally, 5) *OncoBird* comprehensively corrects
for multiple hypothesis testing and computationally validates biomarkers based on
resampling methods. This integrated analysis allows the molecular characterisation of
clinical trials and the identification of subtype-specific biomarkers with treatment
benefits in an interpretable and transparent manner and therefore operates
complementary to existing methods. The utility of *OncoBird* is exemplified by the
application to the FIRE-3 clinical trial, generalises to the ADJUVANT clinical trial, and

in fact, would generalise to any molecularly characterised randomised controlled trials
(RCT) in oncology.”

Furthermore, we have detangled the insights gained from the *OncoBird* biomarker analysis
leveraging the FIRE-3 trial as an example (for details, see responses to point 3 and 5 below).
In our humble opinion, the restructuring of the manuscript improved the read flow and clarified
strengths and novelties of *OncoBird*.

**The CBDT should be compared with other integrative analysis tools, in order to**
**demonstrate its merits and capability.**

We fully agree with the reviewer that a thorough benchmark is a requisite for any novel
methodology, which may have been too shallowly explored in the previous manuscript. In order
to address this, we revised and majorly extended the section ‘Benchmarking of methods for
subgroup analysis’ and added a new **Fig. 6** to the main **Results** section. Since there is no
ground truth on subgroup analysis methods, we compared the clinical standard and reported
a hazard ratio for newly proposed subgroups for the model evaluations:

“**Benchmarking of methods for subgroup analysis**”

For benchmarking *OncoBird*, we compared it to alternative methods which can be used
to investigate predictive biomarkers based on overall survival. Nine algorithms and
implementations were used in order to identify subgroups with differential treatment
effects, i.e. virtual twins (VT)⁹, model-based partitioning (MOB)⁸, an outcome-
weighting method (OWE)¹¹, causal random forests (CRF)¹², policy learning (POL)⁴⁴,
GUIDE⁴⁵ and PRISM⁴⁶ (**Table S3; Methods; Fig. 6a**).

For the evaluation, we referred to the current treatment guidelines for mCRC, according
to which patients should receive cetuximab if their tumours are *RAS* wild type and left-
sided (std; **Fig. 6a**)³⁴. On average, 73% of methods identified cetuximab benefit for a
patient in the std-positive subgroup, whereas only 33% of methods detected further
benefits in the std-negative subgroup (**Fig. 6a**). 7/8 (88%) methods found mutually
exclusive alterations in *KRAS*, *NRAS* or *BRAF* as a predictive biomarker, from which
one, two and four methods proposed this marker in conjunction with tumour sidedness,
CMS and across all patients, respectively (**Table S3**). Only 2/8 (25%) methods
highlighted *TOP1* amplifications as a potential biomarker (**Table S3**). This highlights
that current subgroup analysis methods mostly recover clinical standard practice,
whilst sparsely identifying novel predictive subgroups, thus highlighting the unmet need
for cancer biology driven frameworks such as *OncoBird*.

Ideally, subgroup analysis should reveal novel subgroups with high treatment effects
 for refining treatment strategies and recover subgroups in the standard treatment
 strategy. Therefore, we evaluated the newly proposed subgroup for which standard
 treatment is not recommended (new-std-negative) for each method. We derived the
 hazard ratios for cetuximab benefit based on OS for all methods in the new-std-
 negative subgroups (Fig. 6b). Lower hazard ratios in new-std-negative patients
 indicate the discovery of off-label subgroups for which cetuximab is currently not
 recommended (Fig. 6b). Accordingly, *OncoBird* showed the numerically lowest hazard
 ratio HR = 0.57 (p = 0.16, N = 29) for the new-std-negative subgroup compared to all
 other methods (Table S3; Fig. 6b).

In summary, many of the computational methods reproduced the clinically established
 biomarkers, whilst *OncoBird* empowers advanced biomarker identification by its
 thorough integration of biological priors in the form of tumour subtypes. The simplicity
 of statistical models leveraged in *OncoBird* further increases interpretability and
 transparency.”

 Below you can find the newly added Figure 6:

 **Figure 6: Benchmark and comparison to other methods.** (a) Oncoprint showing identified subgroups for the
 benchmarked methods, including standard treatment guidelines (std), CMS subtypes, tumour sidedness and
 mutations in *KRAS* and *NRAS*. (b) Forest plot showing hazard ratios and amount of patients in the subgroup for
 which standard treatment is not recommended and which was found by subgroup analysis methods (new-std-
 negative).

 **The data in Table S2 should be further explained, including the time efficiency,**
 **accuracy, cost and so on.**

Thanks for this suggestion. We expanded Table S2 (which is now Table S3) by ‘output type’,
‘found subgroups’, ‘hazard ratio for subgroups not in standard treatment’ and ‘execution time’.
Furthermore, we have added legends annotating all columns in the table.

**2. The scientific advances in this paper is not enough. The CBDT is a data-driven**
**subgroup analyzing method. However, in order to fit the standard of Nat Common, the**
**author should also provide some interesting findings about cancer treatment or**
**biomarker development based on this CBDT.**

We thank the reviewer for this invaluable comment, and agree that novelties of our method
and results were not clearly enough communicated before. Our computational framework
offers much more than just a subgroup analysis. In order to address this, we would like to
highlight the novelty of our method itself, which gives 1) deep insights in the molecular and 2)
biomarker landscape of any randomised oncology clinical trial. For highlighting the capabilities
of *OncoBird*, we majorly revised **Fig. 1** and the **Results** section (see above).

In addition, we have exemplified *OncoBird*'s utility with application to the FIRE-3 clinical trial.
In particular, we would like to highlight that we consider this the most comprehensive molecular
characterisation of FIRE-3 so far (**Fig. 2**; see above). Furthermore, we reproduced *KRAS*
mutations as biomarkers in CMS4, and revealed the predictive component of chr20q
amplifications in CMS2, which we consider to be novel. Whilst clinical validation is challenging
and ongoing (out-of-scope for this manuscript), in the meantime, we expanded our
computational validation efforts. This is, we incorporated resampling-based corrections of the
conditional average treatment effects in subgroups for raising the evidence of benefits in the
found subgroups, which is described in the majorly revised **Methods** section:

**“Resampling for correction of conditional average treatment effects**

Lastly, we estimate the conditional average treatment effect (CATE) for the found
biomarkers (function `GET-PREDICTIVE-BIOMARKERS` in **Algorithm 1**). For each
significant X_j in s_k , there is one CATE estimate in each found subpopulation with a
positive (‘mutant’) biomarker $x_j = 1$ and negative (‘wild type’) biomarker $x_j = 0$. In
each population, we estimate the CATE by modelling the outcome Y by

$$\mathbb{E}(Y(S = s_k, X_j = x_j) | \mathbf{X} = \mathbf{x}, T = t) = f(\gamma_0 + \gamma_1 t + \sum_l C_l),$$

where γ_1 estimates the (biased) CATE in terms of either hazard ratios or odds ratios
dependent on outcome type in the subgroup defined by biomarker x_j and subtype s_k .

The population with the larger absolute estimate γ_1 is used to estimate the subgroups
A_{x_j, s_k} .

For each found subgroup A , we assess the significance to the associated CATE
estimate γ_1 and derive the p-value p_{γ_1} . Furthermore, we perform a multiplicity-
adjustment of p_{γ_1} and derive honest estimates of the CATE.

The p-values are adjusted for multiplicity using a permutation-based approach that
takes into account the entire subgroup search strategy³. For that, we permuted the
treatment labels $U = 1000$ times to obtain null datasets without any differential
treatment effects. Next, for each null dataset, we select significant subgroups $A^{(u)}$ for
the same thresholds and record the treatment effect p-value of the best subgroup $p^{(u)}$
with $u = 1, \dots, U$. The adjusted p-values are then given by

$$\tilde{p}_{\gamma_1} = \frac{1}{U} \sum_{u=1}^U I_{\{p^{(u)} \leq p_{\gamma_1}\}}(p^{(u)}),$$

the fraction of p-values $p^{(u)}$ that are smaller or equal than p_{γ_1} with the indicator function
I . Furthermore, we derive an honest estimate of the treatment effect γ_1 . Since
subgroups A are derived from the same data as the treatment effect estimates, the
treatment estimates from resubstitution $\gamma_1(A_{x_j, s_k})$ will be biased. In order to derive a
bias-corrected estimate $\tilde{\gamma}_1$, we use a previously proposed non-parametric bootstrap
approach⁹. For that, we generated $B = 500$ bootstrapped datasets. For each resampled
dataset $b = 1, \dots, B$ we estimate subgroups $\hat{A}_{x_j, s_k}^{(b)}$. The treatment effects can then
be either estimated on the b-th resampled dataset $\gamma_1^{(b)}(A^{(b)})$ or on the original dataset
$\gamma_1(A^{(b)})$. The bias-corrected CATE estimate is then given by

$$\tilde{\gamma}_1 = \frac{1}{B} \sum_{b=1}^B \left(\gamma_1(A) + \gamma_1(A^{(b)}) - \gamma_1^{(b)}(A^{(b)}) \right).$$

The 95% confidence intervals are constructed by the 0.025 and 0.975 quantiles of the
bootstrapped distribution.”

In addition, we would like to highlight that *OncoBird* generalises to any randomised oncology
clinical trial, here, exemplified with application to the ADJUVANT clinical trial for gefitinib in
non-small cell lung cancer. Also for the ADJUVANT clinical trial, we reproduced all anticipated
associations, and generated novel biomarker hypotheses (see response to point 4 for more
details).

Hence the ADJUVANT clinical trial is publicly available, we used this additional dataset to
provide a runnable demo of our computational framework *OncoBird*, which increases
accessibility and interoperability. The reusability is further improved by FAIR (findable,
accessible, interoperable, reusable) sharing the full source code and comprehensive
documentation via codeocean and github:

**“Code Accessibility**

Oncology Biomarker Discovery (OncoBird) is publicly available at
<https://github.com/aljoshoh/OncoBird>. The repository contains an R package as well as
a *Shiny* application with a graphical user interface in a local docker container (**Fig. S1**).
Additionally, a demo run of OncoBird is available on codeocean
(<https://codeocean.com/capsule/3676298/tree>).”

Notably, we put great effort into making *OncoBird* accessible for clinicians by providing a
Graphical User Interface (GUI) based on a shiny application, which produces a report on the
molecular and biomarker landscape of any given clinical trial. For this, we have added
**Supplementary Fig. 1** for highlighting screenshots of *OncoBird*'s graphical user interface:

a

b Uploading Files

Choose "data_mutations"

Browse... data_mutations.csv

Upload complete

Choose "data_clinical"

Browse... data_clinical.csv

Upload complete

Select Clinical Endpoint column

OS

Select treatment column

treatment

Select patientID column

sample

Prepare data

Choose multiple columns for as Tumour subtypes

CMS primary site

Choose 2 Treatments

FOLFIRI/Bevacizumab FOLFIRI/Cetuximab

Choose up to 2 Covariates

resected.1stline metastatic sites

Submit

Recalculate Mutex

c Genomic Enrichments

d Mutual Exclusivity

e

f

g

h

Gene	HRV_10	HRV_15	HRV_20	HRV_25	HRV_30	HRV_35	HRV_40	HRV_45	HRV_50	HRV_55	HRV_60	HRV_65	HRV_70	HRV_75	HRV_80	HRV_85	HRV_90	HRV_95	HRV_100
1. HRV_10	2	0.01	0.02	0.03	0.04	0.05	0.06	0.07	0.08	0.09	0.10	0.11	0.12	0.13	0.14	0.15	0.16	0.17	0.18
2. HRV_15	4	0.01	0.02	0.03	0.04	0.05	0.06	0.07	0.08	0.09	0.10	0.11	0.12	0.13	0.14	0.15	0.16	0.17	0.18
3. HRV_20	4	0.01	0.02	0.03	0.04	0.05	0.06	0.07	0.08	0.09	0.10	0.11	0.12	0.13	0.14	0.15	0.16	0.17	0.18
4. HRV_25	4	0.01	0.02	0.03	0.04	0.05	0.06	0.07	0.08	0.09	0.10	0.11	0.12	0.13	0.14	0.15	0.16	0.17	0.18
5. HRV_30	4	0.01	0.02	0.03	0.04	0.05	0.06	0.07	0.08	0.09	0.10	0.11	0.12	0.13	0.14	0.15	0.16	0.17	0.18
6. HRV_35	2	0.02	0.03	0.04	0.05	0.06	0.07	0.08	0.09	0.10	0.11	0.12	0.13	0.14	0.15	0.16	0.17	0.18	0.19

Figure S1: OncoBird dockerised Shiny application. (a) Each tab corresponds to one analysis step, for example (b) the user interface for data input. Then, results generated for (c) enrichment of genetic alterations in tumour subtypes, (d) mutual exclusivity, treatment-specific biomarkers in (e) all tumours and (f) in tumour subtypes. Also, results for predictive biomarkers using (g) interaction tests and (h) the table summary.

In essence, we strongly believe that OncoBird will become a highly used computational
framework to analyse the molecular and biomarker landscape of historical and new clinical
trials, which has been exemplified with the FIRE-3 and ADJUVANT clinical trials.

**3. How to identify and compare the sensitivity and the reaction time of different**
**biomarker by using this CBBDT?**

If we understand the reviewer correctly, this is a concern regarding the repeatability or
reproducibility of our biomarker analysis. In order to address this, as described above in point
2, we majorly extended our validation framework by a bootstrapping procedure that executes
the whole workflow on bootstrapped datasets. From this, we are able to derive a robust
treatment effect estimate. This raised evidence and benefits in the found subgroups.

Furthermore, assessment of the reaction time would be desirable, but require longitudinal
follow-up of patients, which unfortunately is too sparse or not existing at all. Here, we only
considered common clinical endpoints such as overall survival, progression-free survival and
the objective response rate, which are the focus of our subgroup analysis and are standard
evaluation criteria of most clinical trials, thus increasing generalisability.

**More information should also be considered in CBDT, including the effect biomarkers**
**at molecular levels and the effects at the advanced biological level.**

We fully agree with the reviewer that the ultimate goal would be functionally interpretable
biomarkers. Data-driven, this can be partially achieved by leveraging only frequently altered
cancer genes, e.g. mutations causing oncogenic activation or loss-of-function of tumour
suppressors. *OncoBird* is inevitably enforcing this by leveraging the cancer gene sequencing
panel of FIRE-3, applying conservative frequency filters (**Methods**), and concordant
interpretations (see **Discussion** section for details):

“Several genes were co-amplified in chr20q, i.e. *ARFRP1*, *TOP1*, and *SRC*, thus
determining the drivers among these biomarker candidates is challenging. Among the
prominent chr20q amplifications, *TOP1* was previously proposed as a biomarker for
irinotecan efficacy in metastatic colorectal cancer^{49,50}, which is part of the
chemotherapeutic backbone of the FIRE-3 trial. Literature suggests that *TOP1*
abundance is essential for irinotecan-induced DNA double-strand breaks during DNA
replication⁵¹. Additionally, *TOP1* was identified to regulate EGFR through an
endogenous interaction with the transcription factor c-Jun⁵², which supports the
hypothesis that *TOP1* amplifications may be the actionable biomarker. *SRC* has been
reported to play a role in cancer progression^{53,54}, whereas for *ARFRP1* no functional
evidence has been presented yet. [...]

[...] The prognostic potential of *APC* wild type tumours for bevacizumab has been
previously reported⁵⁶, whereas *OncoBird* did not yield enough evidence to support
this. Indeed, a confounding factor is the enrichment of *BRAF* mutations in the *APC* wild
type tumours ($p = 1.4 \times 10^{-10}$, Fisher’s exact test). This is, 48% of *APC* wild type
tumours were *BRAF* mutated in the bevacizumab treatment arm, whereas in the
cetuximab treatment arm, only 29% were *BRAF* mutated ($p = 0.13$, Fisher’s exact test).
Nevertheless, independently a correlation between *VEGFA* expression and the
mutational status of *APC* has been previously observed in primary colorectal tumour

samples⁵⁷, suggesting that within *APC* mutated tumours, anti-VEGF treatment may
indeed be beneficial.

Furthermore, *RAS/BRAF* mutations are known to harbour prognostic value in terms of
overall survival^{38,43}. Furthermore, we observed that *KRAS* mutations showed highly
CMS-specific responses. In particular, treatment response differed for tumours
classified as CMS4 by *KRAS* status, showing better response for cetuximab in *KRAS*
wild type and for bevacizumab in *KRAS* mutated tumours, respectively. CMS4 has
been reported to be associated with VEGF pathway activation and is thus associated
with angiogenesis²⁹. Thus, patients with tumours resistant towards anti-EGFR
treatment may benefit from VEGF inhibition. Further exclusion of *BRAF* mutations did
not elevate the predictive potential of *KRAS* mutations in CMS4. However, the
statistical power is limited by the fact that only six harboured the prognostically
unfavourable *BRAF* V600E mutation in CMS4²⁰.”

In addition, we would like to highlight that *OncoBird* provides an in-built assessment of mutual
exclusivity which contributes to further functional interpretation. For example, we identified a
novel mutual exclusivity biomarker for the ADJUVANT clinical trial, which we highlight
accordingly:

“[...] In addition, *OncoBird* suggests that the mutual exclusivity patterns play a role in
the biomarker landscape of NSCLC (**Supplementary Note 1**). In more detail, we
observed gefitinib benefits in tumours that were characterised by mutations in either
*TP53*, *SMAD4* or *CDK4* amplifications ($p = 0.0002$, HR = 0.37 [0.21-0.63]), for which
the resampling-based validation of the conditional average treatment effect yielded p_{adj}
= 0.001 with HR = 0.32 [0.14-0.86] (**Supplementary Note 1**). These findings highlight
the accessibility, reproducibility and interoperability of *OncoBird*.”

**Also, the repeatability and individual differences of the selected biomarker analysis**
**need be in the acceptable range. The author should further confirm all these factors**

If we understand the reviewer correctly, these concerns include stability and repeatability of
the found biomarkers. We believe that this concern is partially addressed by the assessment
of permutations and bootstrapping analysis that was presented above. Additionally, we would
like to highlight that we employed a conservative frequency filter for somatic mutations to
increase confidence into found associations, i.e. at least patient group sizes of 10:

“The adjustable default setting of *OncoBird* is to only perform statistical tests if for a
given genetic alteration x and tumour subtype S_k , at least $n = 10$ samples were present
in each mutant and wild type population.”

Furthermore, we believe that inconsistencies in reported FDR annotation and too lenient cut-
offs may have contributed to a confusion (also raised by Reviewer #2). In order to address
this, we conservatively revised the multiplicity corrections in the form of false discovery rates
(FDR) and employed more conservative default cut-offs. Furthermore, we recognised that
there is a high chance for confusion of different FDR thresholds, since the tool incorporates
three types of analyses and adjustments, i.e. molecular landscape analysis (FDR_{mol}),
treatment-specific ($FDR_{cet/bev}$) and interaction analysis (FDR_{int}). This annotation is now
consistently used across the whole manuscript.

For the enrichment test of the molecular landscape, we use a conservative $FDR_{mol} < 0.05$, and
all results remain conserved. For the identification of treatment- and subtype-specific genetic
biomarkers of FOLFIRI plus cetuximab or bevacizumab, we now also apply this more stringent
$FDR_{cet/bev} < 0.1$. One exception, in the **Results** section ‘Genetic biomarkers of bevacizumab’
and ‘Subtype-specific biomarkers of [...] bevacizumab’ we leveraged a more lenient threshold
to explore trends, which is clearly flagged to the reader:

“For exploring bevacizumab biomarker trends, we employed a lenient threshold of
$FDR_{bev} < 0.3$.”

Moreover, for the interaction biomarkers, we now consistently and more conservatively employ
a default $FDR_{int} < 0.2$. Notably, interaction tests subset the data, thereby reducing sample
sizes, and ultimately, decreasing statistical power, which is also the discussed limitation of our
method. Therefore, we believe that this reported interaction threshold is reasonable. For being
inclusive of putative biomarker trends, more lenient thresholds for the interaction biomarkers
are also reported in the supplementary tables (**Table S1**). Most importantly, all previously
highlighted examples remain significant, highlighting the robustness of our framework.

“In total, we found five putative interactions (**Methods**; **Table S2**; $FDR_{int} < 0.2$). For
reporting further biomarker trends, we also included summary statistics of 57 subgroups
with a lenient threshold of $FDR_{int} < 0.6$ (**Table S1**).”

According to the new stringent thresholds of $FDR_{mol} < 0.05$, we adjusted **Fig. 2g,h**:

Figure 2: Molecular landscape of the FIRE-3 clinical trial. [...] (g) Frequently altered cancer genes tested for enrichment in left- or right-sided tumours, and (h) tested against CMS subtypes.

And revised **Fig. 4a,b** with an $FDR_{cet/bev} < 0.1$:

Figure 4: Identification of subtype-specific genetic biomarkers for FOLFIRI plus cetuximab or bevacizumab. Subtype-specific genetic biomarkers for OS of (a) cetuximab and (b) bevacizumab using single Cox regression models including 95% confidence intervals (CI). [...]

And finally adjusted Fig. 5a,b with an threshold of $FDR_{int} < 0.2$:

**Figure 5: Predictive biomarkers in the context of tumour subtypes.** Overview of interaction biomarkers (FDR_{int}
< 0.2) focusing on (a) mutant and (b) wild type tumours when comparing cetuximab and bevacizumab treatment.
The models were fitted on OS, and error bars correspond to 95% CIs. Triangle points and confidence intervals
were obtained from bootstrap-based bias-correction of treatment effects. [...]

**4. How do you use this CBDT to evaluate new drugs or therapies?**

We thank the reviewer for raising this important point, since we believe that the generalisability
of *OncoBird* is a strong advantage of our computational framework. In fact, *OncoBird* can be
run on any randomised controlled clinical trial with genomic data and optional *a priori* defined
disease subtypes. The demo run on codeocean applies *OncoBird* on the publicly accessible
ADJUVANT clinical trial for gefitinib in non-small cell lung cancer (**Supplementary Figure 1**;
see point 2). We added the analysis regarding this publicly accessible and independent clinical
trial (**Supplementary Note 1**):

“Whilst here we focused on the FIRE-3 trial in colorectal cancer, we show the
generalisability of *OncoBird* by applying it to the ADJUVANT clinical trial, which
explores gefitinib in non-small cell lung cancer (NSCLC; **Supplementary Note 1**)^{35–37}.
The ADJUVANT study reported predictive components of five alterations, i.e. *TP53*
mutations, *RB1* alterations and copy number amplifications of *NKX2-1*, *CDK4* and *MYC*
1027³⁷. Four out of five biomarkers were concordantly identified for disease-free survival
with *OncoBird* and are presented in an extensive report ($FDR_{int} < 0.2$; **Supplementary**
**Note 1**). In addition, *OncoBird* suggests that the mutual exclusivity patterns play a role
in the biomarker landscape of NSCLC (**Supplementary Note 1**). In more detail, we
observed gefitinib benefits in tumours that were characterised by mutations in either
*TP53*, *SMAD4* or *CDK4* amplifications ($p = 0.0002$, HR = 0.37 [0.21-0.63]), for which
the resampling-based validation of the conditional average treatment effect yielded p_{adj}
= 0.001 with HR = 0.32 [0.14-0.86] (**Supplementary Note 1**). These findings highlight
the accessibility, reproducibility and interoperability of *OncoBird*.”

**Can machine learning technology be applied to data analysis?**

We thank the reviewer for suggesting this. Indeed, our method is based on regression models,
which is heavily related to other machine learning methods for subgroup analysis such as
virtual twins, generalised random forests, etc. We benchmarked our method against these
state-of-the-art frameworks based on statistical and machine learning models in the new
**Results** section ‘Benchmarking of methods for subgroup analysis’ (see response to point #1).

We experienced that methods which incorporate prior knowledge of cancer biology are
superior over purely data driven methods We raise this challenge in the **Introduction** section,
to sensitise the reader for this caveat:

“For this purpose, a large number of computational methods have been proposed and
discussed ^{3–5}, e.g. tree-based methods using recursive partitioning ^{6–8}, virtual twins ⁹,
outcome weighted methods ^{10,11}, causal forests ¹² and metalearners for estimating
heterogeneous treatment effects ¹³. However, most of these computational methods
neglect cancer biology, i.e. exploiting the molecular landscape of a clinical trial and
customising models to cancer subtypes and mutational patterns.”

**5. In figure 3, more information about using the combination of multiple biomarkers to**
**predict the prognosis of non-progressive or control patients should be provided?**

We regret that this information was not clearly conveyed in **Fig. 3**. Therefore, we expanded
the figure legend that provides more information about biomarker combinations and how they
predict prognosis depending on their mutational status in treatment arms:

Figure 3: Identification of genetic biomarkers for FOLFIRI plus cetuximab or bevacizumab. (a) Volcano plot for genetic biomarkers of cetuximab in the form of mutually exclusive gene modules or single gene mutations. Each point shows the effect of a particular group of alterations summarised by its p-value and hazard ratio. Exemplifying the most significant associations, Kaplan-Meier plots are shown for (b) the mutually exclusive module consisting of *RAS* and *BRAF* mutations, and (c) the amplification of *TOP1* treated with cetuximab. For investigating the biomarker composition, we focus on (d) resistance biomarkers of FOLFIRI plus cetuximab with $FDR_{cet} < 0.1$, (e) the composition of mutually exclusive genes, and (f) an oncoprint highlighting mutational frequencies of biomarker combinations. In like manner, (g) summarising cetuximab sensitivity biomarkers, and (h) their composition. (i) Karyoplot showing transcription start sites of co-amplified genes on chromosome 20q. (j) Volcano plot of the genetic biomarkers of bevacizumab with $FDR_{bev} < 0.3$. Kaplan-Meier plot of (k) mutations in *KRAS* or *BRAF* and (l) *APC* mutations treated with bevacizumab. The compositions of bevacizumab biomarkers are shown in Fig. S5a,b.

The prediction accuracy and the comparison with other analysis tools should be provided.

We thank the reviewer for this comment, which was also raised by Reviewer #2. As described
 and shown above, we 1) majorly extended our validation framework by a bootstrapping
 procedure (**Methods**), 2) conservatively revised our FDR strategy (**Methods**), and 3)
 comprehensively benchmarked *OncoBird* against other computational frameworks (**Fig. 6**; see
 above).

**6. It is quite difficult to understand the configuration of CBDT.**

We thank the reviewer for this comment and agree that the workflow was not accurately
 depicted, which was raised by all reviewers. In order to address this, we have majorly revised
 **Fig. 1** in order to demonstrate the method design better, including inputs and outputs of the
 workflow:

**Figure 1: The Oncology Biomarker Discovery (OncoBird) workflow.** (a) Patients in clinical trials are
 characterised according to tumour somatic mutations and copy number alterations (M), tumour subtypes (S) and
 were treated with two treatment regimens (T). (b) With this input, *OncoBird* outlines (c) the molecular landscape
 and (d) the biomarker landscape. For the latter, (e) somatic alterations are explored for differential patient prognosis
 for each treatment arm. (f) Consecutively, for each treatment arm, subtype-specific biomarkers are derived. (g)
 Finally, interactions between treatment arms are examined. Here exemplified on the FIRE-3 clinical trial, (h) *RAS*

mutations are established biomarkers of cetuximab resistance. (i) Patients with *RAS* wild type tumours showed a
better prognosis when treated with cetuximab within left-sided tumours compared to right-sided tumours. In
addition, (j) the *RAS* wild type subpopulation in left-sided tumours showed benefits when treated with cetuximab
compared to bevacizumab.

Accordingly, we updated the main **Results** section to describe the workflow:

“*OncoBird* is applicable to RCTs accompanied with molecular characteristics including
genetic sequencing panels which yield copy number variations and somatic driver
mutations (**Fig. 1a**). In addition, a second layer of stratification can be supplied in the
form of the predefined tumour subtypes (**Fig. 1a**). Then, *OncoBird* systematically
assesses the genetic landscape in the context of tumour subtypes (**Fig. 1b**) and
outlines the biomarker landscape across multiple clinical responses (**Fig. 1c**), i.e. time-
to-event data (overall or progression-free survival; **Methods**), and binary variables
capturing treatment success (objective response rate; **Methods**). [...]

[...] To reveal the biomarker landscape we employed the following stratification and
modelling strategies (**Methods, Algorithm 1**): We first investigated each alteration for
stratifying patients by their prognosis within each treatment arm (**Fig. 1d**).
Consecutively, we inspected alterations in tumour subtypes (**Fig. 1e**), revealing
subtype-specific biomarkers. Finally, we tested for treatment interactions for the
previously outlined stratification strategies to reveal biomarkers with predictive effects
(**Fig. 1f**). Importantly, subtypes and genetic alterations ought to be independent from
the treatment assignment. The molecular landscape and individual treatment arm
analysis could be applied to any trial design without any limitations.”

Furthermore, in order to ensure the comprehensibility of our software, we demonstrate it in the
form of the mathematical framework and also pseudocode in **Algorithm 1** that constitutes our
method:

“The **Onco**logy **B**iomarker **D**iscovery (*OncoBird*) framework applies to RCTs for which
patients received either treatment $t \in \{0, 1\}$ according to the treatment indicator T ,
had an associated outcome Y and can be classified into q subtypes $\{s_1, \dots, s_q\}$
according to the subtype variable S (clinical data). Additionally, patient tumours are
characterised by m candidate genetic biomarkers $\mathbf{X} = X_1, \dots, X_m$ and the
observed biomarkers for patients are $\mathbf{x} = x_1, \dots, x_m$ (genetic data). Whilst only
genetic data can be used to group functionally similar genes, it is possible to add

additional binary features to \mathbf{X} such as binarised copy number alterations with
appropriate cutoffs or the MSI status of a tumour. Both genetic data (*MUT*) and clinical
data (*CLIN*) are required inputs to the *OncoBird* workflow (**Algorithm 1**). All analysis
steps of *OncoBird* are described in the following sections. [...]

[...] *OncoBird* tests single somatic alterations and previously derived mutually exclusive
somatic alterations for differential prognosis in each treatment arm separately (function
`GET-TREATMENT-SPECIFIC-BIOMARKERS` in **Algorithm 1**). The patient outcome Y
may be defined by survival data (OS or PFS) or a binary variable measuring the
objective response rate (ORR). We modelled the expected outcome for an endpoint
$Y(T = t, S = s_k)$ of the treatment arm $T = t$ in subtype $S = s_k$ with $k = 1, \dots, q$
dependent on biomarkers $\mathbf{X} = \mathbf{x}$ using univariate regression models as a classical
approach for subgroup analysis. The treatment-specific regression models in subtypes
take the form

$$\mathbb{E}(Y(T = t, S = s_k) | \mathbf{X} = \mathbf{x}) = f(\alpha_{0j} + \alpha_{1j}x_j + \sum_l C_l),$$

where f is either a hazard function for a Cox proportional hazards model or a logistic
function for logistic regression dependent on the type of outcome.

Cox proportional hazards regression models for survival endpoints were implemented
with the ‘coxph’ function from the *survival* R package or logistic regression models for
binary response variables were implemented using the ‘glm’ function. We test each
$\mathbf{x} = x_1, \dots, x_m$ first across all tumours, and subsequently in tumour subtypes
$\{s_1, \dots, s_q\}$, i.e. CMS or primary sidedness. α_{1j} is the coefficient estimating the
contribution of candidate biomarker $j = 1, \dots, m$ for patient outcomes in the context
of each treatment arm $T = t$ in the subtype $S = s_k$. The predictors C_1, \dots, C_l include
additional prognostic covariates and their coefficients.

The p-value $p_{\alpha_{1j}}$ derived from the coefficient α_{1j} is multiplicity-adjusted for each
treatment arm t and across all biomarkers x_j with $j = 1, \dots, m$ for either all patients
or across subtypes s_k with $k = 1, \dots, q$ and yields adjusted p-values $\tilde{p}_{\alpha_{1j}}$ using the
Benjamini-Hochberg (BH) method⁶². The false discovery rates (FDR) are controlled at
$\text{FDR}_\alpha = 0.1$ for either treatment-specific component α_{1j} . [...]

[...] For the subsequent comparison of treatment arms, *OncoBird* tests for significant
statistical interactions between treatment arms and genetic alterations in tumour
subtypes (function `GET-PREDICTIVE-BIOMARKERS` in **Algorithm 1**). For that, we

modelled the outcome $Y(S = s_k)$ in subtype $S = s_k$ with $k = 1, \dots, q$ using
 regression models with interactions between T and X_j which take the form

$$\mathbb{E}(Y(S = s_k) | \mathbf{X} = \mathbf{x}, T = t) = f(\beta_{0j} + \beta_{1j}x_j + \beta_{2j}x_jt + \sum_l C_l),$$
 where the coefficients β_{1j} and β_{2j} estimate the prognostic and predictive component
 of biomarker j in subtype s_k , respectively. The p-value $p_{\beta_{2j}}$ derived from the coefficient
 β_{2j} is multiplicity-adjusted across all m biomarkers for either all patients or across
 subtypes s_k with $k = 1, \dots, q$ and yields adjusted p-values $\tilde{p}_{\beta_{2j}}$. The FDR is controlled
 at $\text{FDR}_\beta = 0.2$ for predictive components. The biomarker X_j in subtype s_k is a
 putatively predictive biomarker if $\tilde{p}_{\alpha_{1j}} < \text{FDR}_\alpha$ for either t and $\tilde{p}_{\beta_{2j}} < \text{FDR}_\beta$.”

Input: *MUT* // $n \times m$ dataframe containing n patients and m binary mutational alterations including single nucleotide variants ('SV'), amplifications ('AMP') and deletions ('DEL')
CLIN // $n \times k$ dataframe containing n patients with k columns containing clinical endpoints, a treatment indicator ('treatment'), subtypes ('subtypes') and additional covariates

Algorithm 1 OncoBird workflow

```

1: function GET-MUTATIONS-IN-SUBTYPES(MUT, CLIN)
   // Get enrichments of mutations in tumour subtypes
2:   scores.e ← list() // initialise empty scores list for enrichments
3:    $q \leftarrow \text{LENGTH}(\text{CLIN}[\text{'subtypes'}])$  // count how many subtypes in column
4:   for  $j = 1$  to  $m$  do
5:     for  $s = 1$  to  $q$  do
6:       score ← HYPERGEOMETRIC-TEST( $j, s, \text{MUT}, \text{CLIN}$ ) // test for enrichments
7:       scores.e ← APPEND(score, scores.e)
8:     end for
9:   end for
10:  scores.e.adjusted ← HOLM-CORRECTION(scores.e) // multiplicity adjustment for all scores
11:  return scores.e.adjusted
12: end function
13:
14: function GET-MUTATIONS-MODULES(MUT)
   // Call mutually exclusive modules of somatic mutations
15:  MUT.modules ← MUTEX-ALGORITHM(MUT)
16:  return MUT.modules
17: end function
18:
19: function GET-TREATMENT-SPECIFIC-BIOMARKERS(MUT, CLIN)
   // Model outcome for each treatment arm and FDR multiplicity correction
20:  scores.tsb ← list() // initialise empty scores list for treatment-specific biomarkers
21:   $T \leftarrow \text{LENGTH}(\text{CLIN}[\text{'treatment'}])$  // count how many treatments in column
22:   $q \leftarrow \text{LENGTH}(\text{CLIN}[\text{'subtypes'}])$  // count how many subtypes in column
23:  for  $t = 1$  to  $T$  do
24:    for  $j = 1$  to  $m$  do
25:      score.t ← TREATMENT-SPECIFIC-MODEL-FIT( $t, j, \text{MUT}, \text{CLIN}$ ) // subtype-independent
   // models
26:      scores.tsb ← APPEND(score.t, scores.tsb)
27:    end for
28:    scores.tsb.adjusted ← BENJAMINI-HOCHBERG-CORRECTION( $t, \text{scores.tsb}$ ) // multiplicity
   // adjustment for each treatment arm
29:    for  $s = 1$  to  $q$  do
30:      for  $j = 1$  to  $m$  do
31:        score.t ← TREATMENT-SPECIFIC-MODEL-FIT( $t, j, s, \text{MUT}, \text{CLIN}$ ) // subtype-specific
   // models
32:        scores.tsb ← APPEND(score.t, scores.tsb)
33:      end for
34:    end for
35:    scores.tsb.adjusted ← BENJAMINI-HOCHBERG-CORRECTION( $t, \text{scores.tsb}$ ) // multiplicity
   // adjustment for each treatment arm across subtypes
36:  end for
37:  return scores.tsb.adjusted
38: end function
39:
40: function GET-PREDICTIVE-BIOMARKERS(MUT, CLIN)
   // Model outcome with treatment interactions and FDR multiplicity correction
41:  scores.pd ← list() // initialise empty scores list for predictive biomarkers

```

2

```
42: q ← LENGTH(CLIN['subtypes']) // count how many subtypes in column
43: for j = 1 to m do
44:   score.pb ← INTERACTION-MODEL-FIT(j, MUT, CLIN) // subtype-independent models
45:   scores.pb ← APPEND(score.pb, scores.pb)
46: end for
47: scores.pb.adjusted ← BENJAMINI-HOCHBERG-CORRECTION(scores.pb) // multiplicity
   adjustment for all scores
48: for s = 1 to q do
49:   for j = 1 to m do
50:     score.pb ← INTERACTION-MODEL-FIT(j, s, MUT, CLIN) // subtype-specific models
51:     scores.pb ← APPEND(score.pb, scores.pb)
52:   end for
53: end for
54: scores.pb.adjusted ← BENJAMINI-HOCHBERG-CORRECTION(scores.pb) // multiplicity
   adjustment across subtypes
55: return scores.pb.adjusted
56: end function
57:
58: function GET-CONDITIONAL-AVERAGE-TREATMENT-EFFECT(MUT, CLIN, scores.pb.adjusted,
   scores.tsb.adjusted, fdr.pb, fdr.tsb)
   // Get subgroups from biomarkers with predictive components and estimate conditional average
   treatment effects with resampling
59: scores.cate ← list() // initialise empty scores list for treatment effects
60: subgroup ← GET-PREDICTIVE-GENE-IN-SUBTYPE(
   scores.pb.adjusted < fdr.pb,
   scores.tsb.adjusted < fdr.tsb) // get biomarkers and subtypes with treatment-specific and
   predictive component
61: scores.cate ← GET-CATE(subgroup, MUT, CLIN) // get unadjusted treatment effect in
   subgroup
62: scores.cate.adjusted ← PERMUTATION-CORRECTION(scores.cate) // adjust treatment effect
   p-values
63: scores.cate.adjusted.CIs ← BOOTSTRAP-CORRECTION(scores.cate) // adjust treatment effect
   confidence intervals
64: return scores.cate.adjusted, scores.cate.adjusted.CIs
65: end function
66:
67: // Run OncoBird

68: GET-MUTATIONS-IN-SUBTYPES(MUT, CLIN)
69: GET-MUTATIONS-MODULES(MUT)
70: GET-TREATMENT-SPECIFIC-BIOMARKERS(MUT, CLIN)
71: GET-PREDICTIVE-BIOMARKERS(MUT, CLIN)
72: GET-CONDITIONAL-AVERAGE-TREATMENT-EFFECT(MUT, CLIN,
   GET-PREDICTIVE-BIOMARKERS(),
   GET-TREATMENT-SPECIFIC-BIOMARKERS(),
   fdr.pb,
   fdr.tsb)
```

Furthermore, we comprehensively describe the exact parameterisation of *OncoBird*, which
was chosen for the analysis of the FIRE-3 trial:

**“OncoBird parameterisation for FIRE-3**

We used the function `GET-MUTATIONS-IN-SUBTYPES` to evaluate the primary tumour
side and CMS as tumour subtypes with default settings. In total, we performed 156 and

312 statistical tests for the primary tumour sidedness and CMS, respectively. Using the
`GET-MUTATIONS-MODULES` function with default settings, we analysed 42 genes which
yielded 29 mutually exclusive modules. Mutations in *KRAS* or *NRAS* are the established
clinical biomarkers for anti-EGFR treatment, thus we jointly modelled *KRAS* and *NRAS* as
“*RAS*” mutations resulting in 10 additional modules.

The `GET-TREATMENT-SPECIFIC-BIOMARKERS` function was used with the number of
metastatic sites or the information about a prior tumour resection as added covariates
C_1, C_2 . With the *OncoBird* default setting, we performed 816 statistical tests across all
readouts Y (OS, PFS and ORR), the cetuximab and bevacizumab treatment arm and
tumour subtypes, i.e. CMS1-4, left- and right-sided and across all tumours. FDR cutoffs
are employed for each treatment arm separately and are denoted FDR_{cet} and FDR_{bev} for
the analysis in the cetuximab and bevacizumab treatment arm, respectively. In total, we
found 92 significant associations with $FDR_{\text{cet/bev}} < 0.1$. The criteria $HR < 1$ and $OR < 1$
corresponded to a better prognosis for the mutant tumours compared to the wild type
tumours and vice versa. To consistently report $HR < 1$ and $OR < 1$ as beneficial risk
reduction, reciprocal values of HRs and ORs were used if wild type tumours displayed a
better prognosis. We represent p-values, hazard/odds ratios with the 95% confidence
intervals (CI) in square brackets and the associated FDRs.

In FIRE-3, the `GET-PREDICTIVE-BIOMARKERS` function with default settings resulted in
a total amount of 396 statistical tests across the readouts Y (OS, PFS and ORR) and the
tumour subtypes s_k . FDR cutoffs for the interaction tests across both treatment arms are
denoted by FDR_{int} . Considering only associations with at least 10 patients in each
treatment arm, we explored 57 significant associations with $FDR_{\text{int}} < 0.6$ and $FDR_{\text{cet/bev}} <$
0.1 (**Table S1**). For the interaction tests, we focused on a subset of five biomarkers with
interaction $FDR_{\text{int}} < 0.2$ for OS, i.e. two gene modules and three single genes (**Table S2**).
HRs and ORs > 1 and < 1 corresponded to benefit with cetuximab and bevacizumab,
respectively. To report benefits for cetuximab treatment, the reciprocal values of HRs and
ORs were used in the manuscript in order to display treatment benefits consistently with
$HR < 1$ and $OR < 1$. We reported p-values and hazard/odds ratios with the 95% CIs for
the treatment comparison and the p-values and associated FDRs for the interaction tests.”

**There are too many small letters and curves in all the figures. Thus, the quality of all the**
**figures in this paper should be further improved, where the letters and the sketch**
**should be clear and informative. The author should modify Figures to guarantee the**
**readers’ smooth reading.**

We apologise for poor figure qualities and small font sizes. Now, all figures are provided as

original vector graphics with increased font size.

REVIEWER COMMENTS

Reviewer #1 (Remarks to the Author):

Comments only for the editors

Reviewer #2 (Remarks to the Author):

I would like to thank the authors for their extensive responses to the concerns I raised and their adaptations to the manuscript. I am satisfied by the answers and changes provided.

Reviewer #4 (Remarks to the Author):

The manuscript has been revised and improved.

While majority of comments from reviewer 3 have been addressed, some remain unanswered. For instance, repeatability of the biomarker analysis and effect at the biological levels. In particular, are biomarker levels/thresholds similar across datasets or across two clinical trials that you employed? If not - how will OncoBird deal with this? If the transcriptomic/genomic levels in different datasets differ - will the biomarkers still be discoverable? Are the results repeatable? Perhaps, comparison to the housekeeping genes could be performed? (or any other benchmarking).

It is not clear if Fig 1 describes the design correctly. It looks like randomized clinical trial design should be the first step in the process, followed by measuring treatment response, subtyping, and genetic markers, which are embedded in the clinical trial. Since OncoBird proposed clinical-trial-centered design, I would suggest changing Fig 1 to reflect that.

Furthermore, re-sampling is not a computational validation strategy, but a technique to increase your confidence in specific finding. Thus, "computational validation" language should be toned-down.

RESPONSE TO REVIEWERS' COMMENTS

Reviewer #1 (Remarks to the Author):

Comments only for the editors

Reviewer #2 (Remarks to the Author):

I would like to thank the authors for their extensive responses to the concerns I raised and their adaptations to the manuscript. I am satisfied by the answers and changes provided.

We thank the reviewer for the valuable comments, suggestions and positive feedback. We truly believe these constructive comments substantially improved the manuscript.

Reviewer #4 (Remarks to the Author):

The manuscript has been revised and improved. While majority of comments from reviewer 3 have been addressed, some remain unanswered.

We thank reviewer #4 for acknowledging the improvement of our manuscript based on comments of reviewer #3.

For instance, repeatability of the biomarker analysis and effect at the biological levels.

We agree with reviewer #4 that biological signal and its repeatability is an important aspect, which will be addressed in length below. *OncoBird* is specifically designed to include biological priors. To further clarify this, we revised the objective paragraph of *OncoBird* in the **Introduction** as following:

“[...] consecutively, 4) evaluates their predictive component across treatment arms; and finally, 5) it comprehensively corrects for multiple hypothesis testing and **adjusts treatment effects** of biomarkers based on resampling methods. **To enhance the biological signal**, this analysis integrates the molecular and biomarker landscape of cancer clinical trials by customising models to established cancer subtypes and mutational patterns. In essence, *OncoBird* yields subtype-specific biomarkers with

treatment benefits in an interpretable and transparent manner and therefore operates complementary to existing methods. The utility of *OncoBird* is exemplified by the application to the FIRE-3 clinical trial, [...]"

Furthermore, we revised the **Discussion**:

"A limitation of data-driven subgroup analysis is that these may produce spurious results if not biologically interpretable ⁴⁷. For mitigating this risk, we used established tumour subtypes with distinct tumour biology in mCRC, i.e. here the consensus molecular subtypes (CMS) ²⁹ and primary tumour sidedness ²¹. Furthermore, the grouping of functionally similar mutually exclusive somatic mutations in the cancer gene sequencing panel reinforced the identification of biological signals."

In particular, are biomarker levels/thresholds similar across datasets or across two clinical trials that you employed?

We thank the reviewer for pointing out this inaccuracy and lacking information. Across the two analysed studies, we employed *OncoBird*'s default thresholds, i.e. molecular landscape $FDR_{mol} < 0.05$, treatment-specific $FDR_{arm1/arm2} < 0.1$ and interaction analysis $FDR_{int} < 0.2$. Now, we further clarify this in the **Results** section:

"[...] Whilst we particularly focused on the FIRE-3 trial in colorectal cancer, we also demonstrate the generalisability of *OncoBird* by applying it with the same default biomarker thresholds to the ADJUVANT clinical trial (**Methods**), which explores gefitinib in non-small cell lung cancer (NSCLC; **Supplementary Note 1**) ^{35–37}. [...]"

Furthermore, we refined the **Methods** sections to clarify default parameters:

"[...] The FDR cutoff for this analysis step is denoted by FDR_{mol} and controlled at $FDR_{mol} = 0.05$ as our default setting. [...]"

"[...] In total, we found 92 significant associations with the default setting $FDR_{cet/bev} < 0.1$. [...]"

"[...] and further focused on a subset of five biomarkers with default setting $FDR_{int} < 0.2$ for OS, i.e. two gene modules and three single genes (**Table S2**). [...]"

Any exceptions to the default parameters, i.e. the exploration of biomarker trends and the cross-validation due to reduced sample sizes, are further highlighted now:

"[...] For exploring bevacizumab biomarker trends, we employed a lenient threshold of $FDR_{bev} < 0.3$, which deviates from the default setting (**Methods**). [...]"

"[...] For reporting other biomarker trends, we also included summary statistics of 57 subgroups with a lenient threshold of $FDR_{int} < 0.6$, which deviates from the default setting (**Table S1**)."

"[...] For the cross-validation analysis, a more lenient $FDR_{int} < 0.3$ was employed, which deviated from default setting to account for reduced sample sizes in the training and testing splits. [...]"

If not - how will OncoBird deal with this?

We appreciate the reviewer for raising this. We highlight the default parameter settings of *OncoBird* in the method section (see above), and further clearly state that all parameters are adjustable:

"[...] All implemented thresholds of *OncoBird* can be adjusted by the user, thus empowering more lenient or stringent analyses."

It is a valid concern that other datasets might benefit from further customisation of *OncoBird's* parameters and settings. In order to raise awareness, we added the following to the **Discussion** section:

"The statistical power of detecting biomarkers depends on the amount of screened genes and subtypes, sample sizes and magnitude of treatment effects. For example, subtype-specific analyses reduce patient subgroup sizes, thus limiting the power for detecting interactions. In order to gain statistical power to detect genetic biomarkers with low mutational frequency, *Oncobird* exploits mutually exclusive modules (**Methods**). [...]"

If the transcriptomic/genomic levels in different datasets differ - will the biomarkers still be discoverable? Are the results repeatable? Perhaps,

comparison to the housekeeping genes could be performed? (or any other benchmarking).

We believe the reviewer's main concern is the repeatability of biomarkers in altered datasets, and that our previously proposed resampling-based correction methods only partially address this. We agree with this, and in order to address this valid concern, we expanded our benchmarking efforts by applying *OncoBird* in cross-validation for further challenging generalisability and repeatability of biomarkers. For this, we have revised our result section accordingly:

"[...] numerically longer OS was observed for patients with *KRAS* mutated tumours classified as CMS4 treated with bevacizumab (**Fig. 5e**, *KRAS* mutants: $p = 0.24$, HR = 0.66 [0.33-1.31], with a median OS 28.3 months compared to 18.4 months when treated with cetuximab.

In order to assess the ability of *OncoBird* to discover the same biomarkers for different datasets, we applied 5-fold cross-validation repeated five times and extracted the ten most significant biomarkers for OS across each of the 25 models (**Fig. 6a**). Consistent with our previous findings, gene modules containing *KRAS* mutations for CMS4 were found in 21/25 training sets and chr20q amplifications in CMS2 were reproduced in 22/25 training sets (**Fig. 6a**)."

Furthermore, we have added a corresponding visualisation (**Fig. 6a**):

Figure 6: Stability analysis and benchmark with other methods. (a) The ten most significant biomarkers across 25 models of five times repeated 5-fold cross-validation. (b) Boxplots of treatment effects for the predicted subgroups in the test sets for the benchmarked methods including standard treatment guidelines (std) and overall across all patients (null). (c) Oncoprint showing identified subgroups for the benchmarked methods, including std, CMS subtypes, tumour sidedness and mutations in *KRAS* and *NRAS*. (d) Forest plot showing hazard ratios and amount of patients in the subgroup for which standard treatment is not recommended and which was found by subgroup analysis methods (new-std-negative).

Additionally, this enabled us to investigate the generalisability of *OncoBird*. In essence, we concluded that the subgroups discovered by *OncoBird* are likely repeatable for different datasets. We expanded the **Results** section and **Fig. 6b** accordingly:

“For the evaluation, we first derived hazard ratios for cetuximab benefit based on OS in the subgroups according to the predicted biomarkers for all methods across five times 5-fold cross-validation (**Methods**). We also focused on the current treatment guidelines for mCRC, according to which patients should receive cetuximab if their tumours are *RAS* wild type and left-sided (std; **Fig. 6b**)³⁴. While the treatment benefit was not significant for the std-positive subgroup (**Fig. 6b**, median HR = 0.78, $p_{cv} = 0.129$), the methods that found the highest significant benefits were *OncoBird* (median

HR = 0.74, $p_{cv} = 0.046$), POL (median HR = 0.81, $p_{cv} = 0.048$), MOB (median HR = 0.83, $p_{cv} = 0.048$) and OWE (median HR = 0.84, $p_{cv} = 0.049$) ordered by the magnitude of the hazard ratio (**Fig. 6b**).

Next, we leveraged the whole dataset to identify cetuximab sensitivity biomarkers with each method, and compared them to the treatment guidelines. On average, 73% of methods identified cetuximab benefit for a patient in the std-positive subgroup, whereas only 33% of methods detected further benefits in the std-negative subgroup (**Fig. 6c**). [...]"

Accordingly, we also expanded the **Methods** section:

"All benchmarked models were 5-fold cross-validated with five repetitions. A univariate Cox proportional hazards model assessed performances leveraging the treatment effect based on OS in the subgroups with predicted benefits according to the found biomarkers. This included the treatment effect across the whole test set and in the subgroup defined by the current treatment guidelines, i.e. left-sided and *RAS* wild type tumours³⁴. The significance of the treatment effect in the subgroups of the test set was assessed using a modified t-test for resampled performance metrics⁶⁷, denoted by p_{cv} .

For comparing computational methods and their predicted biomarkers, the models were fitted on the whole dataset. The parameterisation of these methods was followed according to the suggested default settings unless in conflict with the above outlined use case. [...]"

It is not clear if Fig 1 describes the design correctly. It looks like randomized clinical trial design should be the first step in the process, followed by measuring treatment response, subtyping, and genetic markers, which are embedded in the clinical trial. Since OncoBird proposed clinical-trial-centered design, I would suggest changing Fig 1 to reflect that.

We agree with the reviewer that this would be more intuitive and have adjusted **Fig. 1a** accordingly:

Figure 1: The Oncology Biomarker Discovery (OncoBird) workflow. (a) Patients in clinical trials were treated with (T) two treatment regimens T with measured clinical endpoints. Subsequently, their tumours are characterised according to (M) tumour genetic alterations (somatic mutations and copy number alterations) and (S) tumour subtypes. [...]

Furthermore, re-sampling is not a computational validation strategy, but a technique to increase your confidence in specific finding. Thus, "computational validation" language should be toned-down.

We apologise for this inaccuracy, and have accordingly toned down "computational validation" to "resampling-based adjustments". We thus have rephrased the sentences in the Results section accordingly:

"[...] 5) it comprehensively corrects for multiple hypothesis testing and adjusts treatment effects of biomarkers based on resampling methods. [...]"

"[...] In more detail, we observed gefitinib benefits in tumours that were characterised by mutations in either TP53, SMAD4 or CDK4 amplifications (p = 0.0002, HR = 0.37

[0.21-0.63]), for which the resampling-based adjustment of the conditional average treatment effect yielded $p_{\text{adj}} = 0.001$ with HR = 0.32 [0.14-0.86] (**Supplementary Note 1**). [...]"

Also we revised the nomenclature in the **Discussion** section:

"[...] In addition, *OncoBird* thoroughly corrects for multiple hypothesis testing and includes resampling-based adjustments of treatment effects. [...]"

REVIEWERS' COMMENTS

Reviewer #4 (Remarks to the Author):

I appreciate authors' careful consideration of the provided comments. I appreciate additional analyses and updated Figures.

The manuscript has been improved and all my comments have been addressed.